# Ultra-high-scale cytometry-based cellular interaction mapping

Dominik Vonficht[1,2,3,27], Lea Jopp-Saile[1,2,3,4,5,27], Schayan Yousefian [4,5,6,27], Viktoria Flore [4,5,6,7,27], Inés Simó Vesperinas[4,5,6], Ruth Teuber[4,5,6], Bogdan Avanesyan [4,5,6], Yanjiang Luo [8,9], Caroline Röthemeier[4,5,6], Florian Grünschläger [1,2,3], Mirian Fernandez-Vaquero[3,10], Vincent Fregona[4,5,6], Diana Ordoñez-Rueda [11], Laura K. Schmalbrock [4,6,12], Luca Deininger [13,14], Angelo Jovin Yamachui Sitcheu [13], Zuguang Gu[15], Maja C. Funk [16], Ralf Mikut [13], Mathias Heikenwälder[10,17], Angelika Eggert[12,18], Arend von Stackelberg[18], Sebastian Kobold [19,20,21], Jan Krönke[6,12,22], Ulrich Keller [6,12,23], Andreas Trumpp[1,2,24], Ahmed N. Hegazy[8,9], Cornelia Eckert [12,18], Daniel Hübschmann [1,15,24,25,26,28] ✉ & Simon Haas [1,2,4,5,6,7,12,28] ✉

Cellular interactions are of fundamental importance, orchestrating organismal development, tissue homeostasis and immunity. Recently, powerful methods that use single-cell genomic technologies to dissect physically interacting cells have been developed. However, these approaches are characterized by low cellular throughput, long processing times and high costs and are typically restricted to predefined cell types. Here we introduce Interact-omics, a cytometry-based framework to accurately map cellular landscapes and cellular interactions across all immune cell types at ultra-high resolution and scale. We demonstrate the utility of our approach to study kinetics, mode of action and personalized response prediction of immunotherapies, and organism-wide shifts in cellular composition and cellular interaction dynamics following infection in vivo. Our scalable framework can be applied a posteriori to existing cytometry datasets or incorporated into newly designed cytometry-based studies to map cellular interactions with a broad range of applications from fundamental biology to applied biomedicine.

Many fundamental processes in life are shaped by physical interactions between cells, including the orchestration of organismal development, tissue homeostasis and immunity[1–4]. Notably, the immune system is one of the most dynamic biological systems in mammals, operating through an exceptionally complex network of intercellular signaling mediators and cell–cell interactions. During immune responses, a highly ordered sequence of antigen-dependent and antigen-independent interactions among various immune cells collectively orchestrates a comprehensive response of the immune system[5]. In this process, transient cellular interactions act as central hubs for information processing and decision making, collectively driving the outcome of immune responses in diverse physiological and pathological states.

While single-cell genomic technologies have substantially advanced our understanding of cellular ecosystems in health and disease, the spatial context of cells in tissues is lost. To overcome this limitation, spatial transcriptomic and high-plex imaging technologies have been developed[6–17]. Although these approaches are powerful in mapping global structures in static tissues, studying transient and

dynamically changing cellular interactions among single cells remains challenging. In particular, transient cellular interactions among immune cells in semisolid or liquid organs such as the blood, or in body fluids such as lymph, urine, cerebrospinal, synovial fluid or saliva, cannot be studied using spatial technologies. In recent years, specialized technologies to study cellular interactions through single-cell transcriptomic profiling of physically interacting cells (PICs) have been developed[18–24]. However, these technologies are limited by their cellular throughput and costs. In parallel, elegant approaches using murine reporter mouse lines have been developed that track past interactions on transient cellular engagement[25–28]. While these technologies are powerful, they are dependent on complex mouse models and are not applicable to study human samples. Therefore, to systematically unravel the dynamic cellular crosstalk of cells across entire organs, organisms and patient cohorts, approaches capable of quantitatively mapping millions of cellular interactions among all cell types of a given biological system at low cost and rapid turnaround times—without the need of complex model systems—are required.

Here we introduce a cytometry-based framework to accurately map both cellular landscapes and physical cellular interactions across all immune cell types at low costs, high speed, high precision and ultra-high scale. We demonstrate the utility of our approach to decipher the kinetics and mode of action of immunotherapies, to derive insights on mechanisms governing therapy response and to disentangle complex, organism-wide immune interaction networks in vivo. Our approach can be readily implemented into any cytometry-based assay with a broad spectrum of applications, ranging from basic biology to advanced immunology, cancer research and applied biomedicine.

## Results

### Cytometry-based quantification of cellular interactions

To develop a universal and flexible cytometry-based framework for mapping physical interactions among immune cells, we first focused on identifying strategies to accurately discriminate between single cells and PICs in cytometry data. For this purpose, we induced a defined set of cellular interactions among human peripheral blood mononuclear cells (PBMCs) using a bispecific antibody-based reagent (CytoStim) that binds both T cell receptors (TCRs) and major histocompatibility complex molecules, thereby physically engaging T cells with antigen-presenting cells (Fig. 1a and Methods). Subsequently, we used an imaging flow cytometer prototype[29] to generate ground-truth data on cellular interactions and concurrently measured cytometry parameters. These comprised five surface markers broadly indicative for distinct immune cell populations, along with a range of image-based and cytometric parameters, such as light scatter profiles (see Supplementary Table 1 for all parameters). Following data acquisition, we manually classified 1,000 randomly selected cellular events based on imaging information across four replicates into singlets, doublets, triplets or higher-plex cell–cell interactions. To extract cytometric features capable of discriminating between single cells and PICs, we performed a feature importance analysis considering the manually classified images as ground-truth data (Fig. 1a,b). This analysis revealed the ratio between signal intensities of forward scatter area and height (termed the forward scatter channel (FSC) ratio), alongside other scatter properties, as highly indicative for singlet to multiplet discrimination, in line with a common gating-based strategy to exclude multiplets from cytometric analyses (Fig. 1b,c). Indeed, by relying solely on the FSC ratio to distinguish singlets from multiplets, an F1 score between 0.50 and 0.84 was achieved, depending on the thresholding method used (Extended Data Fig. 1a and Methods). Notably, we identified Otsu[30]-based thresholding of the FSC ratio as a robust, reproducible and data-driven approach for scatter-based multiplet identification, while alternative thresholding methods produced similar results (Fig. 1d and Extended Data Fig. 1a). However, a portion of cells remained misclassified when using scattering parameters only, particularly affecting cell

types of the myeloid lineage with distinct scatter properties (Fig. 1d and Extended Data Fig. 1b).

To further improve classification and identification of interacting cells, we explored clustering-based approaches for simultaneous multiplet discrimination and annotation. First, we used Louvain[31] clustering using the features identified in the feature importance analysis for the singlet to multiplet discrimination, including image-based parameters (Fig. 1c). This revealed individual clusters single-positive for distinct lineage-defining markers, largely comprising ground-truth single cells of distinct PBMC cell types with low FSC ratio, as well as separate clusters characterized by the coexpression of mutually exclusive lineage-defining markers and a high FSC ratio, largely comprising PICs (Extended Data Fig. 1c–g). Classifying clusters based on FSC ratio into singlets versus multiplets considerably outperformed the approach using scatter properties only (Fig. 1e), and enabled annotation of interacting cell partners based on the coexpression of mutually exclusive lineage-defining markers.

To explore whether such an approach could also be applied to conventional cytometry without image-based information, we performed Louvain clustering on cell type markers only, followed by FSC ratio-based classification into singlet and multiplet clusters (Extended Data Fig. 1h–k). While this approach outperformed FSC ratio only classification, it remained inferior to using all important features (Fig. 1e). In contrast, incorporating both cell type markers and scatter properties—including the FSC ratio—into the clustering, followed by FSC ratio-based classification into singlet and multiplet clusters, yielded results comparable to those achieved when all important features including image-based features were used (Fig. 1e–h and Extended Data Fig. 1l–n). This result was reproducible across various cluster resolutions (Fig. 1i). A comparison between different clustering methods suggested Louvain clustering, alongside others, as an accurate approach (Extended Data Fig. 1o).

Based on these findings, we established the flow cytometry-based Interact-omics framework, which also comprises the computational workflow for the quantification of cellular compositions and physical interactions of cells (PICtR, section 'PIC toolkit for R' in Methods). Briefly, recorded flow cytometry datasets are preprocessed using standard pipelines without multiplet exclusion and are nonuniformly sampled to preserve rare cell types and cellular multiplets (sketching[32]), followed by clustering based on surface marker expression, scatter properties and FSC ratio (Methods). PIC-containing clusters, characterized by predominantly containing events with a high FSC ratio and combinations of mutually exclusive cell-type-specific markers, are selected and used for further downstream analysis, in-depth annotation and quantification. Notably, while Otsu thresholding of the FSC ratio and Louvain clustering are provided as default settings, alternative approaches can be selected (see Extended Data Fig. 1a,o for benchmarking).

Throughout the paper, we present cellular interaction frequencies using any of the following three normalization approaches. First, we report the relative frequencies of cellular interactions among all live, high-quality events, which indicates how prevalent certain interactions are in relation to all cells and other interactions. Second, we present the relative frequencies of a given type of interaction among all interactions, providing insight into how the relative composition of cellular interactions changes across conditions. Third, in scenarios with unbalanced or rapidly changing frequencies of interacting partners, the harmonic mean can be used to calculate the expected interaction frequency based on singlet frequencies, which can then be compared to the observed interactions to assess relative enrichment (Methods). Since these normalization methods address different biological questions, we apply them separately or in combination as appropriate throughout the paper.

Compared to single-cell genomics-based workflows, the ultra-high cellular throughput, rapid processing time and low costs associated

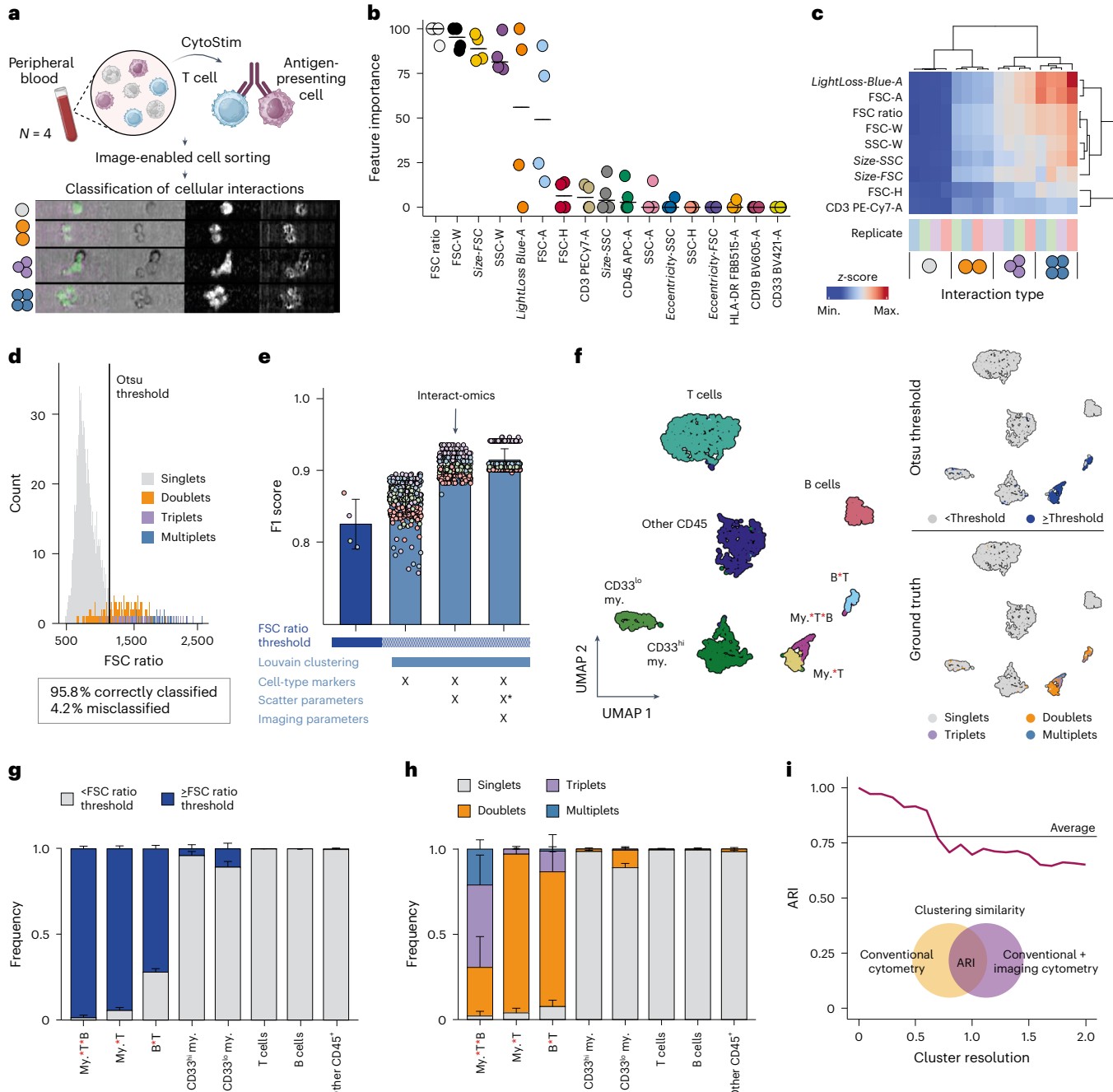

**Fig. 1 | A cytometry-based framework for the accurate identification of physical cellular interactions. a**, Schematic overview of the experimental approach and exemplary ground-truth image data. PBMCs were incubated with the T cell crosslinker CytoStim, followed by manual classification of 1,000 living cells into singlets or multiples across 4 technical replicates. **b**, Importance of features obtained from a decision tree model to classify the data into singlets and multiplets. Features from imaging flow cytometry are written in italic. $n = 4$, horizontal bars indicate the median. **c**, Heatmap of most important features, colored by mean $z$-score of features across replicates grouped into singlets, doublets, triplets and multiplets. **d**, FSC ratio histogram, colored by the ground-truth annotation. The classification into singlets and multiplets by Otsu[30] thresholding is shown. **e**, Performance of different classification methods as measured by the F1 score. In all methods displayed, cells were categorized by Otsu thresholding of the FSC ratio. The first method (dark blue) relies on Otsu thresholding of the FSC ratio only, and all others (light blue) involve Louvain[31] clustering based on different feature sets as indicated below the $x$ axis, followed by assertion of clusters to either singlets or multiplets based on the proportion of cells exceeding the FSC ratio threshold. The third bar represents the Interact-

omics workflow. Louvain clustering was performed for $n = 100$ iterations, and the results for each technical replicate ($n = 4$) are shown in the point plot. Bars indicate the mean F1 score. X* shows only the most important scatter parameters were used (**c**). **f**, Left: UMAP[55] embedding of classified cells ($n = 3,865$) based on conventional flow cytometry parameters, including cell type markers, scatter parameters and the FSC ratio. Right: UMAP embeddings with cells exceeding the Otsu threshold of the FSC ratio highlighted in blue (top) or cells colored by their ground-truth annotation (bottom). **g**, Relative frequency of cells classified according to the FSC ratio. $n = 4$, error bars indicate the standard deviation. **h**, Relative frequency of singlets and interacting cells based on the ground-truth annotation. $n = 4$, error bars indicate the standard deviation. **i**, Adjusted rand index (ARI) of consensus clustering solutions obtained for (1) the important features shown in **c** and cell type markers versus (2) only conventional cytometry features as used in **f** for different resolutions in Louvain clustering. Clustering was performed for $n = 100$ iterations at each resolution. A, area; H, height; max., maximum; min., minimum; my., myeloid; SSC, side scatter; W, width. Interactions between cell types are encoded by a red asterisk between the two cell type labels. Panel **a** created with BioRender.com.

with the presented cytometry-based approach enable the seamless analysis of millions of cellular events within short time periods. Note that the Interact-omics framework is designed to specifically dissect heterotypic PICs and relies on carefully chosen case-control settings with stable experimental conditions to determine an enrichment of true PICs above baseline interactions. A detailed description about the technical aspects affecting the formation of PICs is provided in the 'Limitations and guidelines' section.

### Cellular interaction mapping of complex immune landscapes

To simultaneously map cellular composition and cellular interactions in complex immune landscapes at high resolution, we established ultra-high parametric, data-informed flow cytometry assays for mouse and human. To optimize cell type resolution across all common blood and immune cell populations, we leveraged single-cell proteo-genomic datasets[33,34] to identify optimally discriminating cell type- and cell state-specific markers. Moreover, to enable the simultaneous detection of mutually exclusive cell type markers in multiplets, we assigned cell-type-specific markers to fluorophores with low spectral overlap to reduce spreading errors. While our approach can be applied to standard flow cytometry-based assays, we used full-spectrum flow cytometry[35] due to its superior capacity to disentangle high-plex marker panels. Applying the Interact-omics framework with such an optimized 24-plex panel to human PBMCs revealed an accurate representation of the CytoStim-induced changes in cellular composition and cellular interactions at cell type and cell state resolution (Fig. 2a–e and Extended Data Fig. 2a–f). As expected, interactions between various T cell subsets and antigen-presenting cell populations significantly increased upon CytoStim treatment, whereas other cellular interactions remained unaffected or decreased (Fig. 2d,e). Notably, the results were highly reproducible across replicates and interactions among rare populations could be accurately quantified, including multiple T cell subset and dendritic cell interactions.

Next, to investigate whether the Interact-omics framework is capable of resolving antigen-dependent immune cell interactions, we isolated CD4 T cells carrying a transgenic TCR specific for chicken ovalbumin (OVA) from OT-II mice and cocultured them in the presence or absence of its cognate antigen with a complex cellular mixture of murine splenocytes (Fig. 2f). As expected, cellular interactions between OVA-specific CD4 T cells and a range of antigen-presenting cells were specifically induced in the presence of the respective antigen, whereas cellular interactions of bystander cells remained unaffected or changed only mildly (Fig. 2g–j and Extended Data Fig. 3a–i). Together, these results demonstrate the utility of our approach in resolving antigen-dependent and -independent cellular interactions across complex immune landscapes with a broad range of potential applications.

To evaluate the effects of various experimental conditions on the nonspecific formation of cellular interactions, we conducted a series of ex vivo benchmarking experiments (Supplementary Note 1 and Supplementary Fig. 1). The results demonstrated that ex vivo-induced cellular interactions are relatively stable but highlighted the critical importance of maintaining consistent experimental parameters, including cell concentrations, processing times and cytometer settings, to ensure reliable and reproducible outcomes while limiting technical artifacts.

### Dissecting the mechanism and kinetics of immunotherapies

The molecular mode of action of most cancer immunotherapies is based on the redirection of cancer–immune cell interactions. For example, bispecific antibodies engage cancer cells with immune cells, whereas chimeric antigen receptor (CAR)-T cells are engineered T cells that specifically target epitopes present on cancer cells. To investigate whether the Interact-omics framework is capable of resolving CAR-T-cell-mediated cellular interactions, we used engineered green fluorescent protein (GFP)-tagged murine CAR-T cells targeting CD19-expressing cells in cocultures with murine splenocytes

(Fig. 3a). As expected, our analyses revealed that both CD4 and CD8 CAR-T cell subsets rapidly engaged in specific interactions with CD19-expressing target B cells (Fig. 3b–d and Extended Data Fig. 4a–h). As a consequence, CAR-T cell interactions with B cells were highly enriched when compared to interactions between B cells and endogenous T cells (Fig. 3e), reaching a maximum at 1 hour post-treatment, followed by a gradual decline (Extended Data Fig. 4i–l).

Bispecific antibodies engage T cells with tumor cells. Blinatumomab, which engages CD3-positive T cells with CD19-positive (malignant) B cells is a clinically approved immunotherapy[36,37]. To investigate whether the Interact-omics framework is capable of resolving blinatumomab-induced cellular interactions, we treated human PBMCs with blinatumomab ex vivo (Fig. 3f). As expected, blinatumomab induced a strong increase in cellular interactions among a range of B and T cell populations, peaking 1 hour post-treatment followed by a gradual decline of interactions over time (Fig. 3g–k and Extended Data Fig. 5a–j). As expected, the transient increase in cellular interactions of B cells was mirrored by a transient decrease of free single B cells and a time-delayed decrease in overall B cell-containing events, suggesting a rapid engagement of B and T cells, likely followed by a mild cytotoxic effect induced by blinatumomab (Fig. 3j and Extended Data Fig. 5k). In contrast to blinatumomab-induced B cell–T cell interactions, interactions among other cell types remained unaffected, demonstrating the specificity of the interactions (Fig. 3k).

Notably, following chemical fixation, the quantification of cellular interactions induced by blinatumomab remained unaffected by freeze–thawing associated cryogenic preservation (cryopreservation), enabling a broad range of applications with primary patient material (Supplementary Fig. 1j). Together, these analyses demonstrate the broad utility of the Interact-omics framework to characterize cellular interactions induced by immunotherapies.

### Interact-omics reveals immunotherapy response features

Blinatumomab has been approved for the treatment of B cell acute lymphoblastic leukemia (B-ALL), the most common type of cancer in children, at relapsed or refractory stages[38]. Although blinatumomab is progressing toward becoming the standard of care for relapsed and refractory pediatric ALL, the response rates remain heterogeneous[36,37]. While few clinical and molecular parameters have been associated with outcome to blinatumomab therapy, the underlying mechanisms remain poorly understood and a robust test predicting therapy response is lacking[39–44]. To evaluate whether the Interact-omics framework can be used to extract parameters associated with therapy response, we acquired bone marrow (BM) aspirates from 42 pediatric patients with relapsed B-ALL before blinatumomab treatment. Subsequently, we applied the Interact-omics framework using an adjusted panel on the samples in the presence or absence of ex vivo blinatumomab treatment (Fig. 4a–c). We extracted a range of parameters from the data, including cellular frequencies of singlet populations in the absence of treatment and the induction of cellular interactions on ex vivo blinatumomab treatment (Fig. 4b,c). To explore mechanisms underlying therapy response among patients with residual disease, we compared patients who could unequivocally be categorized into good responders ($n = 18$) and nonresponders ($n = 4$) (Fig. 4d).

In line with previous studies[40,41], high frequencies of various T cell subsets, particularly central and effector memory subsets, were associated with good response to blinatumomab (Fig. 4d,e). However, also the frequencies of cellular interactions before treatment or upon blinatumomab were associated with therapy response (Fig. 4d). For instance, blinatumomab induced interactions of B and T cells, and B, T and myeloid cells more efficiently in good responders compared to nonresponders (fold change good versus nonresponders B–T(–myeloid); Fig. 4d,e). Blinatumomab failed to induce effective B–T cell interactions in patient samples with unbalanced T cell to B cell ratios (Fig. 4f). Similarly, in patient samples with high T–myeloid interactions at baseline,

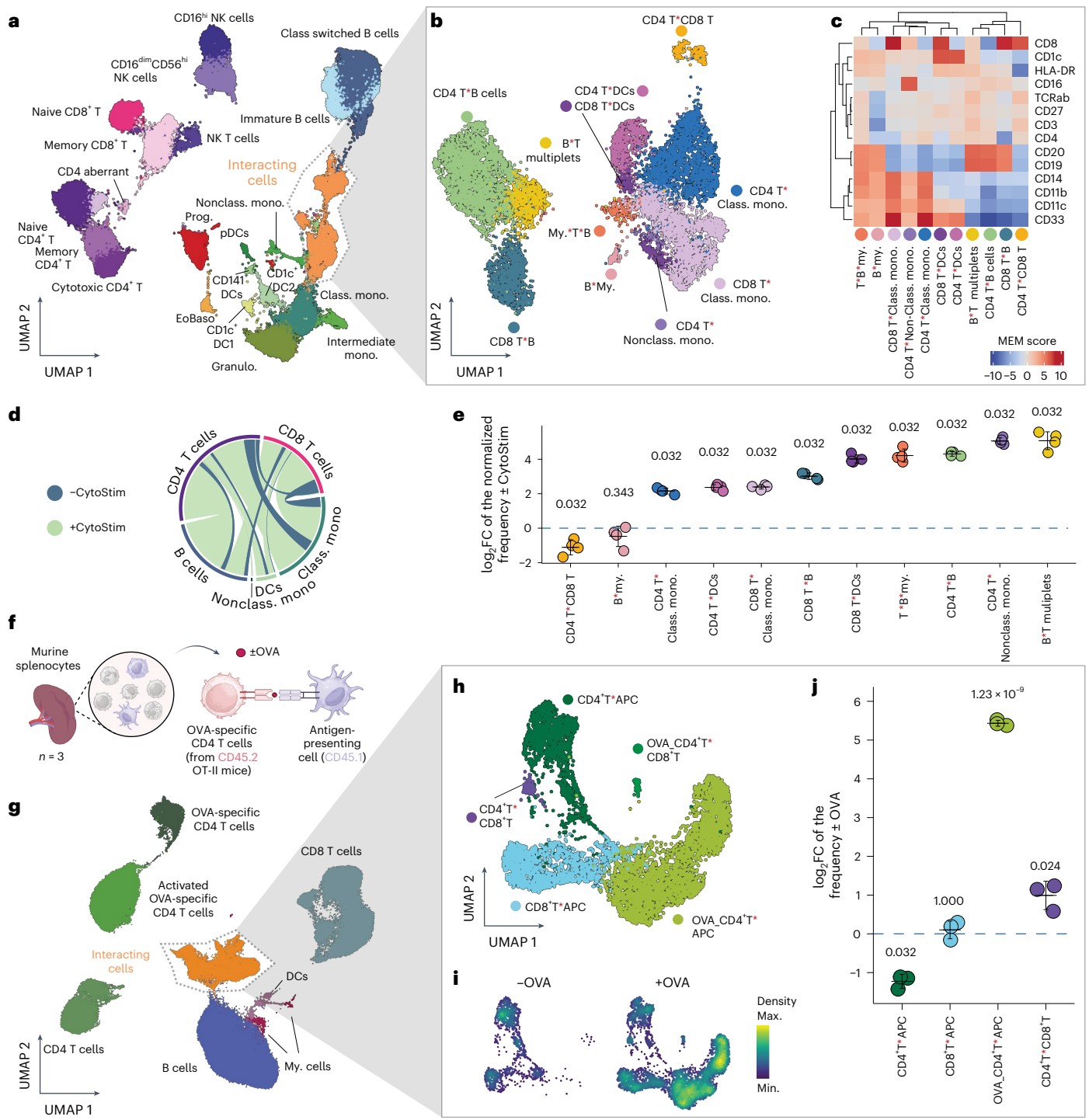

**Fig. 2 | Ultra-high-scale cellular interaction mapping across complex immune landscapes. a**, UMAP display of a 25-plex cytometry dataset of PBMCs cultured in the presence or absence of the crosslinking agent CytoStim, $n = 4$ replicates from a single donor. Recorded cells were processed with PICtR; out of 226,301 cells, 50,000 sketched cells are displayed. Interacting cells are depicted in orange. **b**, UMAP of interacting cells ($n = 9,988$). **c**, Heatmap colored by marker enrichment modeling[56] score of cell type defining markers across the clusters of cellular interactions. **d**, Circos plot displaying the relative enrichment between T and antigen-presenting cells. Colors of the contributing singlets (highlighted on the circumference) are analogous to **a**. **e**, Point plots depicting log$_2$ fold changes (FC) of normalized interactions between the CytoStim treated and untreated conditions. Interaction frequencies were adjusted for the singlet frequencies of the contributing cells (harmonic mean; Methods). *P* values were determined with a two-sided Wilcoxon rank sum test and adjusted for multiple testing according to Benjamini–Hochberg. Error bars indicate mean and standard deviation. $n = 4$ replicates from a single donor. **f**, Schematic overview of the experimental setup of cocultures of chicken OVA-specific OT-II CD4 T cells with murine splenocytes. **g**, UMAP of the overall cellular landscape; $n = 125,554$ events. **h**, UMAP of the interacting cell landscape; $n = 6,399$. **i**, Point density UMAP of H split into the treatment conditions. **j**, The log$_2$FC of frequencies of interacting cells in the presence versus absence of OVA. OVA_CD4+T*CD8+T interactions are not depicted, as they appeared exclusively upon OVA treatment. The *P* values were calculated using least squared means[57] (two-sided) and were Bonferroni corrected. Error bars indicate mean and standard deviation. $n = 3$ technical replicates. class., classical; DCs, dendritic cells; EoBaso, eosinophils and basophils; granulo., granulocytes; MEM, marker enrichment modeling; mono., monocytes; nonclass., nonclassical; pDCs, plasmacytoid dendritic cells; prog., progenitors. Red asterisks in cell type labels indicate interactions between the respective cell types. Panel **f** created with BioRender.com.

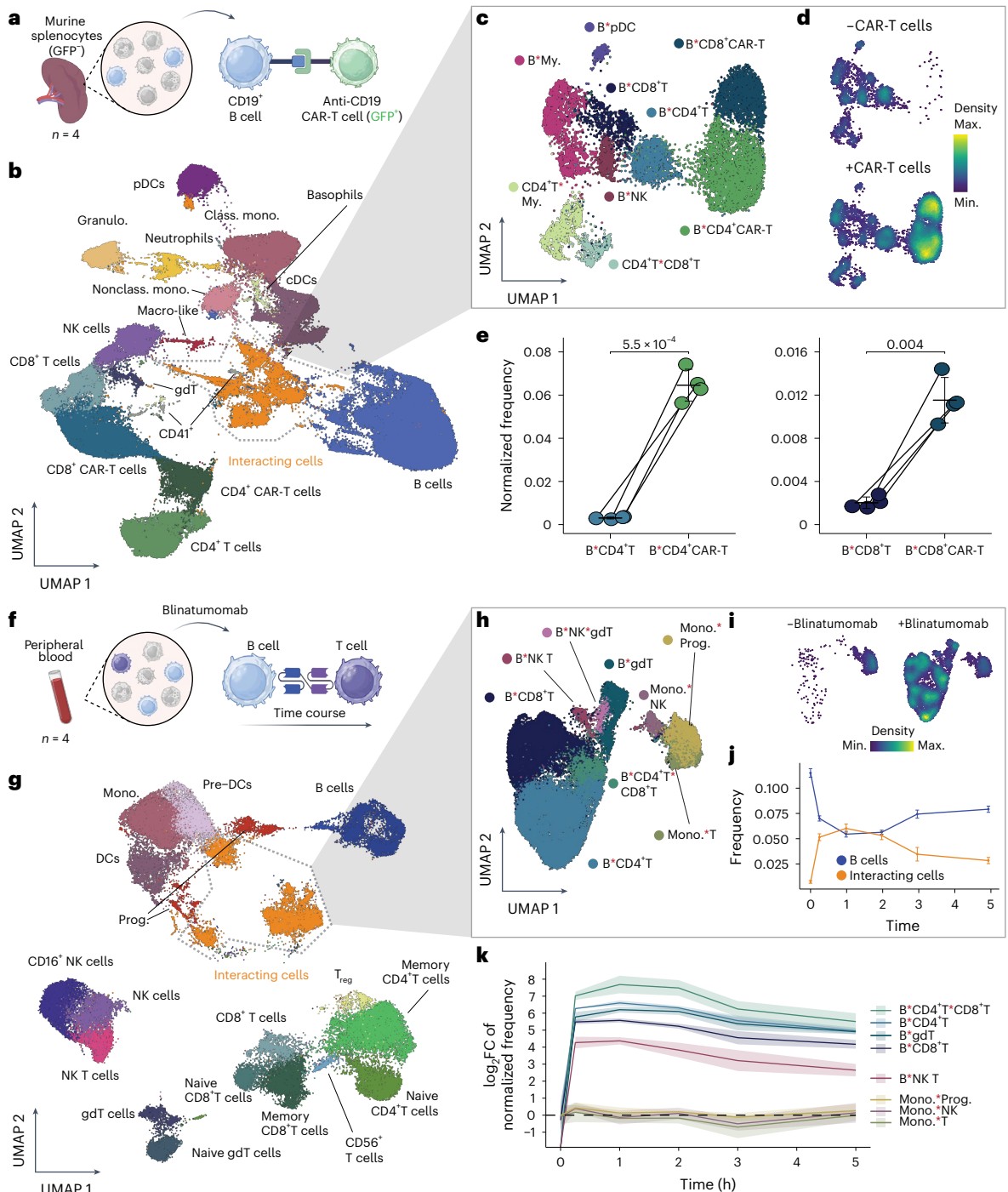

**Fig. 3 | Cellular interaction mapping reveals mechanisms and kinetics of immunotherapies. a**, Schematic overview of the experimental setup of cultures from murine splenocytes and anti-CD19 CAR-T cells. **b**, UMAP of the overall cellular landscape. Recorded cells were processed with PICtR; out of 849,845 cells, 96,988 sketched cells are displayed. **c**, UMAP of the interacting cell landscape, $n = 9,974$. **d**, Point density UMAP of **c** in the absence of CAR-T cells (top) and 1 h after adding CAR-T cells (bottom). **e**, Paired analysis of interactions between B cells and CAR-T cells or B cells and endogenous T cells. Interaction frequencies were normalized by the harmonic mean of the singlet frequencies of the contributing cells (Methods). Paired two-sided Welch's $t$-test, $n = 4$ technical replicates, error bars indicate the mean and standard deviation. **f**, Schematic overview of the experimental setup for investigating cellular interactions upon treatment with blinatumomab. $n = 4$ replicates from a single donor. **g**, UMAP of the overall cellular landscape. Recorded cells were processed with PICtR; out of 985,735 cells, 49,210 sketched cells are displayed. **h**, UMAP of the interacting cell landscape, $n = 34,362$. **i**, Point density UMAP of **h**, in the absence of blinatumomab (left) and 1 h after blinatumomab treatment (right). **j**, Comparison of the B cell frequencies and frequencies of cellular interactions involving B cells over time. $n = 4$ technical replicates, error bars indicate the standard deviation. **k**, Time-resolved $\log_2$FC of distinct cellular interaction frequencies. Interaction frequencies were normalized by the harmonic mean of the singlet frequencies of the contributing cells (Methods). $n = 4$ technical replicates, the shaded area shows the standard deviation. gdT, gamma–delta T cells; macro-like, macrophage-like; $T_{reg}$, regulatory T cells. Panels **a** and **f** created with BioRender.com.

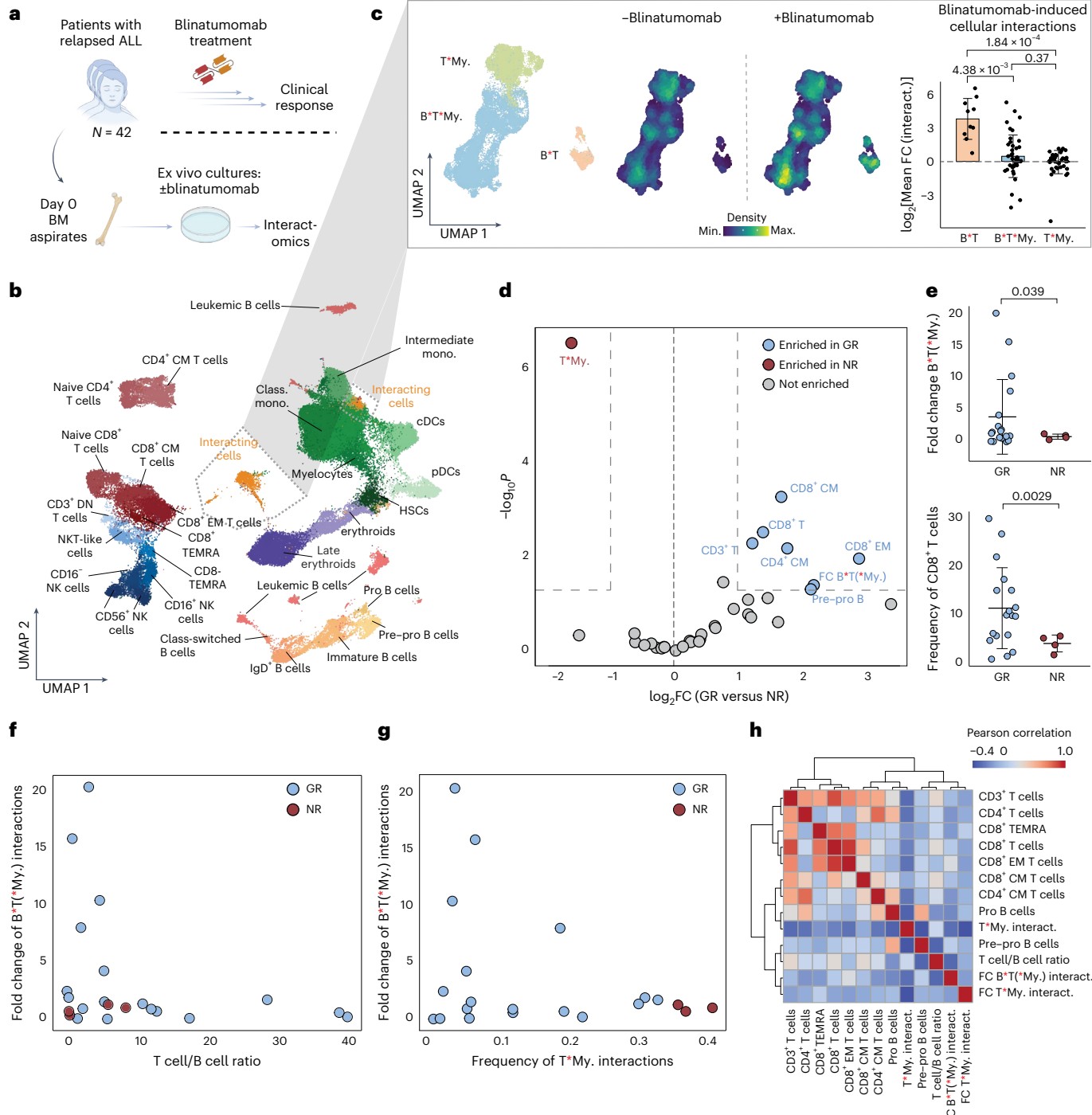

**Fig. 4 | Interact-omics reveals features underlying therapy response to blinatumomab. a**, Schematic overview of the experimental setup of ex vivo treated B-ALL BM aspirates with blinatumomab. **b**, UMAP of the overall cellular landscape. Recorded cells were processed with PICtR; out of 4,292,770 cells, 70,000 sketched cells are displayed. Patient-specific leukemic clusters were merged into a common meta-cluster. **c**, UMAP of interacting cells (*n* = 29,232). Point density UMAP of interacting cells split into the treatment conditions (middle). Bar graph illustrating the log$_2$ mean fold change of cellular interactions (interact.) (blinatumomab-treated versus control) (right). Error bars indicate the standard deviation. B and T cell interactions (B*T), *n* = 10; B–T–myeloid (B*T*My.) interactions, *n* = 42; T*My., *n* = 41 biological replicates (patients). *P* values were determined with a two-sided Welch's *t*-test. Bonferroni-adjusted *P* values are displayed. **d**, Volcano plot representing enrichment and depletion for good responders versus nonresponders for both singlets and cellular interactions. A two-sample *t*-test (two-sided) was applied. **e**, Top: point plot showing the comparison of the fold

change of B*T(*My.) interactions after ex vivo blinatumomab treatment in GR and NR. Interaction frequencies (freq.) were adjusted for singlet frequencies of the contributing cells (harmonic mean; Methods). Bottom: point plot showing the frequency of singlet CD8$^+$ T cells in GR and NR. *P* values were determined with a *t*-test (two-sided). Error bars indicate the mean and standard deviation. GR, *n* = 18; NR, *n* = 4 biological replicates (patients). **f**, Scatter plots displaying the fold change of B*T(*My.) interactions on blinatumomab treatment against the singlet T/B cell ratio at baseline. **g**, Scatter plot displaying the fold change of B*T(*My.) interactions against the frequency of T*My. interactions at baseline. **h**, Heatmap of Pearson correlation coefficients between various features, including frequencies of singlets and cellular interactions as well as fold change induction of cellular interactions after blinatumomab treatment. GR, good responder; NR, nonresponder; TEMRA, terminally differentiated effector memory T cells. Red asterisks in cell type labels indicate interactions between the respective cell types. Panel **a** created with BioRender.com.

blinatumomab treatment failed to effectively induce B–T cell interactions, suggesting that T–myeloid interactions may inhibit or compete with B–T cell interactions (Fig. 4g). Accordingly, high T–myeloid interactions at baseline were associated with therapy failure (Fig. 4d). A correlation analysis of selected parameters associated with therapy response revealed that singlet frequencies of T cell subsets were highly correlated among each other, whereas cellular interactions provided independent and additive information on therapy response (Fig. 4h and Extended Data Fig. 6). Jointly, these analyses provide new insights into the cellular mechanisms mediating response to blinatumomab and may lay the foundation for personalized therapy response prediction.

## Downstream signaling as a consequence of cellular interactions

To evaluate whether Interact-omics can be used to study intracellular signaling in response to cellular interactions, we established a high-plex cytometry panel that includes an antibody detecting phosphorylation (pY142) of intracellular CD3 zeta (CD247), a transmembrane signaling adapter protein phosphorylated upon TCR signaling and T cell activation (Extended Data Fig. 7a). Using this panel, we investigated intracellular TCR signaling in cellular interactions induced in human PBMCs following CytoStim (crosslinks antigen-presenting cells with T cells) and blinatumomab (crosslinks B cells with T cells) treatment.

Consistent with our previous results and the molecular mechanisms of the inducers used, we observed few background interactions at homeostasis but noted specific induction of B–T cell interactions in response to blinatumomab treatment and broader myeloid and B cell interactions with T cells after CytoStim treatment (Extended Data Fig. 7b–d). As expected, CytoStim-induced interactions caused strong phosphorylation of the intracellular CD3 zeta domain in both T–B and T–myeloid interactions, as well as in T–B–myeloid triplets, demonstrating functional TCR engagement in the interacting T cells (Extended Data Fig. 7e,f). In line with its more specific crosslinking activities, blinatumomab caused a specific increase in phosphorylation of the intracellular CD3 zeta domain in T cells involved in interactions with B cells, but to a much lower degree in interactions not involving B cells. Collectively, these findings demonstrate that our approach can be used to study intracellular signaling in response to cellular interactions.

## Organism-wide interaction mapping of viral infections

Infectious agents and pathogens induce complex cascades of organ-specific immune reactions in vivo, comprising cell–cell interactions, cell expansion and cellular trafficking, jointly establishing first line defense, long-lasting adaptive immunity and hematopoietic recovery after pathogen insult. However, our comprehension of such pathogen-induced cellular immune dynamics remains limited due to current technological restrictions. In particular, there is a lack of quantitative insights into organotypic differences in the composition, order and kinetics of cellular interactions induced following pathogen exposure in vivo.

The lymphocytic choriomeningitis virus (LCMV) serves as a well-established murine model pathogen to study key questions in immunology, including the induction of innate and adaptive immunity, pathologic consequences of virus infections, immune evasion mechanisms and virus-induced suppression of hematopoiesis[45,46]. To systematically unravel LCMV-induced alterations in the immune cell and cellular interaction networks across distinct organ systems, we applied the Interact-omics workflow to mesenteric lymph nodes (LNs), spleens and BM of mice at days 0 (naive), 3 and 7 after intraperitoneal LCMV infection (Fig. 5a). To discriminate cellular interactions mediated by antigen-dependent and -independent mechanisms, we transferred congenic, LCMV-specific CD4 and CD8 T cells recognizing epitopes of the LCMV glycoprotein into mice 5 days before infection (Fig. 5a) and included congenic markers (SMARTA:CD90.1; P14:CD45.1) in our cytometry panel (Methods). In total, we quantified more than

34 million single cells from 21 cell types, and around 415,000 cellular interactions from 52 cell type pairs, across 36 samples (Fig. 5b,c and Extended Data Fig. 8). Notably, LCMV infection caused a wide range of alterations in cellular composition and cellular interactions. Principal component analysis (PCA) of cellular abundances and interactions revealed organ- and time-specific changes that were highly reproducible across replicates, demonstrating the robustness of our approach (Fig. 5d,e). To assess the reliability of interactions derived from in vivo settings, we performed extensive benchmarking using imaging flow cytometry and colabeling experiments (Supplementary Note 2 and Supplementary Figs. 2 and 3). The findings revealed that, although extra interactions may be acquired during sample preparation, these interactions are nonrandom, reflect underlying biological effects and probably provide a reliable proxy for cellular interactions occurring in vivo.

Clustering cellular interactions according to their virus-induced alterations over time revealed groups with distinct patterns of interaction dynamics (Fig. 6a and Extended Data Fig. 9). For example, cellular interactions in cluster 4 were rapidly inducible at day 3 and partially normalized toward day 7 post-infection. Cellular interactions in this cluster comprised mainly cell types of the innate arm of the immune system (for example, natural killer (NK) cells, monocytes, macrophages), in line with their rapid response and key role in first line defense, as well as few nonantigen-specific adaptive immune cells. In contrast, clusters 3 and 5 contained a variety of interactions comprising LCMV-specific T cells, which displayed a delayed but pronounced induction of cellular abundances and interactions at day 7, in line with their well-documented response pattern (Fig. 6a,b). Notably, LCMV-specific T cell interactions were more pronounced in spleen when compared to mesenteric LNs (Fig. 6c), likely reflecting a more rapid uptake of LCMV into the spleen after intraperitoneal administration, as previously described[47]. LCMV-specific T cells were also detected in the BM (Fig. 6b), in line with the notion that BM may serve as primary immune organ[48]. However, LCMV-specific BM T cells were less likely to engage in cellular interactions compared to their non-LCMV-specific T cell counterparts, as indicated by a negative odds ratio, taking their singlet frequencies into account (Fig. 6d). In contrast, LCMV-specific T cells in LN and spleen were more likely to engage in cellular interactions when compared to their non-LCMV-specific counterparts, in line with the key role of LNs and spleen in the orchestration of adaptive immune responses (Fig. 6d).

In LCMV infections, BM myelosuppression is associated with a transient activation of NK cells, peaking at day 3 post-infection, followed by a rapid recovery[49,50]. In line with this, we observed a massive increase in NK cell interactions with cells of the myeloid lineage, including myeloid progenitors, peaking at day 3 post-infection in BM samples (Fig. 6e). Subsequently, NK cell–myeloid interactions decreased, followed by an expansion of hematopoietic stem and progenitor cells (HSPCs) and BM monocytes at day 7 (Fig. 6e), suggesting a switch from myelosuppression to active emergency hematopoiesis, in line with previously reported kinetics of LCMV-induced myelopoiesis[51].

Notably, on infection, a rapid infiltration of monocytes into LNs and spleens was observed (Fig. 6f). Recruited monocytes readily engaged with LN and spleen B cells at day 3, partially normalizing at day 7 (Fig. 6g). Such extensive monocyte–B cell interactions have recently been described to serve as an LCMV-specific immune evasion mechanism, hindering early B cell responses in a chronic model of LCMV infection[52]. In line with this, increased interactions between plasmablasts and LCMV-specific CD4 T cells in LN and spleen samples, as well as an expansion of plasma cells coincided with the disappearance of suppressive monocyte–B cell interactions at day 7 (Fig. 6h,i).

Together, these results demonstrate the utility of the Interact-omics approach for dissecting complex immune interaction networks in vivo. Our data accurately recapitulate previous findings and provide a quantitative framework for a systems-level understanding of

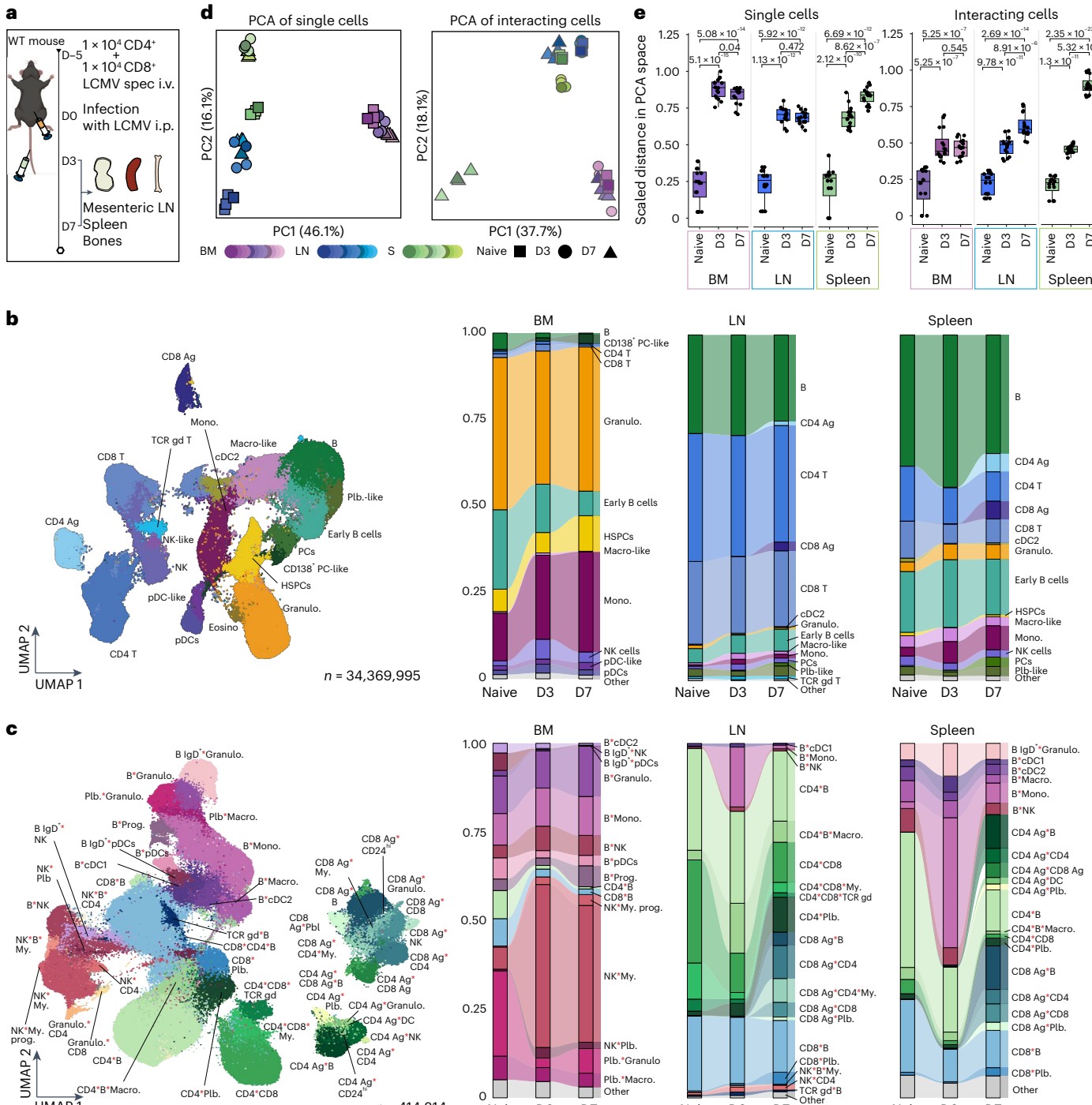

**Fig. 5 | Virus-induced alterations of cellular landscapes and interaction networks. a**, Schematic overview of the experimental design. $n = 4$ biological replicates at each time point. **b**, Left: UMAP display of the cellular landscape. Recorded cells were processed with PICtR; out of 34,369,995 cells, 262,628 sketched single cells are displayed. Right: alluvial plots depicting the change of single-cell frequencies over time and across organs. **c**, Left: UMAP displaying the interacting cell landscape ($n = 414,564$). Right: alluvial plots depicting the change of interacting cell frequencies over time and across organs. **d**, PCA of single-cell and interacting cell frequencies across organs and time points, encoded by color and shape, respectively. **e**, Scaled Euclidean distances from the mean naive state to all samples in PCA space, representing global similarities or differences in single-cell and interaction landscapes. *P* values were calculated with a two-sided *t*-test and adjusted according to Benjamini–Hochberg. Error bars indicate the mean and standard deviation. $n = 16$, box plots display the median, and first and third quartiles and whiskers are defined as 1.5 times interquartile range. D3, day 3; D7, day 7; Ag, LCMV antigen-specific; IgD, immunoglobulin D; i.p., intraperitoneal; i.v., intravenous; macro., macrophages; PCs, plasma cells; PC1 or PC2, principal component 1 or 2; Plb., plasmablast; spec, specific.

virus-induced alterations of the cellular immune interaction networks and how they cooperate across organ systems to elicit intricate immune responses. However, limitations outlined in Supplementary Note 2 and the 'Limitations and guidelines' section should be considered.

## Application to existing cytometry data

To assess the applicability of our approach for analyzing cellular interactions in existing datasets, we applied the PICtR workflow to two publicly available cytometry datasets[53,54] (Supplementary Note 3). In a juvenile

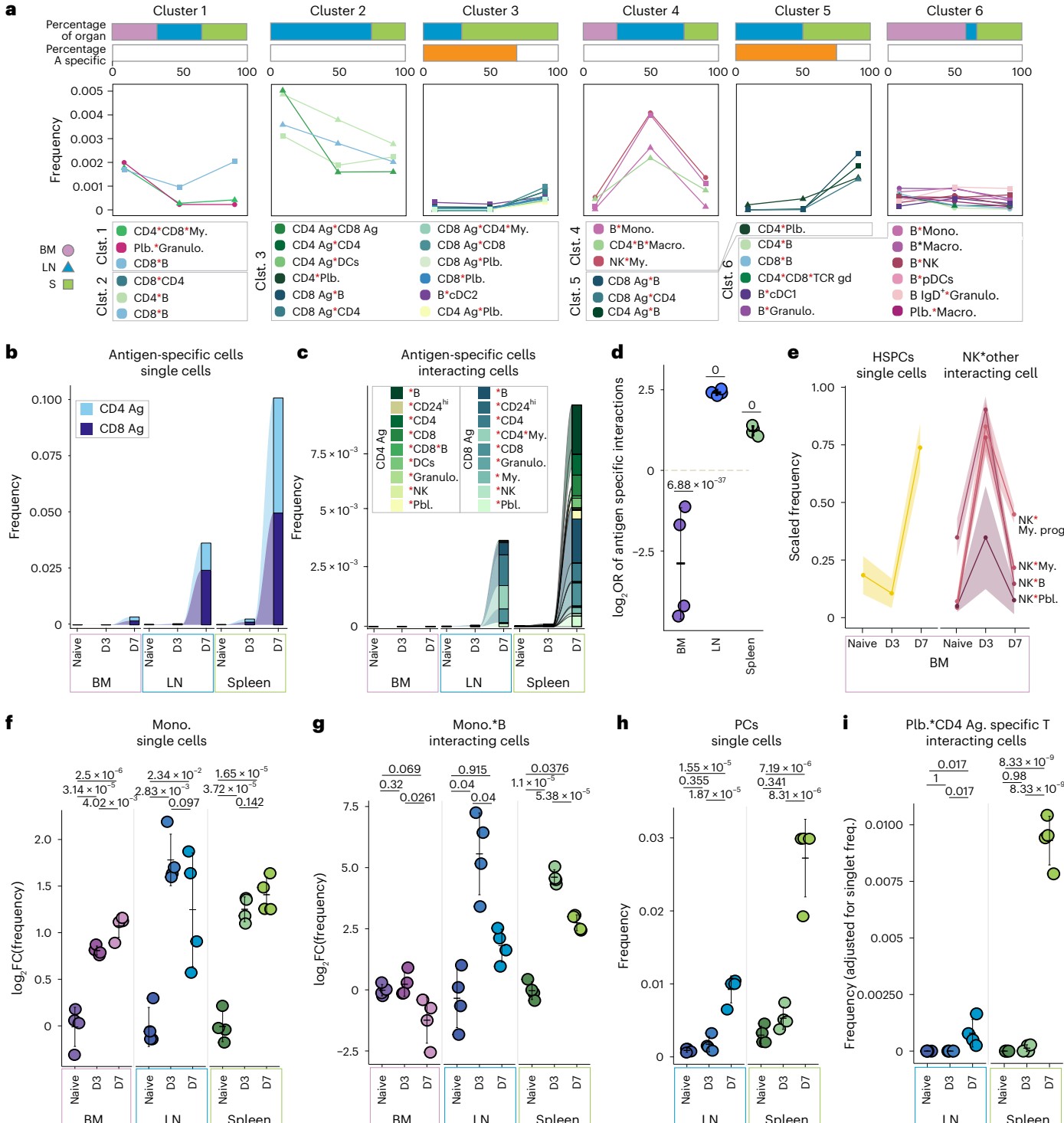

**Fig. 6 | Cellular interaction dynamics underlying immune response to LCMV infection. a**, Line plots depicting the frequency of cell types across time points and organs; obtained from k-means clustering (k = 7). Clusters with no change in dynamics are not shown. Horizontal bar plots at the top indicate the percentage of interactions contributing to each organ and LCMV-specific cells for each cluster (clst.). n = 4 biological replicates. **b**, Alluvial plots showing the fraction of LCMV-specific CD4⁺ and CD8⁺ T cells. **c**, Alluvial plots showing the fraction of cellular interactions comprising LCMV-specific CD4⁺ and CD8⁺ T cells. **d**, Point plots displaying the log₂OR enrichment or depletion of LCMV-specific T cell interaction against nonantigen-specific T cell interactions relative to corresponding singlet population on day seven across the organs. P values were calculated using Fisher's exact test (two-sided) on the sums of the interacting and single cells, aggregated across replicates per condition (n = 4), for antigen-specific and nonantigen-specific cells. Error bars indicate mean and standard deviation. P values: BM, 6.88 × 10⁻³⁷; LN, 0; spleen, 0. **e**, Line plots showing the scaled fraction of HSPCs and of NK−myeloid cellular interactions in the BM. Error bars depict the standard error of the mean. **f**, Point plots depicting the log₂FC of monocytes on each day versus naive. **g**, Point plots showing the log₂FC monocyte−B cell interactions monocytes on each day versus naive. **h**, Point plot showing the frequency of single plasma cells. **i**, Point plot showing the interacting cell frequency (adjusted for the respective single-cell frequencies) between plasmablasts and LCMV-specific CD4⁺ T cells. In **f**−**i**, P values were calculated using a two-sided least squared means test and corrected according to Benjamini−Hochberg. Error bars indicate mean and standard deviation and n = 4 biological replicates. Red asterisks in cell type labels indicate interactions between the respective cell types.

idiopathic arthritis dataset[53], we identified cellular interactions linked to disease activity and tissue localization (Supplementary Note 3 and Supplementary Fig. 4). Applying our framework to a dataset[54] from the murine proximal intestine demonstrated its utility for mapping interactions with nonimmune cells (Supplementary Note 3 and Supplementary Fig. 5). These analyses demonstrate that our approach is adaptable to existing datasets, provided the data acquisition followed the guidelines outlined in this paper (section on 'Limitations and guidelines').

## Discussion

Here we introduce Interact-omics, a highly flexible and scalable cytometry-based framework for the joint mapping of cellular landscapes, such as the immune system, and their physical interactions. We demonstrate its utility in deciphering the kinetics, mode of action and response mechanisms of immunotherapies, and for the quantitative dissection of complex, organism-wide immune interactions networks in vivo.

In contrast to current methods for mapping physical interactions of cells, Interact-omics excels in throughput, cost effectiveness, processing times, required technical prerequisites and ease of implementation. In fact, the Interact-omics framework can be used in conjunction with any multicolor fluorescence flow cytometer and our analytical PICtR pipeline can be applied to mine cellular interactions both in newly acquired and pre-existing cytometry datasets. In contrast to recently developed technologies that map past cellular encounters using transgenically engineered mouse models[25–28,52], Interact-omics can be readily applied to any cellular suspension that is compatible with flow cytometry analysis, and does not rely on reporter mouse lines. We have demonstrated that physical interactions of cells are detected on freeze–thawing and can be stabilized by chemical fixation, enabling the implementation of the Interact-omics framework for the study of biobanked patient material. Notably, the costs for cellular interaction mapping using the Interact-omics framework are orders of magnitudes lower when compared to single-cell genomics-based technologies, while its throughput is orders of magnitudes higher. This enables the study of cellular interactions in currently unexplored settings, such as high-throughput screens, extensive time course experiments, organism-wide studies and large patient cohorts. While the approach presented here is optimized for analyzing cellular interactions among immune cells, the Interact-omics framework can also be used for studying interactions across other cell types, assuming careful panel design and the adaptation of sample processing strategies to minimize technical interactions. Jointly, the aforementioned features render the Interact-omics framework broadly applicable to any research field where alterations in cellular frequencies and interactions may place decisive roles. These encompass, but are not limited to, basic immunology, autoimmune diseases, cancer research, infectious diseases, drug development and personalized medicine.

While the Interact-omics framework focuses on quantifying cellular interactions at ultra-high scale, consequences of cellular interactions can be derived by including flow-based readouts, such as activation and exhaustion markers, phosphorylation status of signal transducers, or by complementing it with lower throughput single-cell genomics-based methods for cellular interaction mapping, such PIC-seq[24] or others[19–23].

Furthermore, we have demonstrated the utility of the Interact-omics workflow for the characterization of cellular states and interactions induced by immunotherapies, including CAR-T cells and bispecific antibodies. Our data illustrate how kinetics and mode of action of immune therapies can be quantified at ultra-high precision and cellular resolution. Owing to its high scalability and low costs, the Interact-omics workflow can be readily implemented into large-scale screens to identify or prioritize candidate immunotherapy drugs. Using blinatumomab as a model, we demonstrated how Interact-omics enables systematic identification of cellular mechanisms underlying

therapy response, validating known biomarkers and revealing novel interaction-based parameters. With its scalability, rapid turnaround and low costs, Interact-omics provides an ideal foundation for developing companion diagnostics and advancing personalized immunotherapy approaches.

The extremely high throughput of the Interact-omics framework enables the quantitative dissection of complex interaction networks across entire organ systems and organisms. In this context, we have mapped cellular interaction networks in response to virus infection in mice across distinct time points and immune organs. This approach revealed organ-specific shifts in single-cell landscapes and cellular interaction networks underlying antiviral immune responses and identified fundamental differences in cellular interaction dynamics between primary and secondary lymphoid organs. Our data confirmed previously known and identified new cellular interaction patterns and provides a quantitative framework for a systems-level understanding of how complex cellular interaction networks cooperate across immune organs to jointly orchestrate immune responses. In the future, the Interact-omics framework could be of great utility to decipher fundamental principles of multilayered immune cell crosstalks underlying complex (patho-) physiological processes, such as age-related decline of the immune system or cancer immunity.

Collectively, the Interact-omics approach represents a highly versatile and scalable cytometry-based framework that can be readily implemented for the joint mapping of cellular immune landscapes and their physical interactions with a wide range of applications across a variety of research fields.

### Limitations and guidelines

The presented approach uses flow cytometry to measure both single-cell and PIC landscapes. As physical interactions can be of biological and technical nature, experimental conditions that affect the formation of cellular interactions need to be carefully chosen and controlled. This includes sample preparation and processing, but also cytometric parameters (Supplementary Fig. 1). Particular attention should be paid to maintaining consistent cell concentrations during sample handling and ensuring stable flow rates across experimental groups. Lower cell concentrations and slower flow rates can help minimize technical interactions. Early fixation can be used to stabilize interactions.

Given the impact of experimental conditions, we strongly recommend using case-control studies where samples from all experimental groups are treated uniformly. Reporting relative enrichments of cellular interactions compared to controls is essential for accurate interpretation.

For in vivo experiments, we have demonstrated that cells with a strong affinity to interact can artificially interact if they are brought into proximity during sample preparation, even if they were physically separated in vivo. Therefore, while these interactions may be biologically meaningful, this should be taken into consideration when interpreting the results, and pooling of organs or samples should be avoided to prevent artificial interactions. For novel in vivo settings, users of our framework may consider colabeling strategies (Supplementary Notes 2 and 3) to assess interactions that may form ex vivo.

## Online content

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

¹Heidelberg Institute for Stem Cell Technology and Experimental Medicine (HI-STEM gGmbH), Heidelberg, Germany. ²Division of Stem Cells and Cancer, German Cancer Research Center (DKFZ), Heidelberg, Germany. ³Faculty of Biosciences, Heidelberg University, Heidelberg, Germany. ⁴Berlin Institute of Health (BIH) at Charité Universitätsmedizin Berlin, Berlin, Germany. ⁵Berlin Institute for Medical Systems Biology, Max Delbrück Center for Molecular Medicine in the Helmholtz Association, Berlin, Germany. ⁶Department of Hematology, Oncology and Tumor Immunology, Charité – Universitätsmedizin Berlin, Corporate Member of Freie Universität Berlin and Humboldt-Universität zu Berlin, Berlin, Germany. ⁷Precision Healthcare University Research Institute, Queen Mary University of London, London, UK. ⁸Department of Gastroenterology, Infectious Diseases and Rheumatology, Charité – Universitätsmedizin Berlin, Corporate Member of Freie Universität Berlin and Humboldt-Universität zu Berlin, Berlin, Germany. ⁹Deutsches Rheuma-Forschungszentrum, ein Institut der Leibniz-Gemeinschaft, Berlin, Germany. ¹⁰Division of Chronic Inflammation and Cancer, German Cancer Research Center (DKFZ), Heidelberg, Germany. ¹¹Flow Cytometry Core Facility, European Molecular Biology Laboratory (EMBL), Heidelberg, Germany. ¹²German Cancer Consortium (DKTK), Partner Site Berlin, and German Cancer Research Center (DKFZ), Heidelberg, Germany. ¹³Institute for Automation and Applied Informatics, Karlsruhe Institute of Technology, Eggenstein-Leopoldshafen, Germany. ¹⁴Division of Pediatric Neurology and Metabolic Medicine, Department I, Center for Pediatric and Adolescent Medicine, Medical Faculty Heidelberg, Heidelberg University, Heidelberg, Germany. ¹⁵Computational Oncology Group, Molecular Precision Oncology Program, National Center for Tumor Diseases (NCT) Heidelberg and German Cancer Research Center, Heidelberg, Germany. ¹⁶Institute of Lung Health and Immunity, Comprehensive Pneumology Center Munich, Helmholtz Zentrum München, Munich, Germany, German Center for Lung Research (DZL), Munich, Germany. ¹⁷The M3 Research Center, Institute for Interdisciplinary Research On Cancer Metabolism and Chronic Inflammation, Medical Faculty, Eberhard Karls University Tübingen, Tübingen, Germany. ¹⁸Department of Pediatric Oncology and Hematology, Charité – Universitätsmedizin Berlin, Corporate Member of Freie Universität Berlin and Humboldt-Universität zu Berlin, Berlin, Germany. ¹⁹German Cancer Consortium (DKTK), Partner Site Munich, and German Cancer Research Center (DKFZ), Heidelberg, Germany. ²⁰Division of Clinical Pharmacology, LMU University Hospital, LMU Munich, Munich, Germany. ²¹Einheit für Klinische Pharmakologie (EKLiP), Helmholtz Zentrum München, German Research Center for Environmental Health (HMGU), Neuherberg, Germany. ²²Innere Medizin C, Universitätsmedizin Greifswald, Greifswald, Germany. ²³Max Delbrück Center for Molecular Medicine in the Helmholtz Association, Berlin, Germany. ²⁴German Cancer Consortium (DKTK), Heidelberg, Germany. ²⁵Innovation and Service Unit for Bioinformatics and Precision Medicine (BPM), German Cancer Research Center, Heidelberg, Germany. ²⁶Division of Translational Precision Medicine, Institute of Human Genetics, Heidelberg University, Heidelberg, Germany. ²⁷These authors contributed equally: Dominik Vonficht, Lea Jopp-Saile, Schayan Yousefian, Viktoria Flore. ²⁸These authors jointly supervised this work: Daniel Hübschmann, Simon Haas. ✉e-mail: d.huebschmann@dkfz-heidelberg.de; simon.haas@bih-charite.de

## Methods

### Animals

Unless otherwise stated, animal experiments were conducted under German law and approved by Regierungspräsidium Karlsruhe (approval numbers DKFZ299, G-55/20, G-56/20) or the Landesamt für Gesundheit und Soziales in Berlin (LAGeSo, G0016/20). Mice were maintained in individually ventilated cages under specific pathogen free conditions in the animal facility of the DKFZ (Heidelberg, Germany) or at the Charité animal facility (FEM, Berlin, Germany) with ad libitum access to water and food (22 ± 2 °C, 45–65% humidity, 12 h light–dark cycle). Mice used in LCMV experiments were 7 weeks old; all other mice were between 6 and 20 weeks old. CD45.1 mice were obtained from in house breeding at DKFZ (Z110I02, B6.SJL-Ptprca Pepcb/BoyJ) or from Charles Rivers (B6.SJL-PtprcaPepcb/BoyCrl). For experiments with antigen-specific T cells, cells were isolated from B6.Cg-Tg(TcraTcrb)425Cbn/J (OT-II) or LCMV-TCRtg P14 (ref. [58]) and SMARTA[59] mice expressing the congenic markers CD45.1 or CD90.1. All mice were female.

### Human samples

All analyses were conducted according to the Declaration of Helsinki and in accordance with local ethical guidelines; written informed consent of patients was obtained. Usage of samples from patients treated with blinatumomab in this study was approved by the ethics committee of Charité Universitätsmedizin Berlin (reference number EA2/147/23). PBMC samples from healthy blood donors were obtained as buffy coats from the blood donation center IKTZ Heidelberg or ZTB Berlin. Mononuclear cells were isolated by Ficoll (GE Healthcare) density gradient centrifugation and stored in fetal calf serum (FCS) 10% DMSO in liquid nitrogen until usage. For the blinatumomab response analysis (below), BM samples from 42 patients with a B-ALL relapse were assessed. Samples were directly collected before the start of the blinatumomab course and processed as part of routine diagnostics by Ficoll density gradient centrifugation and minimal residual disease quantification. Remaining cells were stored in FCS 10% DMSO in liquid nitrogen for research purposes. Good response to blinatumomab ($n = 18$) is defined as minimal residual disease negativity directly after a blinatumomab course (28 days) and all subsequent time points. Nonresponse to blinatumomab ($n = 4$) is defined as leukemic cell persistence (based on morphological or minimal residual disease evaluation) without any reduction after a blinatumomab course. The remaining 20 patient samples could not be unequivocally assigned to these response states (good response versus nonresponse) or had no residual disease at the start of the blinatumomab course. The median age of patients in the study was 9.5 years. Data on sex were collected from patients of the B-ALL cohort. However, given that the patient cohort analyzed here was not part of a clinical trial, sex-specific considerations were not explicitly integrated into the study design. The distribution of male to female participants was, however, balanced (57% male, 43% female).

### Isolation of murine immune cells

For isolation of antigen-specific T cells, the spleen and various LNs (including inguinal, axial, submandibular and mesenteric) were carefully extracted. Tissues were homogenized using a 40-μm filter (Falcon) and a syringe plunger in cold Roswell Park Memorial Institute (RPMI) medium (Sigma Aldrich) with 2% FCS (Gibco by Life Technologies). Subsequently, single-cell suspensions from spleens were treated with erythrocyte lysis solution (ACK buffer, containing 0.15 M $NH_4Cl$, 1 mM $KHCO_3$ and 0.1 mM $Na_2EDTA$ in water from Lonza) for a duration of 5 min. For some readouts, these suspensions were combined with the LN samples or maintained separately. CD4 and CD8 T cells were purified using either the Dynabeads Untouched Mouse CD4 Cells Kit (Invitrogen) or the murine CD4 T cell isolation kit and the murine CD8 T cell isolation kit (Miltenyi) according to the manufacturer's instructions. Purified fractions were stained for further purification using fluorescence-activated cell sorting (FACS) (below). For in vivo

experiments, femurs, spleen and various LNs were dissected and kept separate on ice. LNs and spleens were individually processed as described above. Femurs were flushed using FACS buffer and homogenized using a 40-μm filter (Falcon) and a syringe plunger.

### Ex vivo murine cocultures

Cultures containing OT-II CD4 T cells were incubated at 37 °C with 5% $CO_2$ in U-bottom plates in 200 μl of Dulbecco's Modified Eagle's Medium GlutaMAX (DMEM GlutaMAX, Gibco), supplemented with 10% heat-inactivated FCS (Gibco), sodium pyruvate (1.5 mM, Gibco), L-glutamine (2 mM, Gibco), L-arginine (1×, Sigma), L-asparagine (1×, Sigma), penicillin–streptomycin (100 U ml⁻¹, Sigma), folic acid (14 μM, Sigma), minimum essential medium, nonessential amino acids (1×, ThermoFisher), MEM vitamin solution (1×, ThermoFisher) and β-mercaptoethanol (57.2 μM, Sigma). Next $5 × 10^4$ OT-II cells were incubated with $1 × 10^5$ splenocytes containing various antigen-presenting cell populations in presence or absence of OVA peptide (323–339, InvivoGen).

For murine CAR-T in vitro assays, GFP-expressing CD19 specific CAR-T cells were generated as previously described[60], thawed and washed with PBS. Next, cells were transferred to 10% FCS RPMI 1640 containing 0.05 μg ml⁻¹ IL-15 (Peprotech) and 0.1% β-mercaptoethanol. To recover from freezing procedures, cells were incubated under the same conditions as described above before the coculture assay. Frozen murine splenocytes were thawed and incubated together with CAR-T cells at a ratio of 1/2 target/effector ratio (CAR-T cells/splenocytes) for 0.5 to 3 h. Subsequently, cells were collected, washed with FACS buffer, stained with surface markers and analyzed.

### Ex vivo human cocultures

Cryopreserved PBMCs were thawed in a water bath at 37 °C, transferred to 10% FCS RPMI 1640 and washed twice. After each washing step, cells were centrifuged at 350g for 5 min. Next, $2 × 10^5$ cells were plated in 10% FCS RPMI 1640 and cultured short term for up to 5 h in 200 μl of RPMI 10% FCS. CytoStim (Miltenyi) was used in concentrations recommended by the manufacturer at 37 °C for 2 h before collection.

For experiments using a blinatumomab analog (InvivoGen), a concentration of 50 ng ml⁻¹ was used. The incubation period ranged from 0.25 to 5 h at 37 °C and 5% $CO_2$ in 96-well U-bottom plates. For experiments assessing the stability of blinatumomab-induced interactions on cryopreservation, cells were either incubated for 2 h in presence of the compound and stained with surface antibodies and fixed with 4% paraformaldehyde (PFA) (ThermoFisher) or frozen in Bambanker freezing medium (Nippon Genetics), thawed after 18 h and treated in the same way as the nonfrozen cells.

For in vitro benchmarking experiments, human PBMCs were treated with CytoStim as described above; control groups were left untreated. Then, cells were split into two groups each and stained with CD45-APC-Fire810 or CD45-PE-Fire640, respectively. After mixing the labeled groups, cells were incubated for 0–4 h at 4 °C (200,000 cells per well in 50 μl during staining and acquisition) or processed at seeding densities of 25,000 to 250,000 cells per well in 96-well plates (in 50 μl during staining and acquisition). Subsequently, cells were gathered, washed with FACS buffer, stained with surface markers, fixed with 2% PFA (except the nonfixed control) and analyzed.

For measuring phosphorylated CD247, human PBMCs were seeded at 100,000 cells per well in 200 μl and treated for 1 h with blinatumomab (160 ng ml⁻¹) or CytoStim as described above. Following the stimulation period, cells were fixed immediately by adding CytoFix buffer (15 min, 4 °C). Cells were washed and resuspended in 200 μl of 2.5× Perm/Wash buffer, incubated for 30 min at 37 °C and stained overnight at 4 °C before analysis.

### In vivo mouse experiments

Five days before infection, $1 × 10^4$ LCMV-specific T cells (SMARTA; expressing congenic marker CD90.1) and CD8⁺ T cells (P14; expressing

congenic marker CD45.1) were administered intravenously into C57BL/6J in 300 μl of balanced salt solution, resulting in an approximate seeding of $1 \times 10^3$ cells per mouse[61]. The viral infection was induced intraperitoneally using 200 PFU of the LCMV Armstrong strain[62]. Mice were euthanized on day 3 and/or day 7 post-infection, and various tissues including the spleen, mesenteric LNs and bones were dissected and processed for spectral flow cytometry analysis.

For the in vivo benchmarking experiment, LCMV-specific CD4 T cells were transferred into C57BL6 (CD45.2) hosts 5 days before infection as described above. CD45.1 (B6.SJL-PtprcaPepcb/BoyCrl) and CD45.2 hosts were infected intraperitoneally as described above, and spleens were harvested on day 7 post-infection. Spleens were split into four equal pieces and mixed across CD45.1/CD45.2 hosts for joint tissue homogenization. Mixed samples were processed for spectral flow cytometry analysis.

## Flow cytometry, cell sorting and image-enabled flow cytometry

Unless otherwise stated, cell suspensions were resuspended in 2% FSC PBS (FACS buffer, 0.5 mM EDTA optionally) for performing flow cytometric stainings (Supplementary Tables 2–13). For ex vivo readouts with bispecific engagers and antigen-specific T cells, cells were gathered, centrifuged 5 min at 350g and stained with surface marker panel master mixes using FACS buffer and addition of Brilliant Stain buffer (BD) according to the manufacturer's recommendation. Cells were stained for 30 min on ice in 96-well V-bottom plates, followed by washing with FACS buffer, centrifugation for 5 min at 350g and resuspension in 200 μl of FACS buffer. For more time-consuming in vivo experiments, cells were labeled with fixable dead cell exclusion dyes followed by fixation of obtained single-cell suspensions with cold 2% PFA PBS for 15 min at room temperature. Cells were washed, centrifuged for 5 min at 350g and then stained overnight at 4 °C. After washing and centrifugation for 5 min at 350g, cells were filtered through a 35-μm cell strainer and kept on ice until flow cytometric analysis. For flow cytometric analysis, a Cytek Aurora (Cytek Biosciences) or LSR Fortessa (BD) equipped with five lasers was used. For sorting of naive T cells in ex vivo setups, FACSAria Fusion or FACSAria II sorters equipped with 70-μm nozzles were used. For imaging cytometry, image-enabled cell sorting using the BD CellViewTM Imaging Technology[29] or an ImageStream (Cytek) was used. For image-enabled cell sorting, PBMCs were incubated for 2 h with CytoStim, stained with surface markers followed by fixation with 2% PFA PBS as described above and operated using a 100-μm sort nozzle, with the piezoelectric transducer driven at 34 kHz and automated stream setup by BD FACSChorus Software, and a system pressure of 20 psi. For the ImageStream experiment, data were acquired using the Cytek INSPIRE software.

## Image-enabled flow cytometry analysis

For image-enabled flow cytometric analysis, radiofrequency images underwent processing as previously described[29]. The raw image TIFF files were imported into ImageJ and processed with the BD CellView plugin. The corresponding FCS files were loaded into FlowJo (BD), and cells were gated as living CD45+ cells. Using the flowCore[63], CytoML[64] and flowWorkspace[65] packages, the generated FlowJo workspace was loaded into R (≥v4.3.0) for further processing. Subsequently, images were converted to JPG format, and channels containing the light-loss, FSC and side scatter parameters were kept for downstream analysis. Four replicates, each comprising 1,000 images, were manually categorized as singlet, doublet, triplet or higher-plex multiplets, and the categories were used to train a decision tree-based classification model using Rpart[66] and caret[67]. As features for the model, the image-based features, the conventional flow cytometry parameters and the FSC ratio, defined by the quotient of FSC-A and FSC-H, were used. Feature importance in the model was determined to identify relative contributions of each variable in making accurate predictions.

Otsu[30] thresholding, which minimizes intragroup variance, was applied to a histogram of the FSC ratio divided into at least 1,000 bins, effectively separating the data into two categories based on whether their FSC ratio is above or below the threshold.

Louvain clustering was performed for $n = 100$ iterations (resolution 1) on all or a subset of the following features: image-based parameters, conventional flow parameters and the FSC ratio. Consensus clustering solutions were calculated using soft least squares Euclidean consensus partitions as implemented in the clue[68] package. Data were visualized in uniform manifold approximation and projection (UMAP)[55] embeddings using the same input features as used for clustering. UMAP embeddings were computed across 15 nearest neighbors and a minimum Euclidean distance of 0.1, and populations were annotated based on cell-type-specific markers and their combinations. Furthermore, Louvain clustering was performed for $n = 100$ iterations with different resolution parameters and variation in cluster labels between important features including image-based parameters and conventional flow parameters was assessed using the adjusted rand index.

For ImageStream-based analyses, ImageStream fluorescence intensity values (based on the sum of the pixel intensities in the mask as selected by ImageStream, background subtracted) were compensated and transformed using FlowJo (v.10.10) and IDEAS (v.6.2). Data were processed using PICtR (below). Interacting populations were solely annotated based on mutually exclusive marker expression, since forward scatter properties are not acquired by ImageStream. For conventional gating, gates were selected in FlowJo.

For cell segmentation from brightfield images, the cyto2 model from the Python package CellPose[69] was used with a cell pixel diameter of 20. To remove cellular debris, events that met any of the following criteria were excluded: major axis length <15 pixels or >40 pixels, circularity <0.7, area <100 pixels or >1,000 pixels. Area and major axis length were computed using the Python package scikit-image[70]. Circularity was calculated using the formula $\frac{4 \times \pi \times area}{perimeter^2}$, with perimeter values also obtained from scikit-image. This filtering process excluded approximately 4% of the detected objects.

## Identification and analysis of PICs with PICtR

**Benchmarking.** Benchmarking was performed on the imaged-enabled flow cytometry data with $n = 3,865$ manually classified events across $n = 4$ replicates. Several thresholding methods based on the FSC ratio were used to define a cutoff of events with a high or low FSC ratio (Supplementary Table 14). Otsu[30], IsoData[71], Intermodes[72], RenyiEntropy[73], Li[74], Shanbhag[75], Huang[76] and Mean[77] algorithms were used as implemented in the R package autothresholdr[78], and the Triangle[79] algorithm was ported from the ImageJ implementation in Java. $k$-means clustering was used with $k = 2$ for thresholding and Gaussian mixture models were computed as implemented in the R package mclust[80]. Performance of the methods was evaluated based on the annotation of the image-enabled flow cytometry data and reported as F1 scores, where 1 indicates a perfectly accurate reproduction of the manual ground-truth classification.

Next, different clustering algorithms (Supplementary Table 15) were evaluated regarding their ability to discriminate single and interacting cells considering conventional flow cytometry features (forward scatter, side scatter, cell type markers CD45, CD3, CD19, HLA-DR and CD33, and the FSC ratio). Candidates were selected based on their popularity in the single-cell and flow cytometry fields or based on their performance on high-dimensional single-cell flow and mass cytometry data as evaluated by Weber and Robinson[81]. Louvain[31] and Leiden[82] clustering (implemented through igraph) were used on a shared nearest neighborhood graph with $k = 5$ nearest neighbors, HDBSCAN[83,84] (hierarchical density-based spatial clustering of applications with noise) was used on a UMAP embedding with $k = 15$ nearest neighbors and Phenograph[85] was used with Louvain or Leiden clustering. FlowSOM[86] (Spectre implementation), FlowMeans[87], Rclusterpp[88,89]

and Immunoclust[90] were used directly on the features. Each method was run for $n = 100$ iterations and the performance was reported as F1 scores based on the ground-truth classification.

**Flow cytometric data preprocessing.** For full-spectrum flow cytometry data, raw FCS files were spectrally unmixed using the inbuilt unmixing function of the SpectroFlo (Cytek Biosciences) software. FCS files were imported into FlowJo (BD) to assess unmixing by visualizing $N \times N$ plots. Axes were adjusted wherever needed and parameters for logicle[91] or generalized bi-exponential transformation of data were defined for every surface marker individually. PeacoQC[92] was used as an automatic quality control mechanism for cytometry data where needed. The populations of interest were exported using channel values defined by the inbuilt export function of FlowJo. Raw and processed cytometry data for key experiments are provided at https://doi.org/10.5281/zenodo.10637096 (ref. [93]).

**PICs toolkit in R (PICtR).** Usage and processing of reduced exemplary data are provided in a vignette.

The workflow starts by importing compensated and transformed cytometry data (CSV files) into R ($\geq$v.4.3.0). BPCells[94] is used for bit-packing compression on a high-performance computing cluster to manage extensive data. For each measured event, the FSC ratio, defined by the ratio of FSC-A and FSC-H, is calculated and scaled to transform the data into a similar range as recorded marker expression values. For the downstream analysis, the measured marker expression values, forward scatter, side scatter and the determined FSC ratio parameter are used as features. Next, the data are sampled using an atomic sketching approach as implemented in Seurat v.5 (ref. [32]). This approach is particularly effective in preserving rare events, including cellular interactions.

Sampled data are further processed with the Seurat workflow. For the datasets in this paper, $n - 1$ principal components were chosen for dimensionality reduction, however, the number of components can be adjusted. The resulting principal component space is used to construct a shared nearest-neighbor graph across the 20 nearest neighbors and to determine the UMAP embeddings using 30 neighbors and a minimum cosine distance of 0.3 for the manifold approximation. Furthermore, the shared nearest-neighbor graph is used as input for Louvain clustering. Other clustering methods are provided as alternatives. For cells not included in the initial sketching process, cluster labels are determined using linear discriminant analysis as implemented in the R package MASS[95].

Clusters that contain interacting cells are selected based on the FSC ratio distribution. By default, a discriminating threshold is obtained using the Otsu method, but alternative thresholding methods are also available. Next, the fraction of cells above and below the FSC ratio threshold is determined per cluster. Finally, based on the predicted cluster labels from linear discriminant analysis, interacting cells within the entire dataset are identified. Interacting cells are subjected to PCA, shared nearest-neighbor graph construction, clustering and UMAP analysis to obtain a refined characterization.

**Annotation of PICs.** Clusters of single cells are annotated based on known cell identity markers and expert knowledge. Similarly, clusters identified as interacting cell populations are annotated based on the combination of mutually exclusive surface markers (for example, evaluated through marker enrichment modeling[56]). For example, coexpression of the B cell marker CD19 and the T cell markers CD3 and CD4 within an interacting cluster indicates an interaction of a B and CD4$^+$ T cell.

Of note, interacting cell clusters that express markers from only one cell identity might represent homotypic cellular interactions (for example, interactions between two B cells). Since alternative explanations, such as preceding cytokinesis, cannot be ruled out, clusters

such as these should be excluded from downstream analysis to avoid low-confidence annotations.

**Adjustment of counts of PICs.** Frequencies of interactions are reported as the frequency among all live, high-quality events or the frequency among all interacting cells. Alternatively, interaction frequencies are normalized by taking the frequency of the respective interaction partners into account: $f_A$ denotes the fraction or rate of cell type A, and analogously, $f_B$ denotes the fraction or rate of cell type B. Furthermore, let $f_{AB}$ denote the fraction or rate of interacting cells of types A and B. To assess the number of such interacting cells, we introduce the enrichment term $e_{AB} = \frac{O_{AB}}{E_{AB}}$ where $O_{AB} = f_{AB}$ denotes the observed and $E_{AB}$ denotes the expected rate of interacting cells. The expected rate is given by the harmonic mean $H(f_A, f_B)$ of the two singlet rates:

$H(x,y) = \frac{2}{\frac{1}{x} + \frac{1}{y}} = \frac{2xy}{x+y}$. We thus get for the enrichment $e_{AB}$:

$$e_{AB} = \frac{f_{AB}}{\left(\frac{2f_A f_B}{f_A + f_B}\right)} \tag{1}$$

Of note, the harmonic mean of a list of numbers tends strongly toward the least element of the list. In our case with two entries, in case $f_A \gg f_B$, we get:

$$E_{AB} = H(f_A, f_B) = \frac{2f_A f_B}{f_A + f_B} \approx \frac{2f_A f_B}{f_A} = 2f_B \tag{2}$$

The frequency of expected interactions between two cell types with strongly different abundance is thus given by the less abundant cell type. Still, even for the more abundant cell type A, $E_{AB}$ increases with increasing $f_A$:

$$\frac{\partial E_{AB}}{\partial f_A} = \frac{2f_B(f_A + f_B) - 2f_A f_B}{(f_A + f_B)^2} = \frac{2f_A f_B}{(f_A + f_B)^2} > 0 \,\forall f_A > 0 \,\forall f_B > 0 \tag{3}$$

### Blinatumomab response analysis

BM aspirates obtained from 42 relapsed B-ALL patients were thawed in a water bath at 37 °C, transferred to 10% FCS RPMI 1640 and washed twice. After thawing, each sample was split into two. One half of the sample was cultured in 200 μl of RPMI 1640 (10% FCS) supplemented with 50 ng ml$^{-1}$ blinatumomab analog (InvivoGen) for 1 h at 37 °C and 5% $CO_2$ in a 96-well U-bottom plate. The other half of the sample was cultured in RPMI 1640 (10% FCS) without blinatumomab supplementation for 1 h at the same conditions. After the incubation, cells were collected, washed with FACS buffer, stained with the surface marker panel and analyzed. Raw FCS files and CSV files were processed as described above. To analyze whether certain features are associated with the two response groups (good responders and nonresponders) the mean value for each feature in the dataset (singlet frequency, interacting cell frequencies and fold changes of interacting cells after blinatumomab treatment), was calculated for both groups. Subsequently, the fold change for each feature was computed as the ratio of the mean value in the good responder group to that in the nonresponder group. Furthermore, a two-sided $t$-test was performed for each feature to test for significance between groups. Before performing the correlation analysis, a feature selection was conducted to refine the dataset for more targeted analysis. This selection was based on the results of a univariate analysis, by which features were selected based on an abs($t$-value) threshold greater than 1.5. For these selected features, a correlation matrix was computed using the function cor() from the stats package. Afterward, the distance matrices were created, and a hierarchical clustering was performed on the rows and columns of the correlation matrix separately.

## Statistics and reproducibility

Numerical data were processed with R (≥v.4.3.0) or Python v.3.12.5; see Supplementary Table 16 for details. Two sample groups were compared by parametric tests (two-tailed Welch $t$-tests), or nonparametric tests (two-sided Wilcoxon rank sum tests or estimated marginal means[57]) depending on the distribution of the underlying data points as evaluated by Shapiro–Wilk tests. Analysis of variance was used for multiple groups after evaluation of the distribution of the underlying data points by Shapiro–Wilk tests. Details about adjustments for multiple comparisons can be found in the respective figure legends. No statistical method was used to predetermine sample size.

Where applicable, PeacoQC or FlowAI[96] were used to exclude low-quality flow cytometry events and cells were gated according to the provided gating strategies. Furthermore, cells were removed when high autofluorescence or signal anomalies suggested a low-quality event. For the B-ALL cohort ($n$ = 42), we compared patients who could unequivocally be categorized into good responders ($n$ = 18) and non-responders ($n$ = 4) to explore mechanisms underlying therapy response among patients with residual disease. The remaining 20 patient samples were therefore excluded from the downstream analysis. Data points were excluded from the downstream analysis if a population of cells was not detectable across all conditions and the excluded populations are noted in the respective figure legends. Clusters of PICs without a cell type exclusive marker combination might represent homotypic interactions and were excluded from the downstream analysis.

The investigators were not blinded to allocation during experiments and outcome assessment. Mice, murine samples and PBMC samples from healthy blood donors were randomly allocated to groups. The B-ALL experiments were not randomized since all patient samples were measured in the presence and absence of blinatumomab.

## Reporting summary

Further information on research design is available in the Nature Portfolio Reporting Summary linked to this article.

## Data availability

Raw and processed cytometry data for key experiments are available on Zenodo at https://doi.org/10.5281/zenodo.10637096 (ref. 93). Source data are provided with this paper.

## Code availability

PICtR is available as an open-source R package available on GitHub at github.com/agSHaas/PICtR. Code to reproduce key analysis results is available on GitHub at github.com/agSHaas/ultra-high-scale-cytometry-based-cellular-interaction-mapping/.

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

## Acknowledgements

We acknowledge the technical support of the BIH, DKFZ and EMBL Flow Cytometry Core Facilities. We thank D. Boesel for technical support in animal experiments. This project is cofunded by the European Union (ERC-StG, InteractOmics, grant no. 101078713 and ERC-PoC, Interact-ALL, grant no. 101185344 to S.H.). Views and opinions expressed are, however, those of the author(s) only and do not necessarily reflect those of the European Union or the European Research Council. Neither the European Union nor the granting authority can be held responsible for them. S.H. received extra support by the Heisenberg program of the German Research Foundation (DFG), the e:Med LeukoSyStem consortium (BMBF), the HEROES-AYA consortium (BMBF), the TEP-CC consortium (Bruno and Helene Jöster Foundation), the DFG CRCs 1588 and 1444, and the DFG project no. HA 8790/3-1. L.J.-S. and D.V. were supported by the Joachim Herz Stiftung. D.H. received funding from the Molecular Precision Oncology Program (MPOP) of the National Center for Tumor Diseases (NCT) Heidelberg and the HEROES-AYA consortium (BMBF). L.K.S. is a participant in the BIH–Charité Clinician Scientist Program funded by the Charité – Universitätsmedizin Berlin and the Berlin Institute of Health. A.N.H. is supported by a Lichtenberg fellowship and 'Corona Crisis and Beyond' grant by the Volkswagen Foundation, a BIH Clinician Scientist grant and German Research Foundation grant no. DFG-TRR241-A05 (project-ID no. 375876048) and grant no. INST 335/597-1, as well as with the ERC-StG 'iMOTIONS' grant (no. 101078069). S.K. received extra support from the Bavarian Cancer Research Center (BZKF) (TANGO), the Deutsche Forschungsgemeinschaft (DFG, grant nos KO5055-2-1 and KO5055/3-1), the international doctoral program 'i-Target: immunotargeting of cancer' (funded by the Elite Network of Bavaria), Marie Sklodowska-Curie Training Network for Optimizing Adoptive T Cell Therapy of Cancer (funded by the Horizon 2020 programme of the European Union; grant no. 955575), Else Kröner-Fresenius-Stiftung (IOLIN), German Cancer Aid (AvantCAR. de.), the Wilhelm-Sander-Stiftung, the Go-Bio-Initiative, the m4-Award of the Bavarian Ministry for Economical Affairs, Bundesministerium für Bildung und Forschung (CONTRACT), European Research Council (Starting grant no. 756017, PoC grant no. 101100460 and CoG grant no. 101124203), by the SFB-TRR grant no. 338/1 2021–452881907, Fritz-Bender Foundation, Deutsche José Carreras Leukämie Stiftung, Hector Foundation, Bavarian Research Foundation (BAYCELLATOR), the Bruno and Helene Jöster Foundation (360° CAR). C.E. and A.v.s. received support from the German Childhood Cancer Foundation (grant no. DKS-2022.11). C.E. received funding from the German Cancer Consortium (DKTK Call 2021-2023) and the German Childhood Cancer Foundation (grant no. DKS-2022.11). A.T. received funding from the ERC Advanced Grant SHATTER-AML (grant no. AdG-101055270) and the HI-STEM gGmbH was supported by the Dietmar Hopp Foundation. The work of L.D., A.J.Y.S and R.M. was supported by the Helmholtz Association under the Program 'Natural, Artificial and Cognitive Information Processing (NACIP)' and the 'HIDSS4Health' Helmholtz Information and Data Science School for Health.

## Author contributions

S.H. and D.H. conceived of and supervised the study. D.V., S.Y., R.T., B.A., I.S.V., V. Flore and V. Fregona performed in vitro coculture experiments with support from C.R. D.V., D.O.-R. and V. Fregona performed the imaged-enabled flow cytometry experiments. Mouse experiments were performed by D.V. and A.N.H. and supported by M.F.-V., I.S.V., Y.L., M.H., F.G. and C.R. L.J.-S. and D.H. developed the PICtR analysis workflow with input from V. Flore. Benchmarking was performed by V. Flore and Z.G. The PICtR R package was written by L.J.-S. and V. Flore. Data analyses were performed by L.J.-S., V. Flore, D.V. and S.Y. L.K.S., S.K., J.K. and U.K. contributed to murine CAR-T cell experiments. Image segmentation of imaging cytometry data was performed by L.D. and A.J.Y.S. under supervision of R.M. C.E., A.v.s. and A.E. contributed to sample acquisition and clinical expertise for B-ALL experiments. M.C.F. mediated access and interpretation of the intestinal dataset. S.H., D.H. and A.T. provided funding and resources. S.H., D.H., L.J.-S., D.V., S.Y. and V. Flore drafted the paper. All authors read and approved the paper.

## Funding

## Competing interests

The authors declare the following competing interests: S.H., S.Y., V. Flore, I.S.V., C.E., R.T., D.H., D.V., L.J.-S. and A.T. are listed as inventors for the European Patent Application no. 24181153.8 from the Charité – Universitätsmedizin Berlin and Deutsches Krebsforschungszentrum Stiftung des öffentlichen Rechts (priority application filed). The patent covers the core technology and applications of the Interact-omics framework. S.K. has received honoraria from Cymab, Plectonic, TCR2 Inc., Novartis, BMS, Miltenyi and GSK. S.K. is inventor of several patents in the field of immuno-oncology. S.K. received license fees from TCR2 Inc. and Carina Biotech. D.H. owns stock from Platomics GmbH. The other authors declare no competing interests.

## Additional information

**Extended data** is available for this paper at https://doi.org/10.1038/s41592-025-02744-w.

**Correspondence and requests for materials** should be addressed to Daniel Hübschmann or Simon Haas.

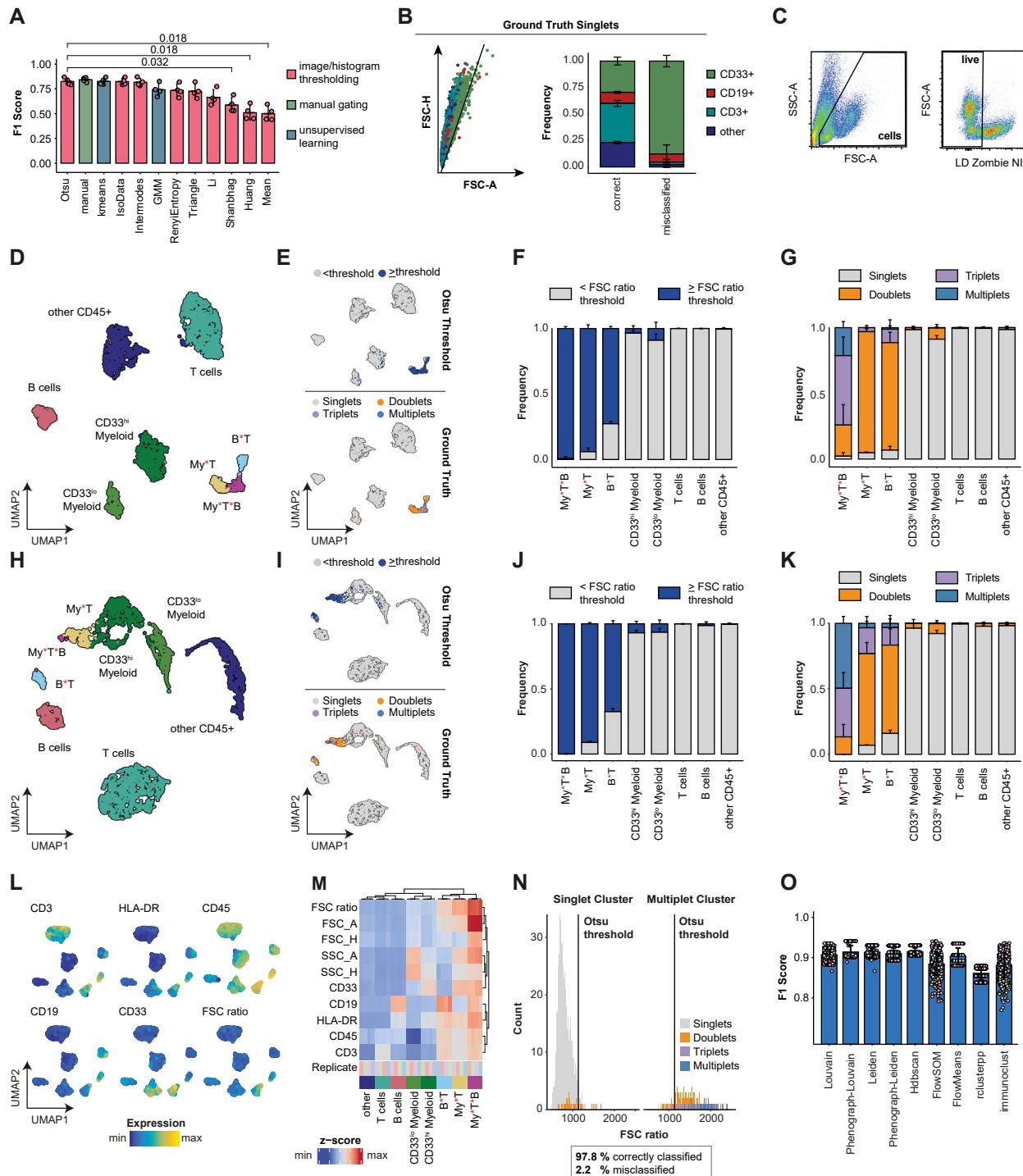

**Extended Data Fig. 1 | See next page for caption.**

**Extended Data Fig. 1 | Technical aspects of cytometry-based cellular interaction mapping. a**. Performance of different classification methods based on the FSC ratio as measured by the F1 score. Manual image annotation served as the ground truth; see Methods for details. $n = 4$ technical replicates are shown in the scatter plot; bars indicate the mean F1 score. Error bars indicate the standard deviation. **b**. Dot plot displaying the forward scatter area (FSC-A) and forward scatter height (FSC-H) properties of ground truth singlets; the Otsu threshold of the FSC ratio is shown as a diagonal line. The bar plots show all ground truth singlets split into correctly classified and misclassified events according to the FSC ratio threshold and are colored by marker expression. $n = 4$ technical replicates; error bars indicate the standard deviation. **c**. Gating strategy to select Lymphocytes and live cells using scatter properties and a live-dead (LD) marker. This gating strategy was employed throughout the manuscript. **d-g**. Louvain clustering performed on the top important features from the feature importance analysis (see Fig. 1c) and cell type markers. **d**. Annotated UMAP representation. **e**. UMAP embedding from panel **d** with cells exceeding the Otsu threshold of the FSC ratio highlighted in blue (top) or cells colored by their ground truth annotation (bottom). **f**. Relative frequency of singlets and interacting cells in each population classified according to the FSC-ratio. Error bars indicate the standard deviation. **g**. Relative frequency of cells in each population based on the ground truth annotation. Error bars indicate the standard deviation.

**h-k**. Louvain clustering performed on cell type markers only. **h**. Annotated UMAP representation. **i**. UMAP embedding from panel **h** with cells exceeding the Otsu threshold of the FSC ratio highlighted in blue (top) or cells colored by their ground truth annotation (bottom). **j**. Relative frequency of singlets and interacting cells in each population classified according to the FSC-ratio. Error bars indicate the standard deviation. **k**. Relative frequency of cells in each population based on the ground truth annotation. Error bars indicate the standard deviation. **l**. Feature plots showcasing cell type marker expression in the UMAP embedding from Fig. 1f. **m**. Heatmap depicting normalized mean feature expression (rows) within merged clusters derived from Louvain clustering (columns, on conventional flow parameters) across replicates. Populations are the same as in Fig. 1f. **n**. Histogram colored by the ground truth annotation and split by the identified singlet and multiplet clusters in Fig. 1f. The Otsu threshold is shown. **o**. Performance of different clustering methods evaluated regarding their ability to resolve singlet and interacting populations. All algorithms were used for $n = 100$ iterations on conventional flow parameters including forward scatter parameters, side scatter parameters, cell type markers and the FSC ratio, see Methods for details. $n = 4$ technical replicates are shown in the point plot; bars indicate the mean F1 score. Error bars indicate the standard deviation. Abbreviations: UMAP: uniform manifold approximation and projection, CD33: myeloid marker, CD19: B cell marker, CD3: T cell marker.

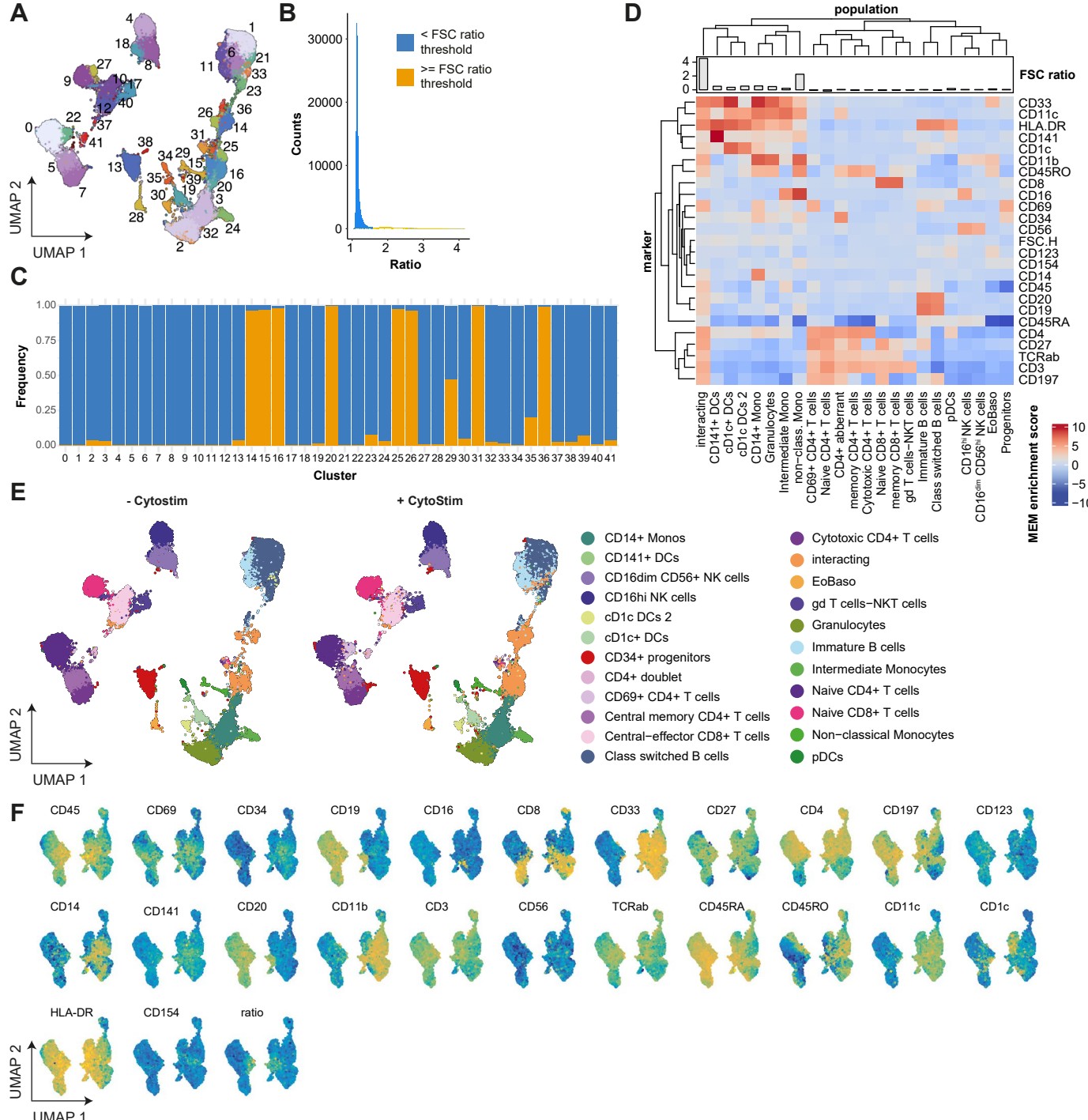

**Extended Data Fig. 2 | Interact-omics resolves complex cellular interactions induced by CytoStim. a**. UMAP embedding of Fig. 2a, depicting human PBMCs treated with or without CytoStim. Merged UMAP of *n* = 4 replicates from a single donor per condition. **b**. Histogram of the FSC ratio. Cells exceeding Otsu's threshold are highlighted in orange. **c**. Stacked bar plot displaying the fraction of PICs (orange) in each cluster obtained from Louvain clustering. **d**. Heatmap showing scores of marker enrichment modeling (MEM) on clusters shown in Fig. 2a. Interacting cells have an enrichment of the FSC ratio (bar plots on top) and co-express various cell type specific markers, like CD3, CD33 and CD19, and HLA-DR. **e**. Split UMAP display based on CytoStim treatment. Interacting cells show specific enrichment upon co-incubation with CytoStim. **f**. Feature plots, showing UMAP embeddings from Fig. 2b color-coded by expression levels of selected markers.

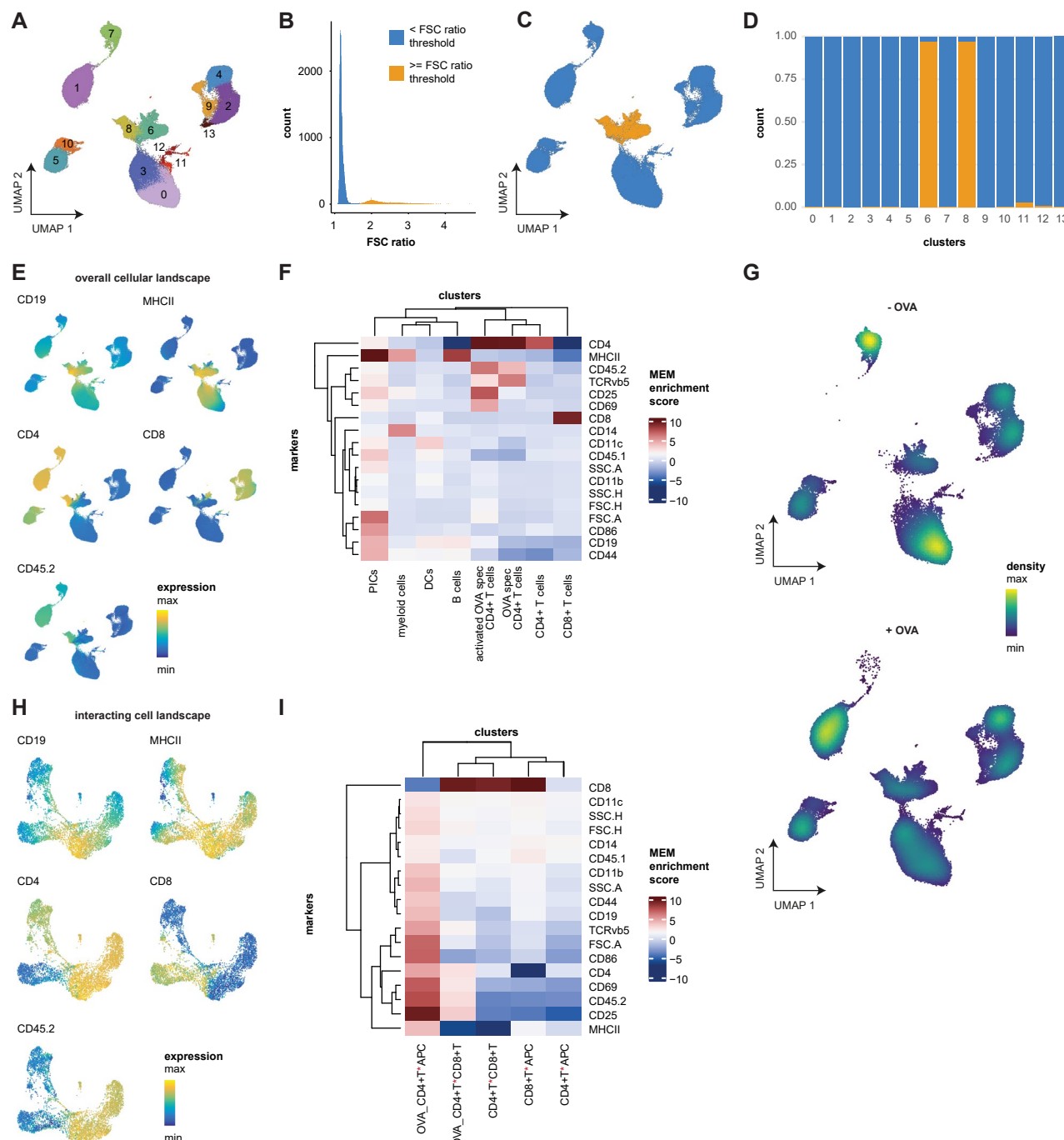

**Extended Data Fig. 3 | Interact-omics resolves antigen-specific interactions in the ovalbumin (OVA)-OT-II model. a**. UMAP of the overall cellular landscape (numeric cluster labels instead of the annotation in Fig. 2g); the numeric cluster labels correspond to panel **d**. **b**. Histogram of the FSC ratio. Cells exceeding Otsu's threshold are highlighted in orange. **c**. UMAP of the overall cellular landscape with cells with an FSC ratio above Otsu's threshold highlighted in orange. **d**. Proportions for singlets and interacting cells in each cluster. Clusters 6 and 8 were selected as interacting cell clusters (85th percentile). **e**. Selected feature plots for the overall cellular landscape. **f**. Marker Enrichment Modelling (MEM) heatmap for the overall cellular landscape. **g**. Singlet point density UMAPs in the presence or absence of ovalbumin. **h**. UMAP of the interacting landscape from Fig. 2h with selected features highlighted. **i**. Marker Enrichment Modelling (MEM) heatmap for the interacting cell landscape. Abbreviations: UMAP: uniform manifold approximation and projection, OVA: ovalbumin.

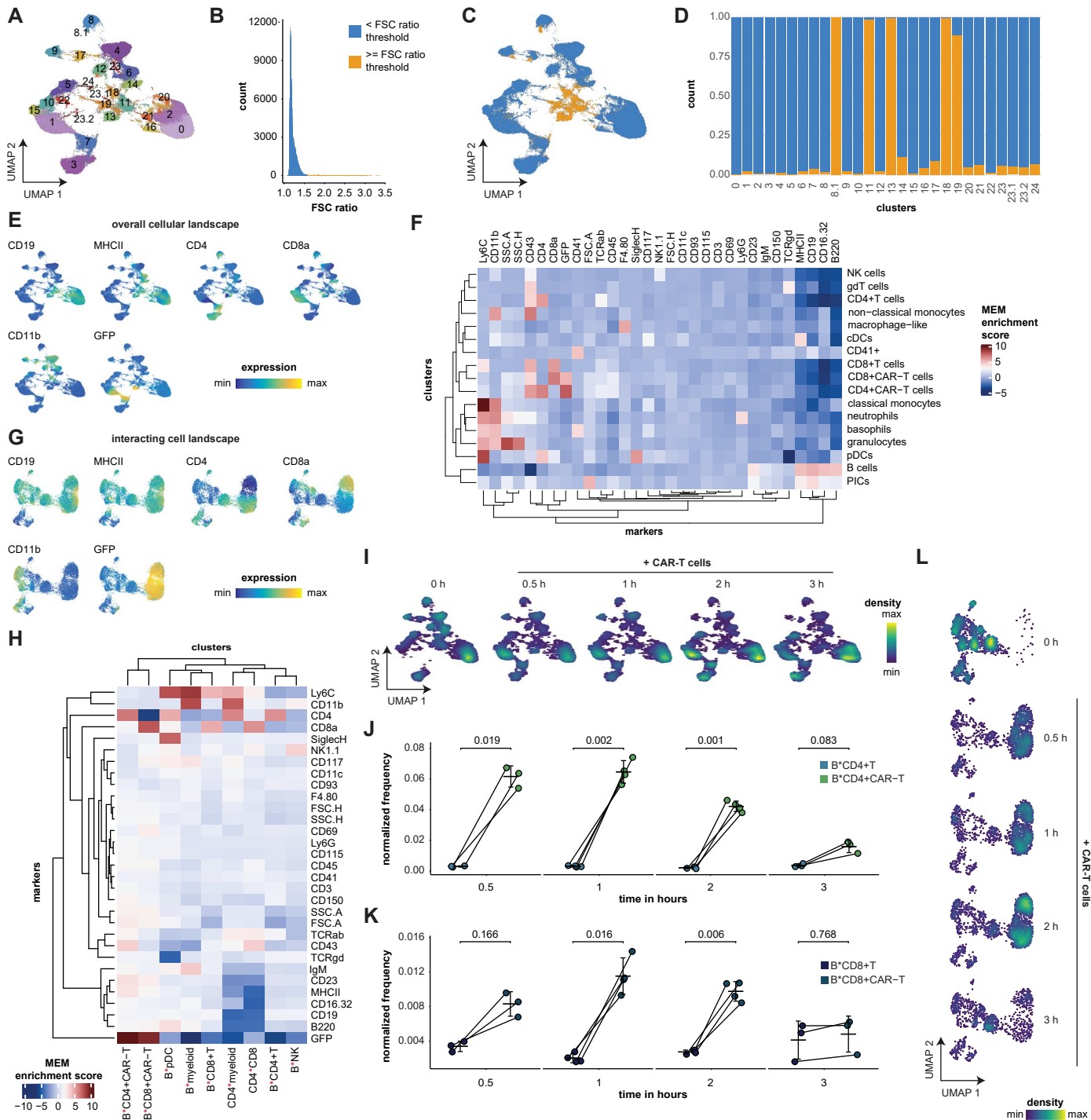

**Extended Data Fig. 4 | Interact-omics maps cellular interaction dynamics of CAR-T cells. a**. UMAP of the overall cellular landscape, corresponding to the UMAP in Fig. 3b; labels correspond to panel **d**. Decimal points indicate subclustered populations. **b**. Histogram of the FSC ratio. Cells exceeding Otsu's threshold are highlighted in orange. **c**. UMAP of the overall cellular landscape with cells above Otsu's threshold highlighted in orange. **d**. Proportions for singlets and doublets in each cluster. Clusters 8.1, 11, 13, 18, and 19 were selected as interacting cell clusters. **e**. Selected feature plots for the overall cellular landscape. **f**. Marker Enrichment Modelling (MEM) heatmap for the overall cellular landscape. **g**. Overall point density UMAPs for the control condition and CAR-T cell-treated samples at each timepoint. **h**. Interacting cell landscape corresponding to the UMAP of Fig. 3c, highlighting selected features. **i**. Marker

Enrichment Modelling (MEM) heatmap for the interacting cell landscape. **j**. Paired analysis of interactions between B cells and CD4⁺ CAR-T cells or B cells and endogenous CD4⁺ T cells. Interaction frequencies were adjusted for the singlet frequencies of the contributing cells at each timepoint (harmonic mean, see Methods), $n = 4$ technical replicates, error bars indicate the mean and standard deviation. Paired two-sided Welch's $t$-test. **k**. Paired analysis of interactions between B cells and CD8⁺ CAR-T cells or B cells and endogenous CD8⁺ T cells. Interaction frequencies were adjusted for the singlet frequencies of the contributing cells at each timepoint (harmonic mean, see Methods), $n = 4$ technical replicates, error bars indicate the mean and standard deviation. Paired two-sided Welch's $t$-test. **l**. Interacting cell point density UMAPs of the control condition and after adding CAR-T cells for the indicated time points.

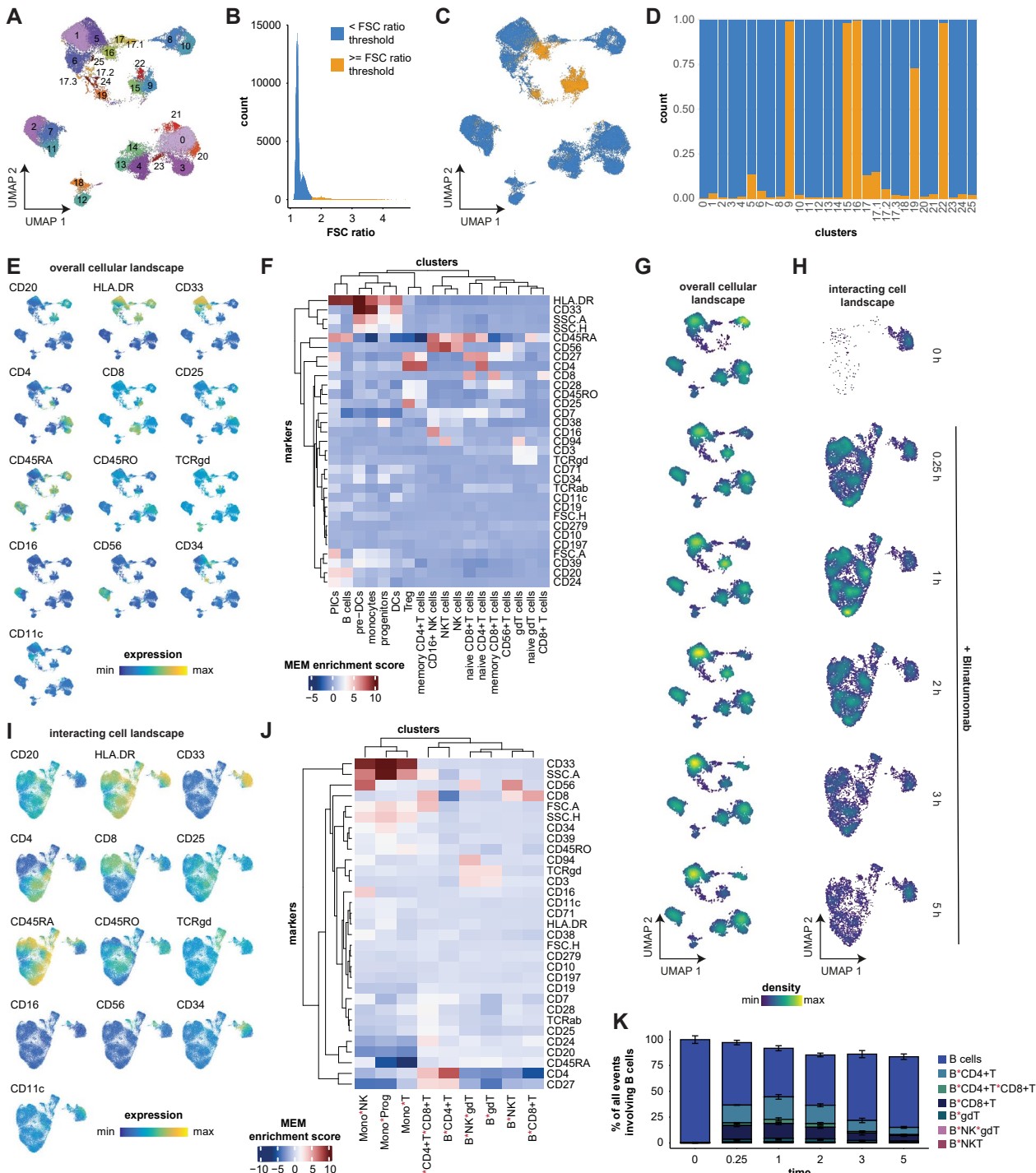

**Extended Data Fig. 5 | Cellular interaction dynamics upon blinatumomab treatment. a.** UMAP of the overall cellular landscape, corresponding to the UMAP in Fig. 3g; labels correspond to panel **d**. Decimal points indicate subclustered populations. **b**. Histogram of the FSC ratio. Cells exceeding Otsu's threshold are highlighted in orange. **c**. UMAP of the overall cellular landscape with cells exceeding Otsu's threshold highlighted in orange. **d**. Proportions for singlets and interacting cells for each cluster. Clusters 9, 15, 16, 19, and 20 were selected as interacting cell clusters. **e**. Selected feature plots for the overall cellular landscape. **f**. Marker Enrichment Modelling (MEM) heatmap for the overall cellular landscape. **g**. Overall point density UMAPs for the control condition and blinatumomab-treated samples at each timepoint. **h**. Point density UMAPs of the cellular interaction landscape for the control condition and blinatumomab-treated samples at each timepoint, corresponding to Fig. 3h. **i**. UMAP corresponding to Fig. 3h, displaying selected features of the interacting cell landscape. **j**. Marker Enrichment Modelling (MEM) heatmap for the interacting cell landscape. **k**. Composition of all B cell events across time, including single B cells and cellular interactions that involve B cells. $n = 4$ replicates from a single donor, error bars indicate the mean and standard deviation.

**A**

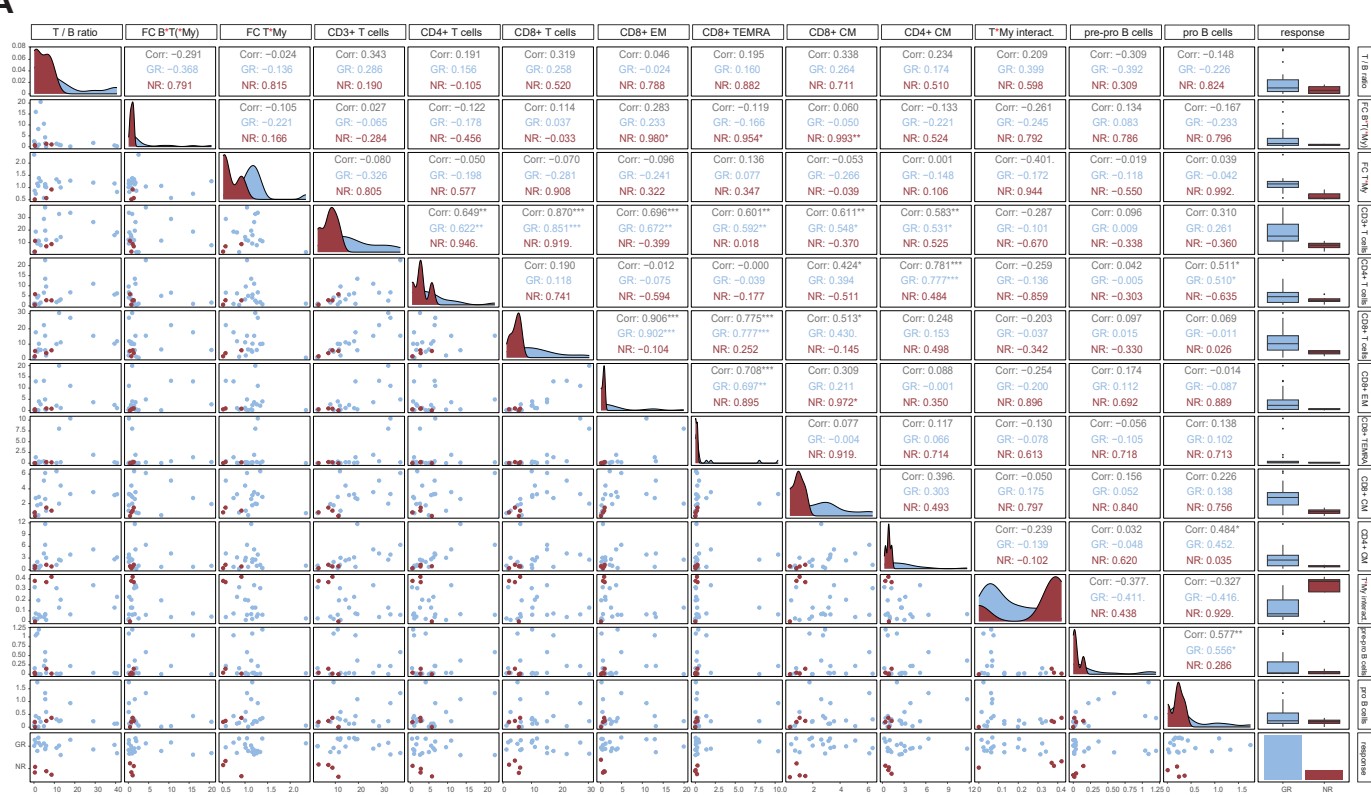

**B**

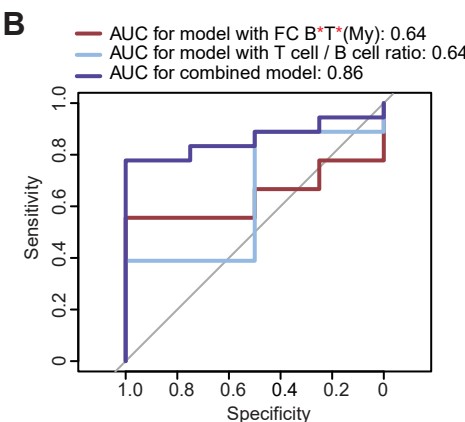

**Extended Data Fig. 6 | Interact-omics identifies independent and additive features of blinatumomab response. a**. GGally plot displaying pairwise scatter plots for the top features identified in the univariate analysis. Histograms and distributions of all features are shown on the diagonal. The Pearson correlation coefficients (Corr) are shown for all samples together and separately for the response groups (NR: nonresponder or GR: good responder), respectively. Box plots in the column on the right show comparisons of good responders (blue) vs. nonresponders (red) for all features. **b**. The Receiver Operating Characteristic (ROC) curves showing the accuracy of three logistic regression models with either the T cell/B cell ratio (light blue line), the fold change of B*T(*My) interactions between good responders and nonresponders upon blinatumomab treatment (red line) and a combined model with both features (purple line). The Area Under the Curve (AUC) metrics are indicated above the plot. Abbreviations: GR = good responder, NR = nonresponder, FC = fold change. Red asterisks in cell type labels indicate interactions between the respective cell types.

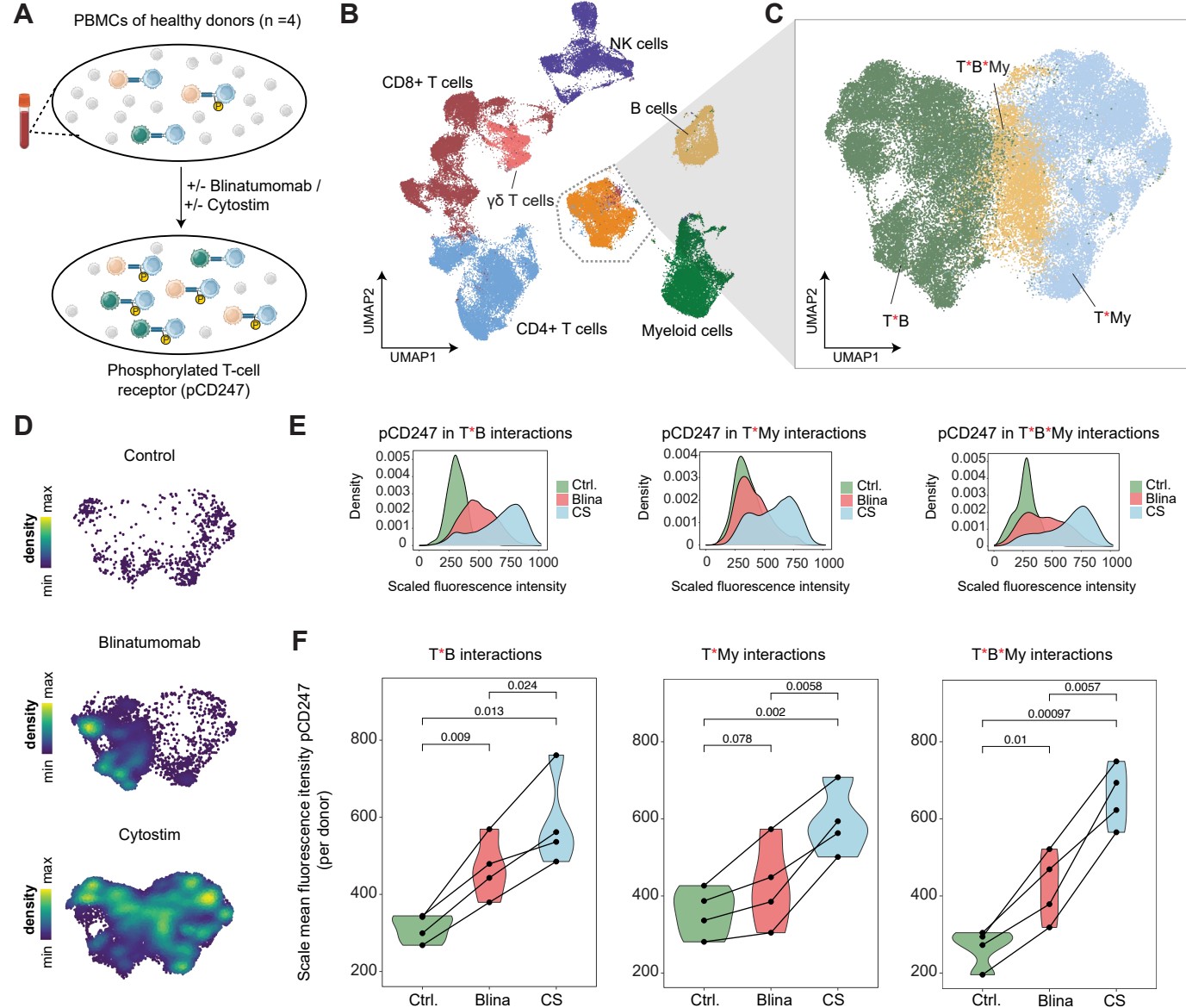

**Extended Data Fig. 7 | Characterization of functional interactions by assessment of phosphorylated CD247. a.** Schematic overview of the experimental approach. PBMCs from 4 healthy donors were incubated in the presence or absence of blinatumomab (Blina) or Cytostim (CS). **b.** UMAP of the overall cellular landscape. Recorded cells were processed with PICtR, out of 1,204,382 cells, 70,954 sketched cells are displayed. **c.** UMAP of interacting cells (*n* = 52,239) **d.** Point density UMAP of interacting cells split into the conditions. **e.** Histograms of scaled fluorescence intensity of pCD247 for each interacting cell population from panel **c.** Left panel: T*B interactions. Middle panel: T*My interactions. Right panel: T*B*My. **f.** Mean fluorescence intensity of pCD247 per donor and condition. *P* values were determined with a two-sample paired t-test (two-sided). Left panel: T*B interactions. Middle panel: T*My interactions. Right panel: T*B*My. Abbreviations: UMAP = uniform manifold approximation and projection, Ctrl. = control, Blina = blinatumomab, CS = Cytostim. Red asterisks in cell type labels indicate interactions between the respective cell types Source data. Panel **a** created using BioRender.com.

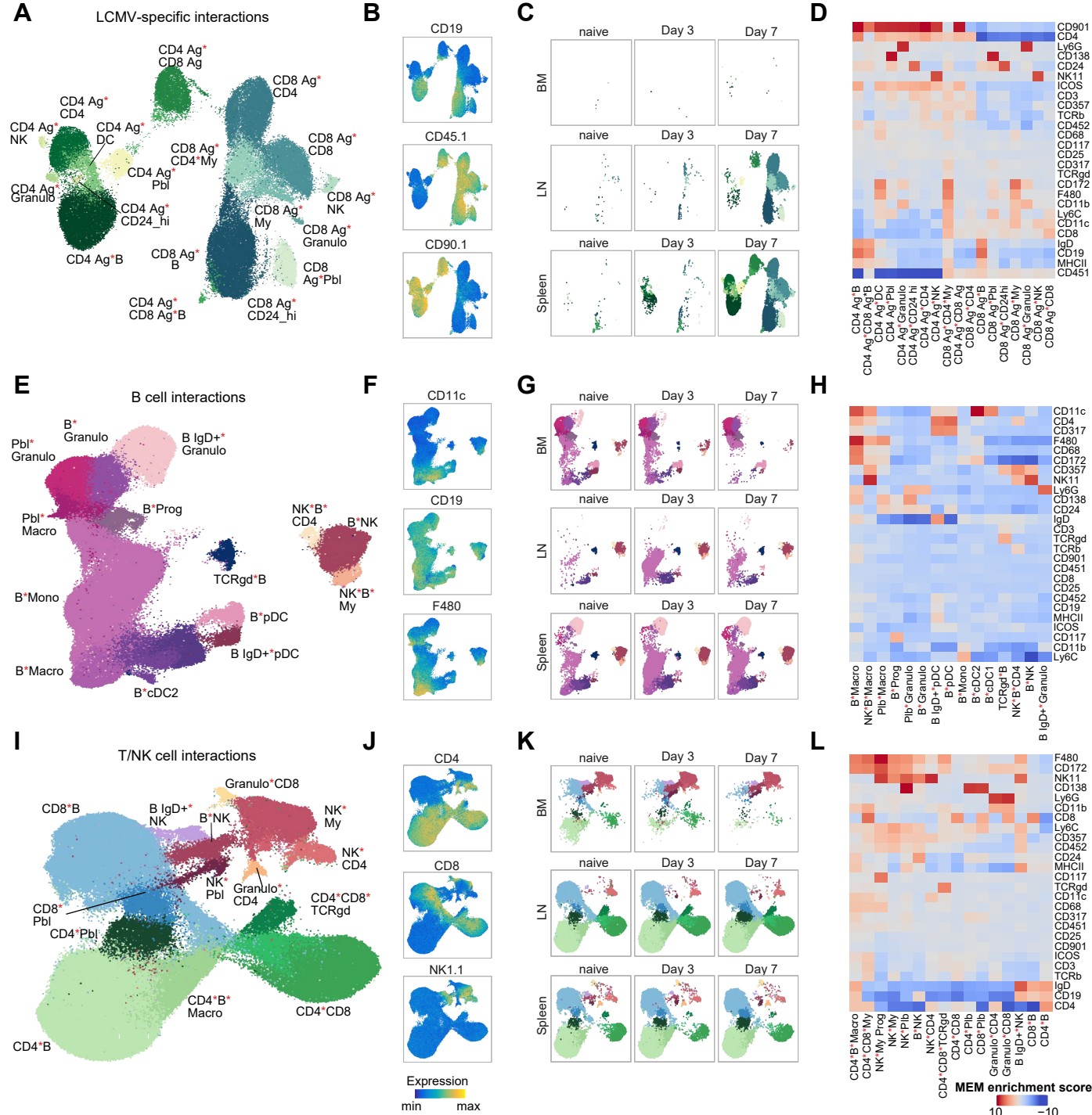

**Extended Data Fig. 8 | Interacting cell clusters used for annotation of LCMV-infected mice. a**. Cellular interactions sampled for CD45.1⁺ and CD90.1⁺ cells and based on Otsu's thresholding method, clustered, annotated and visualized in a UMAP. **b**. UMAP as in panel **a** colored by three exemplary markers **c**. UMAP as in panel **a** split by time and organ. **d**. Marker Enrichment Modelling (MEM) heatmap for LCMV-specific interactions. **e**. Cellular interactions sampled for CD19⁺ cells and based on Otsu's thresholding method, clustered, annotated and visualized in a UMAP. **f**. UMAP as in panel **e** colored by three exemplary markers **g**. UMAP as in panel **e** split by time and organ. **h**. Marker Enrichment Modelling (MEM) heatmap for cellular interactions involving B cells. **i**. Cellular

interactions sampled for CD3⁺ cells and based on Otsu's thresholding method, clustered, annotated and visualized in a UMAP. **j**. UMAP as in panel **i** colored by three exemplary markers. **k**. UMAP as in panel **i** split by time and organ. **l**. Marker Enrichment Modelling (MEM) heatmap for T and NK interactions. Abbreviations: D3 = day 3, D7 = day 7, BM = bone marrow, LN = lymph node, Ag = LCMV antigen-specific, Plb = plasmablast, HSPC = hematopoietic stem and progenitor cells, Eosino = eosinophil/basophil, PC = plasma cells, Granulo = granulocytes, Macro = macrophages, IgD = immunoglobulin D, Prog = progenitor cells, cDC = classical dendritic cells, pDC = plasmacytoid dendritic cells, My = myeloid, NK = natural killer, Mono = monocytes.

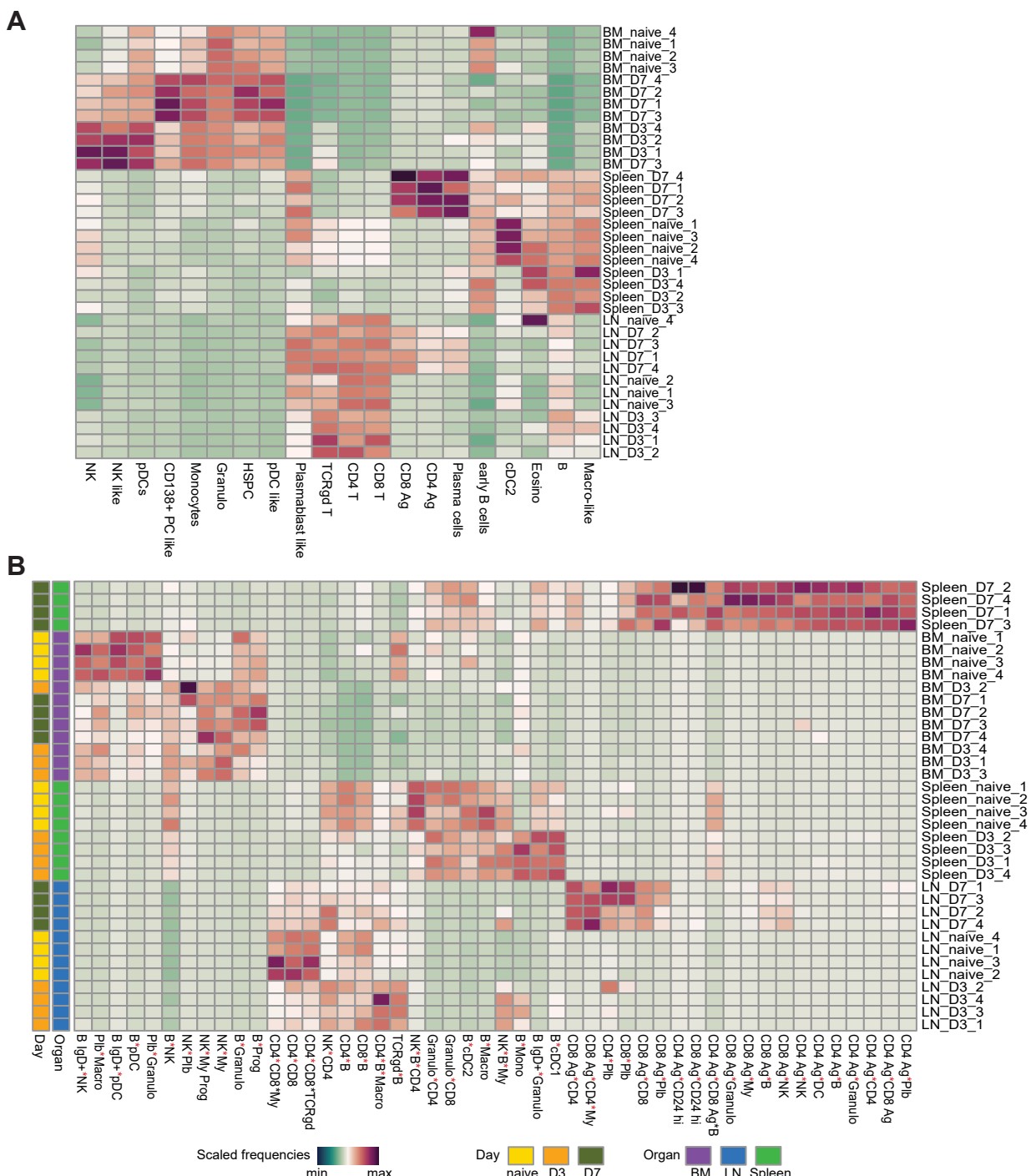

**Extended Data Fig. 9 | Cluster frequencies of single and interacting cell landscapes of LCMV-infected mice. a**. Heatmap of single cell cluster frequencies across organs, time points and replicates. **b**. Cellular interaction cluster frequencies across the organs, time points and replicates. Samples and clusters are ordered based on hierarchical clustering (for reasons of readability, the dendrograms are not shown). Abbreviations: D3 = day 3, D7 = day 7, BM = bone marrow, LN = lymph node, Ag = LCMV antigen-specific, Plb = plasmablast, HSPC = hematopoietic stem and progenitor cells, Eosino = eosinophil/basophil, PC = plasma cells, Granulo = granulocytes, Macro-like = macrophage-like, IgD = immunoglobulin D, Prog = progenitor cells, cDC = classical dendritic cells, pDC = plasmacytoid dendritic cells, My = myeloid, NK = natural killer, Mono = monocytes.

# Reporting Summary

## Statistics

For all statistical analyses, confirm that the following items are present in the figure legend, table legend, main text, or Methods section.

| n/a | Confirmed | |
|---|---|---|
| ☐ | ☒ | The exact sample size (*n*) for each experimental group/condition, given as a discrete number and unit of measurement |
| ☐ | ☒ | A statement on whether measurements were taken from distinct samples or whether the same sample was measured repeatedly |
| ☐ | ☒ | The statistical test(s) used AND whether they are one- or two-sided <br> *Only common tests should be described solely by name; describe more complex techniques in the Methods section.* |
| ☐ | ☒ | A description of all covariates tested |
| ☐ | ☒ | A description of any assumptions or corrections, such as tests of normality and adjustment for multiple comparisons |
| ☐ | ☒ | A full description of the statistical parameters including central tendency (e.g. means) or other basic estimates (e.g. regression coefficient) AND variation (e.g. standard deviation) or associated estimates of uncertainty (e.g. confidence intervals) |
| ☐ | ☒ | For null hypothesis testing, the test statistic (e.g. *F*, *t*, *r*) with confidence intervals, effect sizes, degrees of freedom and *P* value noted <br> *Give P values as exact values whenever suitable.* |
| ☒ | ☐ | For Bayesian analysis, information on the choice of priors and Markov chain Monte Carlo settings |
| ☒ | ☐ | For hierarchical and complex designs, identification of the appropriate level for tests and full reporting of outcomes |
| ☒ | ☐ | Estimates of effect sizes (e.g. Cohen's *d*, Pearson's *r*), indicating how they were calculated |

*Our web collection on statistics for biologists contains articles on many of the points above.*

## Software and code

Policy information about availability of computer code

| | |
|---|---|
| Data collection | For the acquisition of flow cytometry data, SpectroFlo® (Cytek Biosciences, v.3.2.1) or FACSDiva™ (BD, v.8) softwares were used. For imaging cytometry, FacsChorus (v1.3.82) or Cytek® INSPIRE® software (v201.1.0.826) were used. |
| Data analysis | Flow cytometry data was analyzed using FlowJo v10.10.0, PeacoQC for FlowJo v1.5.0 and R 4.3.0. The following packages were used: Seurat v5.0.0, ComplexHeatmap v2.16.0, cytoMEM v1.4.2, MASS v7.3-60, ggplot2 v3.4.4, RColorBrewer v1.1-3, pals v1.8, dplyr v1.1.3, purrr v1.0.2, readr v2.1.4, tibble v3.2.1, tidyr v1.3.0, tidyverse v2.0.0, data.table v1.14.8, rstatix v0.7.2, viridis v0.6.4, khroma v1.11.0, ggpointdensity v0.1.0, clustree v0.5.0,circlize v0.4.15, ggraph v2.1.0, viridisLite v0.4.2, pbapply v1.7-2, ggsci v3.0.0, ggpubr v0.6.0, BPCells v0.1.0, kableExtra v1.3.4, lubridate v1.9.3, forcats v1.0.0, stringr v1.5.0, sp v2.1-1, Spectre v1.0.0, GGally v2.2.0, SeuratObject v5.0.0, R v4.3.0, rpart v4.1.19, mclust 6.1.1, igraph (R) 2.0.3, CellPose 2.2.3, IDEAS 6.2, scikit 0.19.3, Python 3.12.5 , caret 6.0-94, pROC 1.18.5, autothresholdr 1.4.2, bluster 1.15.0,flowCore 2.17.0, CytoML 2.17.0, flowWorkspace 4.17.0, dbscan 1.2-0, flowSOM 2.10.0, flowMeans 1.65.0, Rclusterpp 0.2.6, immunoClust 1.37.111, clue 0.3-65, numpy 2.1.0, pandas 2.2.2, phenograph 1.5.7, leidenalg 0.10.2, matplotlib 3.9.2, seaborn 0.13.2, scipy 1.14.1, ggrepel 0.9.5, igraph (Python) 0.11.6, ggalluvial 0.12.5, PerformanceAnalytics 2.0.4, Hmisc 5.1.-3, corrplot 0.92, cola 2.11.0, e1071 1.7-14, tree 1.0-43, randomForest 4.7-1.1, Bioconductor 3.20, IDEAS® 6.2.189.0, PICtR 0.1.0 / 0.2.1, writexl 1.5.1, FACSDiva v8, SpectroFlo v.3.2.1 <br> A detailed description of the data analysis is provided in the Methods. <br> PICtR is available as an open-source R package at github.com/agSHaas/PICtR. Code to reproduce key analysis results can be found at github.com/agSHaas/ultra-high-scale-cytometry-based-cellular-interaction-mapping/. |

For manuscripts utilizing custom algorithms or software that are central to the research but not yet described in published literature, software must be made available to editors and reviewers. We strongly encourage code deposition in a community repository (e.g. GitHub). See the Nature Portfolio guidelines for submitting code & software for further information.

## Data

Policy information about availability of data

All manuscripts must include a data availability statement. This statement should provide the following information, where applicable:
- Accession codes, unique identifiers, or web links for publicly available datasets
- A description of any restrictions on data availability
- For clinical datasets or third party data, please ensure that the statement adheres to our policy

Raw and processed cytometry data for key experiments are provided at doi.org/10.5281/zenodo.10637096. Source data are provided with this paper and can be used to reproduce the analyses and figures in this manuscript.

## Human research participants

Policy information about studies involving human research participants and Sex and Gender in Research.

| | |
|---|---|
| Reporting on sex and gender | Data on sex was collected from patients of the B-ALL cohort. However, given that the patient cohort analyzed here was not part of a clinical trial, sex-specific considerations were not explicitly integrated into the study design. Furthermore, due to the limited sample size of patients with residual disease, a sex-based analysis would undermine the statistical power and reliability of such subgroup analysis and was therefore not performed. Consequently, we do not provide disaggregated sex data but the distribution of male to female participants was balanced (57% male, 43% female). |
| Population characteristics | The median age of patients in the B-ALL study was 9.5 years. Based on the Immuno-phenotype, patients were classified into distinct subgroups: pro B-ALL, pre B-ALL, common B-ALL. or MPAL The majority of patients had a unique karyotype characterized by very specific genetic alterations which included translocations, mutations, deletions, diploidy or haploidy. Due to this extensive heterogeneity, the genetic alterations were not used as covariates in the analysis. |
| Recruitment | Patients received Blinatumomab based on the decision of the treating physicians and recommendations of the national ALL-REZ BFM study center at Charité. This was not part of a clinical trial. Patients were selected for this study according to availability of sample material in the biobank and an informed consent |
| Ethics oversight | Ethics committee of Charité Universitätsmedizin Berlin (reference number: EA2/147/23). |

Note that full information on the approval of the study protocol must also be provided in the manuscript.

# Field-specific reporting

Please select the one below that is the best fit for your research. If you are not sure, read the appropriate sections before making your selection.

☒ Life sciences ☐ Behavioural & social sciences ☐ Ecological, evolutionary & environmental sciences

For a reference copy of the document with all sections, see nature.com/documents/nr-reporting-summary-flat.pdf

# Life sciences study design

All studies must disclose on these points even when the disclosure is negative.

| | |
|---|---|
| Sample size | a) No statistical methods were used to pre-determine the sample size. For mouse experiments, sample sizes were chosen based on the 3R principle, aiming to keep the number of animals to a minimum while obtaining at least 3 biological replicates. For human in-vitro data, experiments were performed with at least three technical replicates.<br>b) Experience suggests that 50 cells are sufficient to robustly identify a cell type and that a given dataset contains roughly 1-5 % of physically interacting cells. The number of cells acquired was sufficient for the analysis. |
| Data exclusions | a) Where applicable, PeacoQC or FlowAI were used to exclude low quality flow cytometry events based on inconsistencies in signal acquisition and speed. Cells were gated accoring to the gating strategy described in the "Flow Cytometry" section. Additionally, cells were removed when high autofluoresence or signal anomalies suggested a low quality event.<br>b) For the B-ALL cohort (n = 42), we compared patients that could unequivocally be categorized into good responders (n = 18) and non-responders (n = 4) in order to explore mechanisms underlying therapy response among patients with residual disease. The remaining 20 patient samples were therefore excluded from the downstream analysis.<br>c) Data points were excluded from the downstream analysis if a population of cells was not detectable across all time points. The excluded populations are noted in the respective figure legends.<br>d) Clusters of physically interacting cells without a cell type exclusive marker combination might represent homotypic interactions and were excluded from the downstream analysis. |
| Replication | All mouse and in-vitro human experiments were performed with at least n = 3 independent or technical replicates and findings could be replicated successfully. |

| Randomization | Mice, murine samples, and PBMC samples from healthy blood donors were randomly allocated to groups. Randomization is not applicable to the B-ALL cohort, since all patient samples were measured in the presence and absence of Blinatumomab. |
|---|---|
| Blinding | Blinding was not feasible due to the necessity of knowing the treatment/control group allocations for accurate data interpretation and analysis. |

# Reporting for specific materials, systems and methods

We require information from authors about some types of materials, experimental systems and methods used in many studies. Here, indicate whether each material, system or method listed is relevant to your study. If you are not sure if a list item applies to your research, read the appropriate section before selecting a response.

## Materials & experimental systems

| n/a | Involved in the study |
|---|---|
| ☐ | ☒ Antibodies |
| ☒ | ☐ Eukaryotic cell lines |
| ☒ | ☐ Palaeontology and archaeology |
| ☐ | ☒ Animals and other organisms |
| ☒ | ☐ Clinical data |
| ☒ | ☐ Dual use research of concern |

## Methods

| n/a | Involved in the study |
|---|---|
| ☒ | ☐ ChIP-seq |
| ☐ | ☒ Flow cytometry |
| ☒ | ☐ MRI-based neuroimaging |

## Antibodies

| Antibodies used | Antibodies (Epitope, Fluorochrome, Vendor, Identifier (RRID), Clone):<br><br>Imaging Flow Cytometry:<br>Anti-CD33 BV421 Biolegend Cat# 303416 WM53<br>Anti CD19 BV605 Biolegend Cat# 363023 SJ25C1<br>Anti-HLA-DR BB515 BD Biosciences Cat# 564516 G46-6<br>Anti-CD3 PE-Cy7 Biolegend Cat# 300419 UCHT1<br>Anti-CD45 APC Thermo Fisher Scientific Cat# 17-0459-42 HI30<br>Zombie NIR Biolegend Cat# 423105<br><br>CytoStim Experiment:<br>Anti-CD16 BUV395 BD Biosciences Cat# 563785 3G8<br>Anti-CD8 BUV496 BD Biosciences Cat# 741199 SK1<br>Anti-CD33 BUV563 BD Biosciences Cat# 741369 WM33<br>Anti-CD27 BUV661 BD Biosciences Cat# 741609 M-T271<br>Anti-CD4 BUV737 BD Biosciences Cat# 612748 SK4<br>Anti-CD14 BUV805 BD Biosciences Cat# 612902 M5E2<br>Anti-CD141 BV421 BD Biosciences Cat# 565321 1A4<br>Anti-CD197 Pacific Blue BioLegend Cat# 353210 G043H7<br>Anti-CD20 BV480 BD Biosciences Cat# 566132 2H7<br>Fixable viability dye efluor506 Thermo Fisher Scientific Cat# 65-0866-14<br>Anti-CD11b BUV605 BD Biosciences Cat# 563015 M1/70<br>Anti-CD3 CD650 BD Biosciences Cat# 563851 UCHT1<br>Anti-CD56 BV711 BD Biosciences Cat# 563169 NCAM16.2<br>Anti-TCRab BV750 BD Biosciences Cat# 747180 IP26<br>Anti-CD45RA BV785 BD Biosciences Cat# 564552 HI100<br>Anti-CD45RO FITC Biolegend Cat# 304242 UCHL1<br>Anti-CD123 PerCP-Cy5.5 BioLegend Cat# 306016 6H6<br>Anti-CD19 BB700 BD Biosciences Cat# 566396 SJ25C1<br>Anti-HLA-DR PE Thermo Fisher Scientific Cat# 12-9956-42 LN3<br>Anti-CD1c PE-Dazzle 594 BioLegend Cat# 331532 L161<br>Anti-CD154 PE-Cy5 Thermo Fisher Scientific Cat# 16-1541-82 MR1<br>Anti-CD11c PE-Cy7 BD Biosciences Cat# 561356 B-ly6<br>Anti-CD45 APC Thermo Fisher Scientific Cat# 17-0459-42 HI30<br>Anti-CD34 Alexa Fluor700 BD Biosciences Cat# 659123 8G12<br>Anti-CD69 APC-Cy7 BD Biosciences Cat# 560912 FN50<br><br><br>OT-II Experiment:<br>Anti-CD19 BUV395 BD Biosciences Cat# 563557 1D3<br>Anti-CD69 BUV737 BD Biosciences Cat# 612793 H1.2F3<br>Anti-CD11b BUV805 BD Biosciences Cat# 568345 M1/70<br>Anti-CD11c BV421 BD Biolegend Cat# 117343 N418<br>Anti-CD45.1 BV605 BioLegend Cat# 110737 A20<br>Anti-TCRb BV711 BD Cat# 743002 H57-597 |
|---|---|

Anti-MHC-II BV786 BioLegend Cat# 107645 2G9
Anti-CD44 FITC eBioscience Cat# 11-0441-82 IM7
Anti-CD86 PE BioLegend Cat# 105105 PO3
Anti-CD4 RY586 BD Biosciences Cat# 568161 GK1.5
Anti-CD14 PE-Cy7 BioLegend Cat# 123316 Sa14-2
Anti-CD25 APC  BioLegend Cat #102053 PC61
Anti-CD8a AF700 eBioscience Cat# 56-0081-82 53-6.7
Anti-CD45.2 APC efluor780 eBioscience Cat# 47-0454 104

CAR-T Experiment:
Anti-CD11c BUV395 BD Biosciences Cat# 564080 HL3
Anti-CD4 BUV496 BD Biosciences Cat# 612952 GK1.5
Anti-CD115 BUV563 BD Biosciences Cat# 748478 CDS-1R
Anti-CD43 BUV615 BD Biosciences Cat# 752307 S7
Anti-CD16/32 BUV737 BD Biosciences Cat# 612783 2.4G2
Anti-MHCII BUV805 BD Biosciences Cat# 748844 M5/114.15.2
Anti-F4/80 SB436 Thermo Fisher Scientific Cat# 562606 BM8
Anti-Ly6G BV480 BD Biosciences Cat# 746448 A18
Fixable viability dye efluor506 Thermo Fisher Scientific Cat# 65-0866-14
Anti-CD3 BV570 Biolegend Cat# 100225 17A2
Anti-CD11b BV605 Biolegend Cat# 101237 M1/70
Anti-CD23 BV650 BD Biosciences Cat# 740456 B3B4
Anti-CD117 BV711 Biolegend Cat# 105835 2B8
Anti-CD150 BV785 Biolegend Cat# 115937 TC15-12F12.2
Anti-CAR GFP
Anti-NK1.1 PerCP Biolegend Cat# 108725 PK136
Anti-CD93 BB700 BD Biosciences Cat# 742187 AA4.1
Anti-SiglecH PerCP efluor710 Thermo Fisher Scientific Cat# 46-0333-82 eBio440c
Anti-CD69 PE Dazzle Biolegend Cat# 104535 H1.2F3
Anti-IgM PE-Cy5 Biolegend Cat# 406544 RMM-1
Anti-CD8a PE-Fire700 Biolegend Cat# 100792 53-6.7
Anti-Ly6C PE-Cy7 Biolegend Cat# 128018 HK1.4
Anti-B220 PE-Fire 810 Biolegend Cat# 103287 RA3-6B2
Anti-CD41 APC Biolegend Cat# 133914 MWReg30
Anti-TCRab AF647 Biolegend Cat# 109218 H57-597
Anti-CD19 SPARK-NIR Biolegend Cat# 115568 6D5
Anti-TCRgd R718 BD Biosciences Cat# 751919 GL3
Anti-CD45 APC-Fire810 Biolegend Cat# 103174 30-F11
Blinatumomab Time Course:
AAnti-CD16 BUV395 BD Biosciences Cat# 563785 3G8
Anti-CD19 BUV496 BD Biosciences Cat# 612938 SJ25C1
Anti-CD33 BUV563 BD Biosciences Cat# 741369 WM53
Anti-CD24 BUV615 BD Biosciences Cat# 751122 ML5
Anti-CD27 BUV661 BD Biosciences Cat# 741609 M-T271
Anti-CD8 BUV737 BD Biosciences Cat# 612754 SK1
Anti-CD45RO BUV805 BD Biosciences Cat# 748367 UCHL1
Anti-CD28 BV421 BD Biosciences Cat# 742525 L293
Anti-CD39 BV480 BD Biosciences Cat# 746454 TU66
Anti-CD71 BV510 BD Biosciences Cat# 743305 M-A712
Anti-CD11c BV605 Biolegend Cat# 301636 3-Sept.
Anti-CD279 BV650 BD Biosciences Cat# 564104 EH12.1
Anti-CD94 BV711 BD Biosciences Cat# 743952 HP-3D9
Anti-TCRab BV750 BD Biosciences Cat# 747180 IP26
Anti-CD45RA BV786 BD Biosciences Cat# 563870 HI100
Anti-Caspase 3/7 probe Thermo Fisher Scientific Cat# C10423
Anti-CD3 Spark Blue Biolegend Cat# 344852 SK7
Anti-CD38 PerCP Biolegend Cat# 303520 HIT2
Anti-CD10 PerCPVio700 Miltenyi Cat# 130-114-5067 REA877
Anti-CD197 PE BD Biosciences Cat# 561008 3D12
Anti-CD56 PE-CF594 BD Biosciences Cat# 564963 R19-760
Anti-CD7 PE-Cy5 Biolegend Cat# 343110 CD7-6B7
Anti-CD25 PE-Fire700 Biolegend Cat# 356145 M-A251
Anti-TCRgd PE-Cy7 BD Biosciences Cat# 655410 11F2
Anti-CD4 PE-Fire810 Biolegend Cat# 344677 SK4
Anti-CD20 APC Biolegend Cat# 302309 2H7
Anti-CD34 APC-R700 BD Biosciences Cat# 659123 8G12
Anti-HLA-DR APC-Cy7 Biolegend Cat# 307618 L243

T Cell Sort:
Anti-CD44 FITC Thermo Fisher Scientific Cat# 11-0441-82 IM7
Anti-CD62L PE-Cy7 BioLegend Cat# 104418 MEL-14
Anti-CD4 APC-Cy7 BioLegend Cat# 100414 GK1.5
Anti-CD8 BUV395 BD Biosciences Cat# 563786 53-6.7

Blinatumomab Treatment:
Anti-CD16 BUV395 BD Biosciences Cat# 563785 3G8

Anti-CD19 BUV496 BD Biosciences Cat# 612938 SJ25C1
Anti-CD33 BUV563 BD Biosciences Cat# 741369 WM53
Anti-CD24 BUV615 BD Biosciences Cat# 751122 ML5
Anti-CD27 BUV661 BD Biosciences Cat# 741609 M-T271
Anti-CD8 BUV737 BD Biosciences Cat# 612754 SK1
Anti-CD45 BUV805 BD Biosciences Cat# 612891 HI30
Anti-CD10 BV421 BD Biosciences Cat# 312218 HI10a
Anti-IgD Pacific Blue Biolegend Cat# 348223 IA6-2
Anti-CD39 BV480 BD Biosciences Cat# 746454 TU66
Anti-CD71 BV510 BD Biosciences Cat# 743305 M-A712
Anti-CD20 PacOrange, BV570 Biolegend Cat# 302331 2H7
Anti-CD11c BV605 Biolegend Cat# 301636 3.9
Anti-CD123 BV650 BD Biosciences Cat# 563405 7G3
Anti-CD56 BV711 Biolegend Cat# 318336 HCD56
Anti-TCRab BV750 BD Biosciences Cat# 747180 IP26
Anti-CD45RA BV786 BD Biosciences Cat# 563870 HI100
Anti-CD57 BB515 BD Biosciences Cat# 565945 NK-1
Anti-CD11b FITC Biolegend Cat# 101206 M1/70
Anti-CD3 SparkBlue Biolegend Cat# 344852 SK7
Anti-CD38 PerCP Biolegend Cat# 303520 HIT2
Anti-CD94 BB700 BD Biosciences Cat# 566534 HP-3D9
Anti-TCRgd PerCP eF710 Invitrogen Cat# 46-9959-42 B1.1
Anti-CD30L PE R&D systems Cat# FAB1028P 116614
Anti-CD279 RY586 BD Biosciences Cat# 568119 EH12.1
Anti-CD1c PE Dazzle Biolegend Cat# 331532 L161
Anti-Tigit PE-Fire 640 Biolegend Cat# 372743 A15153G
Anti-CD25 PE-Fire700 Biolegend Cat# 356145 M-A251
Anti-CD14 PE Cy7 Tonbo Cat# 60-0149-T100 61D3
Anti-CD4 PE-Fire810 Biolegend Cat# 344677 SK4
Anti-CD197 APC BD Biosciences Cat# 566762 2-L1-A
Anti-CD160 AF647 Biolegend Cat# 341203 BY55
Anti-CD69 SPARK-NIR Biolegend Cat# 310957 FN50
Anti-CD127 APC R700 BD Biosciences Cat# 565185 HIL-7R-M21
Anti-CD34 APC-Cy7 Biolegend Cat# 343514 581
Anti-HLA-DR APC-Fire810 Biolegend Cat# 307674 L243

LCMV Experiment:
Anti-MHC-II BUV395 BD Biosciences Cat# 743876 2G9
Anti-Live Dead blue Thermo Fisher Scientific Cat# L34961
Anti-CD24 BUV563 BD Biosciences Cat# 749336 M1/69
Anti-CD11c BUV615 BD Biosciences Cat# 751222 N418
Anti-ICOS BUV737 BD Biosciences Cat# 567919 C398.4A
Anti-CD11b BUV805 BD Biosciences Cat# 568345 M1/70
Anti-CD45.1 BV421 Biolegend Cat# 110732 A20
Anti-CD19 BV480 BD Biosciences Cat# 566167 1D3
Anti-CD117 BV510 Biolegend Cat# 105839 2B8
Anti-Ly6G BV570 Biolegend Cat# 127629 1A8
Anti-CD138 BV605 Biolegend Cat# 142515 281-2
Anti-Ly6C BV650 Biolegend Cat# 128049 HK1.4
Anti-CD90.1 BV711 Biolegend Cat# 202539 OX-7
Anti-CD25 BV785 Biolegend Cat# 102051 PC61
Anti-CD45.2 FITC Biolegend Cat# 109806 104
Anti-CD317 PerCP-efluor710 Biolegend Cat# 127021 927
Anti-CD8 RB780 BD Biosciences Cat# 568692 53-6.7
Anti-CD68 PE Biolegend Cat# 137013 FA-11
Anti-CD4 RY586 BD Biosciences Cat# 568161 GK1.5
Anti-CD172 PE-Dazzle594 Biolegend Cat# 144015 P84
Anti-F480 PE-Cy5 Biolegend Cat# 123111 BM8
Anti-CD357 PE-Cy7 Biolegend Cat# 126309 DTA-1
Anti-TCRb APC Biolegend Cat# 109212 H57-597
Anti-IgD SparkNIR 685 Biolegend Cat# 405749 11-26c.2a
Anti-TCRgd R718 BD Biosciences Cat# 751919 GL3
Anti-CD3 APC-Cy7 BD Biosciences Cat# 561042 145-2C11
Anti-NK1.1 APC-Fire810 Biolegend Cat# 156519 S17016D

Intracellular signaling (revision):
Anti-CD4 Spark UV387 Biolegend Cat# 344686 SK3
Anti-CD3 BUV395 BD Biosciences Cat# 563548 SK7
Anti-CD8 BUV496 BD Biosciences Cat# 741199 SK1
Anti-CD45RO BUV805 BD Biosciences Cat# 748367 UCHL1
Anti-CD19 BV421 Biolegend Cat# 302233 SJ25C1
Anti-CD14 violetFluor 450 Tonbo Cat# 75-0149-T100 61D3
Anti-CD20 BV570 Biolegend Cat# 302331 2H7
Anti-CD56 BV711 Biolegend Cat# 318336 HCD56
Anti-CD45RA BV786 BD Biosciences Cat# 563870 HI100

Anti-CD27 RB705 BD Biosciences Cat# 757295 L128
Anti-Ki67 RB744 BD Biosciences Cat# 570503  B56
Anti-PLCy1 PE Miltenyi Cat# 130-104-969 REA341
Anti-CD94 RY586 BD Biosciences Cat# 753479 HP-3D9
Anti-CD33 PE-Dazzle594 Biolegend Cat# 303431 WM53
Anti-CD197 PE-Fire640 Biolegend Cat# 353261 G043H7
Anti-CD25 PE-Fire700 Biolegend Cat# 356145 M-A251
Anti-TCRgd PE-Cy7 BD Biosciences Cat# 655410 11F2
Anti-pCD247 AF647 BD Biosciences Cat# 558489 K25-407.69
Anti-CD16 cFluor R720 Cytek Biosciences Cat# R7-20006 3G8
Live Dead Zombie NIR Biolegend Cat# 423105
Anti-HLA-DR APC-Cy7 Biolegend Cat# 307618 L243
Anti-CD45 APC-Fire810 Biolegend Cat# 304076 HI30

Imaging flow cytometry (revision):
Anti-CD45.2 RB545 BD Biosciences Cat# 756290 104
Anti-CD8 PE BioLegend Cat# 100707 53-6.7
Anti-CD4 Pe-Fire640 BioLegend Cat# 100481 GK1.5
Anti-CD90.1 Pe-Cy7 BioLegend Cat# 202518 OX-7
Anti CD45.1 BV421 BioLegend Cat# 110732 A20
Anti-CD3 BV510 BioLegend Cat# 100234 17A2
Anti-CD19 SPARK-NIR 587 BioLegend Cat# 115568 6D5

LCMV experiment (revision):
Anti-CD19 Spark UV 387 Biolegend Cat# 115585 6D5
Anti-CD48 BUV395 BD Biosciences Cat# 740236 HM48-1
Anti-CD4 BUV496 BD Biosciences Cat# 612952 GK1.5
Anti-CD44 BUV563 BD Biosciences Cat# 741227 IM7
Anti-CD43 BUV615 BD Biosciences Cat# 752307 S7
Anti-CD71 BUV661 BD Biosciences Cat# 741481 C2
Anti-CD24 BV737 BD Biosciences Cat# 612832 M1/69
Anti-CD62L BUV805 BD Biosciences Cat# 741924 MEL-14
Anti-CD45.1 BV421 Biolegend Cat# 110732 A20
Anti-SiglecF SB436 Thermo Fisher Scientific Cat# 62-1702-82 1RNM44N
Anti-CD105 Pacific Blue Biolegend Cat# 120411 MJ7/18
Anti-Ly6G BV480 BD Biosciences Cat# 746448 A18
Anti-CD3 BV510 Biolegend Cat# 100234 17A2
Anti-NK1.1 BV570 Biolegend Cat# 108733 PK136
Anti-CD172a BV605 BD Biosciences Cat# 740390 P84
Anti-CD23 BV650 BD Biosciences Cat# 740456 B3B4
Anti-CD117 BV711 Biolegend Cat# 105835 2B8
Anti-CD138 BV785 Biolegend Cat# 142534 281-2
Anti-CD45.2 RB545 BD Biosciences Cat# 756290 104
Anti-CD21/35 APC BD Biosciences Cat# 123412 7E9
Anti-Ly6C PerCP Biolegend Cat# 128028 HK1.4
Anti-CD317 BB700 BD Biosciences Cat# 747601 927
Anti-IgM PerCP-efluor710 Thermo Fisher Scientific Cat# 46-5790-80 II/41
Anti-CD11b RB744 BD Biosciences Cat# 570513 M1/70
Anti-MHCII PE Tonbo Cat# 50-5321-U100 M5/114.15.2
Anti-F4/80 Spark YG 593 Biolegend Cat# 157311 QA17A29
Anti-CD64 PE-Dazzle594 Biolegend Cat# 164412 W18349C
Anti-CD25 PE-Fire640 Biolegend Cat# 102071 PC61
Anti-CD11c PE-Cy5 Biolegend Cat# 117316 N418
Anti-CD8a PE-Fire700 Biolegend Cat# 100792 53-6.7
Anti-CD90.1 PE-Cy7 Biolegend Cat# 202518 OX-7
Anti-B220 PE-Fire810 Biolegend Cat# 103287 RA3-6B2
Anti-CD41 APC Biolegend Cat# 133914 MWReg30
Anti-TCRab AF647 Biolegend Cat# 109218 H57-597
Anti-TCRgd R718 BD Biosciences Cat# 751919 GL3
Live Dead Zombie NIR Biolegend Cat# 423105
Anti-Sca1 APC-Cy7 BD Biosciences Cat# 560654 D7

Validation

All antibodies used in this study are commercially available, broadly established, and validated by the respective manufacturers for the indicated species and applications, as detailed on their websites (see RRIDs above for each antibody). Validation information for each primary antibody includes species reactivity, specificity, and application data provided by the manufacturers.

In addition, all primary antibodies have been routinely used in our laboratory with reproducible and consistent results across multiple experiments and independent batches. This includes verification of expected staining patterns in positive control tissues/cells and the absence of non-specific staining in negative controls

# Animals and other research organisms

Policy information about studies involving animals; ARRIVE guidelines recommended for reporting animal research, and Sex and Gender in Research

| | |
|---|---|
| Laboratory animals | Mice were maintained in individually ventilated cages under SPF conditions in the animal facility of the DKFZ (Heidelberg, Germany) or at the Charité animal facility (FEM, Berlin, Germany) with ad libitum access to water and food (22 ± 2 °C, 45-65 % humidity, 12h light-dark cycle). Mice used in LCMV experiments were 7 weeks old; all other mice were between 6-20 weeks old. CD45.1 mice were obtained from in house breeding at DKFZ (Z110I02, B6.SJL- Ptprca Pepcb/BoyJ) or from Charles Rivers (B6.SJL-PtprcaPepcb/BoyCrl). For experiments with antigen-specific T cells, cells were isolated from B6.Cg-Tg(TcraTcrb)425Cbn/J (OT-II) or LCMV-TCRtg P1454 and Smarta55 mice expressing the congenic markers CD45.1 or CD90.1. All other mice were C57BL6/J. |
| Wild animals | The study did not involve wild animals. |
| Reporting on sex | All mice were female. |
| Field-collected samples | No samples were collected from the field. |
| Ethics oversight | Unless otherwise stated, animal experiments were conducted under German law and approved by Regierungspräsidium Karlsruhe (approval number DKFZ299, G-55/20, G-56/20) or the Landesamt für Gesundheit und Soziales in Berlin (LAGeSo, G0016/20). |

Note that full information on the approval of the study protocol must also be provided in the manuscript.

# Flow Cytometry

## Plots

Confirm that:

☒ The axis labels state the marker and fluorochrome used (e.g. CD4-FITC).

☒ The axis scales are clearly visible. Include numbers along axes only for bottom left plot of group (a 'group' is an analysis of identical markers).

☒ All plots are contour plots with outliers or pseudocolor plots.

☒ A numerical value for number of cells or percentage (with statistics) is provided.

## Methodology

| | |
|---|---|
| Sample preparation | a) PBMC samples from healthy blood donors were obtained as buffy coats from the blood donation center IKTZ Heidelberg. Mononuclear cells were isolated by Ficoll (GE Healthcare) density gradient centrifugation and stored in FCS 10% DMSO in liquid nitrogen until usage. Cryopreserved PBMCs were thawed in a water bath at 37°C, transferred to 10% FCS RPMI-1640 and washed twice. After each washing step, cells were centrifuged at 350g for 5 min. $2 \times 10^5$ cells were plated in 10% FCS RPMI-1640 and cultured short term for up to 5h in 200 µl RPMI 10% FCS. CytoStim (Miltenyi) was used in concentrations recommended by the manufacturer at 37°C for 2 hours before harvest. For experiments using a Blinatumomab analog (Invivogen), a concentration of 50 ng/ml was used. The incubation period ranged from 0.25 to 5 hours at 37°C and 5% CO2 in 96-well U-bottom plates. For experiments assessing the stability of Blinatumomab-induced interactions upon cryopreservation, cells were either incubated for 2h in presence of the compound and stained with surface antibodies and fixed with 4% PFA (ThermoFisher) or frozen in Bambanker freezing medium (Nippon Genetics), thawed after 18h and treated in the same way as the non-frozen cells. |

b) Blinatumomab response analysis:  Bone marrow aspirates obtained from 42 relapsed B-ALL patients were thawed in a water bath at 37°C, transferred to 10% FCS RPMI-1640 and washed twice. After thawing, each sample was split into two. One half of the sample was cultured in 200 µl RPMI 1640 (10% FCS) supplemented with 50ng/ml Blinatumomab analog (Invivogen) for 1 hour at 37°C and 5% CO2 in a 96 well U-bottom plate. The other half of the sample was cultured in RPMI 1640 (10% FCS) without Blinatumomab supplementation for 1h at the same conditions. After the incubation, cells were harvested, washed with FACS buffer, stained with the surface marker panel and analyzed.

c) For in vitro benchmarking experiments, human PBMCs were treated with CytoStimTM as described above; control groups were left untreated. Subsequently, cells were split into two groups each and stained with CD45-APC-Fire810 or CD45-PE-Fire640, respectively. After mixing the labelled groups, cells were incubated for 0-4h at 4 °C (200 000 cells/well in 50 µL during staining/acquisition) or processed at seeding densities of 25,000 to 250,000 cells per well in 96-well plates (50 µL during staining/acquisition). Subsequently, cells were harvested, washed with FACS-buffer, stained with surface markers, fixed with 2 % PFA (except the non-fixed control) and analyzed.

d) For measuring phosphorylated CD247, human PBMCs were seeded at 100,000 cells/well in 200 µL and treated for 1h with Blinatumomab (160 ng/mL) or CytoStimTM as described above. Following the stimulation period, cells were fixed immediately by adding CytoFix buffer (15 min, 4°C). Cells were washed and resuspended in 200 µL 2.5x Perm/Wash buffer, incubated for 30 min at 37°C, and stained overnight at 4°C before analysis.

e) Mice were euthanized through cervical dislocation. For isolation of antigen-specific T cells, the spleen and various lymph nodes (including inguinal, axial, submandibular, and mesenteric) were carefully extracted. Tissues were homogenized using a 40µm filter (Falcon) and a syringe plunger in cold RPMI (Sigma Aldrich) with 2% FCS (Gibco by Lifetechnologies).

Subsequently, single-cell suspensions from spleens were treated with erythrocyte lysis solution (ACK buffer, containing 0.15 M NH4Cl, 1 mM KHCO3, and 0.1 mM Na2EDTA in water from Lonza) for a duration of 5 minutes. For some readouts, these suspensions were combined with the lymph node samples or maintained separate. CD4+ and CD8+ T cells were purified using either the Dynabeads Untouched Mouse CD4 Cells Kit (Invitrogen) or the murine CD4+ T cell isolation kit and the murine CD8+ T cell isolation kit (Miltenyi) according to the manufacturer's instructions. Purified fractions were stained for further purification using FACS (see section Flow cytometry, cell sorting and image cytometry). For in vivo experiments, femurs, spleen and various lymph nodes were dissected and kept separate on ice. Lymph nodes and spleens were individually processed as described above. Femurs were flushed using FACS buffer and homogenized using a 40µm filter (Falcon) and a syringe plunger.

f) For the in vivo benchmarking experiment, LCMV-specific CD4+ T cells were transferred into C57BL6 (CD45.2) hosts 5 days prior to infection as described above. CD45.1 (B6.SJL-PtprcaPepcb/BoyCrl) and CD45.2 hosts were infected intraperitoneally as described above, and spleens were harvested on day 7 post-infection. Spleens were split into 4 equal pieces and mixed across CD45.1/CD45.2 hosts for joint tissue homogenization (see Supplementary Figure 11A). Mixed samples were processed for spectral flow cytometry analysis.

Unless otherwise stated, cell suspensions were resuspended in 2% FSC 0,5 mM EDTA PBS (FACS buffer) for performing flow cytometric stainings. For ex vivo readouts with bi-specific engagers and antigen specific T cells, cells were harvested, centrifuged 5 min at 350 g and stained with surface marker panel master mixes using FACS buffer and addition of Brilliant Stain buffer (BD) according to the manufacturer's recommendation. Cells were stained for 30 min on ice in 96-well V-bottom plates, followed by washing with FACS buffer, centrifugation for 5 min at 350g and resuspension in 200 µl FACS buffer. For more time-consuming in vivo experiments, cells were labeled with fixable dead cell exclusion dyes followed by fixation of obtained single-cell suspensions with cold 2% PFA PBS for 15 min at room temperature. Cells were washed, centrifuged for 5 min at 350 g and then stained for 12h at 4°C. After washing and centrifugation for 5 min at 350g, cells were filtered through a 35-µm cell strainer and kept on ice until flow cytometric analysis.

For image-enabled cell sorting, PBMCs were incubated for 2h with CytoStim, stained with surface markers followed by fixation with 2% PFA PBS as described above and operated using a 100 µm sort nozzle, with the piezoelectric transducer driven at 34 kHz and automated stream setup by BD FACSChorusTM Software, and a system pressure of 20 psi.

For the ImageStream®X experiment, data was acquired using the Cytek® INSPIRE™ software. ImageStream®X fluorescence intensity values (based on the sum of the pixel intensities in the mask as selected by ImageStream®X, background subtracted) were compensated and transformed using FlowJo (v10.10) and IDEAS (v6.2). Data was processed using PICtR (see below). Interacting populations were solely annotated based on mutually exclusive marker expression, since forward scatter properties are not acquired by ImageStream®X. For conventional gating, gates were selected in FlowJo according to the strategy shown in Supplementary Figure 10G.

| | |
|---|---|
| Instrument | For flow cytometric analysis, a Cytek Aurora (Cytek Biosciences) or LSR Fortessa (BD) equipped with 5 lasers were used. For sorting of naive T cells in ex vivo setups, FACSAria Fusion or FACSAria II sorters equipped with 70 µm nozzles were used. For imaging cytometry, image-enabled cell sorting using the BD CellViewTM Imaging Technology was used. For the ImageStream®X experiment, data was acquired using the ImageStream®X (Cytek) and the Cytek® INSPIRE™ software. |
| Software | Software used for the acquisition and analysis of flow cytometry and image cytometry data is described above in the "Software and Code" section. |
| Cell population abundance | Purity in post sort fractions was not directly determined. Post sort cytometry data gave detailed insights into the biology of sorted cell populations. |
| Gating strategy | FSC-SSC gates were set so that FSC-low and SSC-high events were excluded (cell gate). If applicable, dead cells were removed by gating on cells low in viability dyes (Anti-Live Dead blue, efluor506, or Zombie NIR). |

☒ Tick this box to confirm that a figure exemplifying the gating strategy is provided in the Supplementary Information.

