## [Peer Review File · Nature Methods]

Ultra-high scale cytometry-based cellular interaction mapping

Corresponding Author: Professor Simon Haas

Version 0:

Decision Letter:

6th May 2024

Dear Simon,

Your Article, "Ultra-high scale cytometry-based cellular interaction mapping", has now been seen by 3 reviewers. As you will see from their comments below, although the reviewers find your work of considerable potential interest, they have raised a number of concerns. We are interested in the possibility of publishing your paper in Nature Methods, but would like to consider your response to these concerns before we reach a final decision on publication. We therefore invite you to revise your manuscript to fully address these concerns.

Generally speaking, we thought the concerns raised by the referees were reasonable. While revising, we ask that you focus on benchmarking, validating that these interactions are real cell-cell interactions and that the data biologically meaningful and also testing the sensitivity of the method. Please make sure the code is accessible and easily followed.

I am sure you understand that we cannot promise to send the paper back to reviewers until we've seen the new data.

Link Redacted

We hope to receive your revised paper within 8 weeks. If you cannot send it within this time, please let us know. In this event, we will still be happy to reconsider your paper at a later date so long as nothing similar has been accepted for publication at Nature Methods or published elsewhere.

OPEN SCIENCE REQUIREMENTS

REPORTING SUMMARY AND EDITORIAL POLICY CHECKLISTS

DATA AVAILABILITY

All novel DNA and RNA sequencing data, protein sequences, genetic polymorphisms, linked genotype and phenotype data, gene expression data, macromolecular structures, and proteomics data must be deposited in a publicly accessible database, and accession codes and associated hyperlinks must be provided in the "Data Availability" section.

CODE AVAILABILITY

Please include a "Code Availability" subsection in the Online Methods which details how your custom code is made available. Only in rare cases (where code is not central to the main conclusions of the paper) is the statement "available upon request" allowed (and reasons should be specified).

MATERIALS AVAILABILITY

SUPPLEMENTARY PROTOCOL

To help facilitate reproducibility and uptake of your method, we ask you to prepare a step-by-step Supplementary Protocol for the method described in this paper. We [encourage authors to share their step-by-step experimental protocols](https://www.nature.com/nature-research/editorial-policies/reporting-standards#protocols) on a protocol sharing platform of their choice and report the protocol DOI in the reference list. Nature Portfolio's Protocol Exchange is a free-to-use and open resource for protocols; protocols deposited in Protocol Exchange are citable and can be linked from the published article. More details can be found at www.nature.com/protocolexchange/about.

ORCID

Sincerely,
Madhura

Madhura Mukhopadhyay, PhD
Senior Editor
Nature Methods

Reviewers' Comments:

Reviewer #1:

Remarks to the Author:

In the manuscript, Vonficht et al describe a method to use standard cytometry to map changes in interaction between cell types. They apply the method to different tasks for studying the immune system - an artificially induced interaction using an engineered antibody, a murine CAR-T cell activity model and an ex-vivo experiment using a therapeutic antibody that crosslinks leukemic cells and T-cells, and viral infection in a mouse model. I am not an immunologist, but from my horizon the experiments are impressive and could potentially be developed further into individual projects, if proper validation experiments and further experimentation to understand mechanistic details were included. The biology here is not my area of expertise, and I leave this to the other reviewers.

However, the methodological innovation is somewhat less clear to me. The method consists of 1. Cytometry of cells stained with cell-type specific markers. 2. Dimensionality reduction based on all measured parameters, followed by classification to create clusters of cells (each hopefully representing either a single cell type or a specific combination of cells). 3. Supervised classification of each cluster to establish which cell type or combination of cell types constitutes it. Duplet detection is still done based on FSC area/width, but data points in a cluster "votes" for classifying the whole cluster rather than just affecting their own classification into single or doublet.

Using double/multi-positive cytometry events to identify specific interactions is not new. The novel step of this method is point 2: an initial classification of cells in a pre-processing step, to enable downstream work to be done on the level of clusters instead of single events (singlets/multiplets). Therefore, a lot is riding on the assumption that this step is important for the quality of inferences, and that it is robust enough to be used broadly.

The major issue is that the methodological choices are not well motivated, alternatives not explored and the limits/drawbacks of the method not quantified. This is the major issue precluding publication, and the authors have to significantly strengthen this part of the manuscript. Below are some specific points that have to be addressed. However, it is not an exhaustive list: the manuscript would greatly benefit from a more thorough treatment of the methodology in general.

The authors start with a simple experiment where they obtain a microscopy-based ground-truth of the number of cells along with the other cytometric measurements. Based on this, they do three analyses: 1. Split the data into singlets/doublets based

on “FSC ratio”, where the threshold is determined based on minimizing variance in the two groups (“Otsu’s thresholding”). 2. Classify cells in singlets and doublets using a random forest classifier. 3. Leiden-classification based on all marker data. Per-cell thresholded FSC ratio is compared to per-cluster classification and since the latter has better precision/recall the authors use this approach.

Other options need to be explored. For example, how well does a random forest classifier trained on the FSC/SSC parameters perform? Or some other ML approach trained on the non-marker measurements? Using markers to identify doublet clusters and then use the same markers to determine which combination interacted could theoretically create artifacts where a double-positive cell population is confused with doublets of two single-positives. This cannot happen if only signal shape is used to determine doublets. Due to the large amounts of data involved, I am not convinced that optimizing precision/recall is critical here, and that avoiding potential systematic errors might be more important. This area needs to be fully explored.

How sensitive is the method to systematic technical artifacts? The problem is mentioned, but needs to be quantified. This should be done with a fully analyzed experiment to determine if actual inferences are materially different. Could possibly be done with simulated data, although a real experiment would be much more convincing.

How well does the proposed FSC thresholding work compared to alternative strategies? For conventional cytometry, one would create a “singlet” and a “doublet” box/ellipse/polygon gate rather than a single threshold, and typically double-gate based on both FSC-H vs FSC-W and FSH-area vs FSC-W. Such an approach removes borderline doublet events from the analysis. How does this conventional approach compare to an unsupervised threshold?

How well does the method perform when the markers do not perfectly mark all the cell types in the data? For exploratory research this will often be the case - especially if non-immune cells are analyzed, since specific markers are often not available. Experiments can be done *in silico* based on existing experiments, by removing the data for progressively more markers and determining if results are corrupted.

Clustering is such a critical part of this method that better guidance is needed. The potential for issues is well demonstrated when comparing Fig. S1A, which shows the 14 unsupervised clusters to Fig 1D, which shows the expert-curated 5 clusters. The only clusters that remain untouched are the three doublet/triplet clusters. How were the parameters for the initial clustering selected? Did you already know the expected doublet clusters and adjusted based on this?

“Singlet” clusters seem to often contain around 10% doublets (by FSC), considerably higher than I would expect if they were homotypic interactions (See Fig. S5D). Also in the ground-truth experiment Fig 1E, there seems to be ~5-10% of singlets in the doublet classes. In Fig S1B it does not look like there is a threshold that separates the two populations, but rather a continuum. Are these issues experiment-specific? Is it a side-effect of the sketching procedure (i.e selecting rare events)?

Minor issues

Line 38: “Saliva”, not “Salvia”

Line 41-43: “These technologies mostly focus on interaction of pre-defined cell types...”. It is not relevant what the majority of methods do; both Andrews and Boisset papers allow analysis of non-pre-defined cell types and are cited here. The difference in throughput and price is a major advantage, and of course highly relevant. However, these differences are similar to differences in price between any mRNA and cytometry based method, and the drawbacks are also the same: cytometry-based methods operate only on cell types with known markers whereas mRNA-based methods are hypothesis-free. It is not appropriate to compare such vastly different methods mainly on price per cell.

Fig 1H: How many data points constitute the boxplots? Boxplots show a distribution and are only suitable if the number of observations is high. Please indicate $n=$ and plot individual points instead of boxplots if n is low.

Fig 2D: This figure is confusing - what does the color shading mean? What does the thickness mean? It looks like the left/right side were one plot that was later separated, which makes the plot look a little bit awkward (lines originate at random positions within fields) - maybe better to keep them together and indicate +/-CytoStim with color?

Fig 3E: Here n is clearly too low for drawing boxplots. Please just show the individual observations instead.

Fig 6F-I: Same thing, no boxplots with low n .

Fig S4 J,K: Same issue with boxplots.

Fig S1 F and G. These figures are meant to show that cell concentration does not have a strong effect on doublet formation in PBMCs, but the ranges are too different - the lowest spleen concentration is higher than the highest PBMC concentration.

Line 710-718: Should be rewritten to precisely describe the algorithm, or if this is not possible removed and simply replaced by the reference (Hao et al?). Currently it just reads as word-salad and does not aid understanding.

Line 721-722: What is “the maximum number of dimensions in the data”? Do you mean the actual dimensionality of the dataset (eg. the number of measured values per cytometry event? In that case reducing this to $n-1$ seems strangely redundant - the data is (almost) the same size, so why the dimensionality reduction? How have different n been evaluated to arrive at $n-1$ as the optimum?

Line 726-: Clustering is clearly critical to obtaining accurate results using this method, so some guidelines would be in order here.

Line 735-737: This cannot be universally true. Each cluster has a fraction of cells with FSC ratio $>$ threshold, and the top 15-20% of the clusters are classified as doublets. But in Fig. 1 three out of seven clusters are doublet/triplet clusters, which is $>40\%$. Even if you mean the initial 14 clusters, it amounts to $>20\%$ of the clusters. This part of the method has to be better described!

Line 749-751: This section (“Annotation of physically interacting cells”) describes a different method for determining interacting cell clusters “The key criterion for identification of interacting cell clusters was a high FSC-ratio and the presence of more than

one cell type-specific marker." Which one is correct? Or has different heuristics been applied to different parts of the manuscript?

Reviewer #2:

Remarks to the Author:

The paper proposes a new method for detecting and studying physical cell-cell interactions using flow cytometry. The task is very important and there's great interest and progress recently in developing methods for solving it. The proposed method has potential advantages as compared with existing methods, e.g. in cell throughput, potential ease of use, and applicability e.g. to human samples because no genetic engineering is required. The method relies on using parameters measured in flow cytometry that are demonstrated to work well in discriminating single cells from cell doublets and multiplets. The results of application of the method in several contexts are reasonable and actually very interesting, especially in analysis of immunotherapy response. The paper is well written. The software can be run and the results can be reproduced (but see below). However, I have questions about robustness, precision and recall of the approach with respect to true physiological cell interactions in vivo, which may affect applicability and interpretability of the method.

More specific comments:

- While I am not an expert in technical aspects of flow cytometry, it is convincing that the method can detect and report some cell interactions. However, it is unclear how physiologically relevant these detected cell interactions are, and how representative they are of all interactions in vivo. How many and which of the detected cell interactions form as a result of sample manipulations? In principle some cells (e.g. DCs and T cells) may be "sticky" and may form interactions in a suspension as a result or byproduct of some technical manipulations with samples, but not in vivo. But does it really happen, and if yes, does it happen in a biased manner, more so for some cell types and cell states than others? Also, which of the cell interactions happening in vivo in physiological contexts are actually preserved in all sample preparation and sample handling and then detected with this method? Is it for the strongest interactions? or only interactions of some specific types? or some random ones?

The method is interesting, new and worthy, and some limitations are OK and expected but then should be stated and acknowledged clearly. The above questions may be hard to address fully, but should probably be addressed at least somewhat. E.g. how the results of the method will change after varying some parameters of sample preparation or tissue dissociation? Or some additional experimental manipulations with the sample, to varying degree, in the attempt to break more of the cell-cell interactions, or to create more of artificial ones?

The CytoStim or the antigen experiments address these questions but I think only very partially.

- A related but different question is which of the detected cell-cell interactions are functional and which are just by random chance because the cells were next to each other even if in vivo (e.g. in a tissue)?

- The core of the method is application of their software PICtR to compensated and transformed FACS data. It is unclear if the software can detect physical cell interactions in any such previously published data from other experiments and other labs (if yes, it should be demonstrated), or some specific experimental details are important to make it work successfully (then these details should be clearly stated).

- In the viral infection experiment, why aren't T cell-DC interactions found? Is it possible that the flow panel doesn't allow to distinguish between DCs and monocytes and macrophages? Also, I can see cDC2 annotation in Fig 5B, but not cDC1, though cDC1 are mentioned in Fig 5C.

- Not sure if reporting relative frequencies of cell interactions (e.g. in Fig 5C) is relevant. Shouldn't it be normalized by frequencies of individual cell types which are probably also changing between time points?

- The exact Zenodo link to the software provided in the submission actually leads to an older version of the software, v2 dated Feb 2014. We tried to run this but it is not well documented and returns an error. However, there's v5 from Feb 27, 2024, which we tried to use to reproduce some analysis in the manuscript, specifically the results for LCMV experiment. It worked quite well. However, it was unclear how exactly subsampling (or sketching?) works, because following the instructions to the best of our ability resulted in 5000 cell analysis, but Fig 5 reports much larger numbers of cells. This Zenodo link to v5 includes the file PICtR-manual.pdf, which is indeed a manual of the R package. But it will be helpful to include some examples of running the analysis to reproduce at least some figures in the paper exactly, and walk the users through all steps (a notebook and step-by-step instructions, or what sometimes is called "vignette" for R packages).

Reviewer #3:

Remarks to the Author:

In this study, the authors developed Interactomics, which is a cytometry-based workflow that can detect multiplet interactions. In principle, it does fill an open niche in the investigation of immune cell interactions in high-dimensional space with high throughput. However, it is not deployed to its full potential in this manuscript at present. One could imagine performing a lower throughput or more time consuming approach such as PIC-seq or LIPSTIC that yields more information than using Interactomics. I find several key weaknesses with the manuscript at present: 1) A lack of mechanistic depth in any conclusions

that are generated; 2) Concerns about the ab initio classification of the interactions detectable by the method; and 3) The long-term utility and flexibility of the method to detect more specific (and probably more valuable) interactions among more narrowly defined cell subsets. The strengths of the study are the authors' development of a creative platform that could be widely adopted, demonstration of proof-of-concept data across multiple mouse and human systems, and well-designed studies across time and stimulation. In general, if the authors can address the above concerns, I believe it could be a valuable tool. Additional points are listed below:

1. The advantages over pre-existing approaches to study immune cell interactions are not clear. For example, "Interactomics" is entirely descriptive, while LIPSTIC and PIC-seq gain mechanistic information. Investigation and validation of the findings by Interactomics is needed in mouse and human systems.
2. Additional validation of the functional implications of the interactions detected is needed on the level of intracellular signaling. Even if cell types are interacting, no mechanistic data are included in the manuscript to define that the interactions are controlling cellular physiology using this method. Many findings reported in the manuscript could be discerned using other approaches.
3. One main limitation of the study is the focus on CytoStim as an inducer of "ground truth" cellular interactions. This approach would primarily label memory lymphocytes and APCs. The argument for developing this method is that it is more high throughput and generalizable than current approaches, but the training dataset undermines this. Additional "ground truth" interaction-driving approaches should be incorporated into the study and tested. Ideally, multiple antibody panels/stimuli that would detect distinct classes of interactions, compiled into a PICtR database, would make the method more robust.
4. The CytoStim experiments would also induce artificial interactions between cell types, which may not reflect true physiology. A trained dataset focused on physiologic interactions during immunization, or equivalent, might make a better training dataset and be more comprehensive.
5. The algorithm was trained on human data, which was then applied to the mouse. How generalizable are interactions in the two systems to one another?
6. Almost all experiments are performed in vitro or from PBMCs, so while the throughput is increased, the relevance is somewhat diminished. To showcase the power of this tool, the authors could investigate additional fluids not tractable with other tools, and glean new insights not previously known, as they mentioned in the introduction.
7. An important feature of this approach is to delineate specific molecular insights that are not possible with conventional flow cytometry. However, the authors only capitalized on broad cell lineage markers. How well are interactions actually detected and validated when a panel of markers is used that is focused on subsets of intra-lineage cells? For example, can Interactomics be applied to CD4 Th cell subset differentiation?
8. The inability to detect homotypic interactions is a weakness. If different starting panels of antibodies were used to train the parameters of the system, would different interactions be prioritized as detected in test datasets? For example, if the 24-plex panels contained much more specific markers for B cell subsets, could these interactions be detected?
9. Almost all findings are descriptive. If only a fraction of interactions are detectable by Interactomics, and there is no in vivo validation of the method, it is hard to take much new information from the study. At least one substantial finding detected by Interactomics should be validated in vivo (or in a related human system).
10. Can PICtR reevaluate cellular interactions in previously reported datasets? It would be valuable to mine new insights from previous datasets in an agnostic way with orthogonal antibody panels.
11. Even if multiplets are detected in flow cytometry data, it is not clear how they are differentiated as "real" interactions and not simply noise, for example, cells clumping within a tube for technical reasons. Controls guarding against the possibilities of technical contamination would help enrich for meaningful signals.

Minor points:

1. Biological replicates are not mentioned in at least some figure legends.
2. Fig. 1E it is not clear how a My*^T*B interaction (3 cells) could be represented by a doublet or singlet. Same with other interactions in that panel. It should be shown more clearly.
3. In Fig. 3K there is not information learned, since the bifunctional antibody simply induces interactions mostly non-specifically.
4. In Fig. 5E, the distance in PCA space analysis is unusual. A different metric should be used.
5. The text formatting reads like a single long paragraph and would be more readable if broken into demarcated sections.
6. The name of the method "Interactomics" should be changed as many methods at this point perform interactomics. A more

specific name should be used.

Version 1:

Decision Letter:

Our ref: NMETH-A55672A

25th Nov 2024

Dear Simon,

Thank you for submitting your revised manuscript "Ultra-high scale cytometry-based cellular interaction mapping" (NMETH-A55672A). It has now been seen by the original referees and their comments are below. The reviewers find that the paper has improved in revision, and therefore we'll be happy in principle to publish it in Nature Methods, pending minor revisions to satisfy the referees' final requests and to comply with our editorial and formatting guidelines.

Thanks for sending us a quick response to our queries on the review. We think it would really help support the broad applicability of the method if you could add an example on epithelial cells as demonstration. Also please make sure to discuss limitations in the discussion.

TRANSPARENT PEER REVIEW

ORCID

Sincerely,
Madhura

Madhura Mukhopadhyay, PhD
Senior Editor
Nature Methods

Reviewer #1 (Remarks to the Author):

The revised manuscript is much improved, and I can see that the authors have made an honest effort to address my specific concerns. They have added several new experiments addressing quality control issues, and improved figures and text throughout.

My only remaining concerns are about novelty and generality

1. As I wrote in my original comments, counting co-stained doublets is not novel. Previous mRNA-based doublet methods derived their claim to originality on using the highly sensitive transcriptional profile to find new states and boost sensitivity. This method introduces an automated clustering step to the co-staining protocol - is this novel enough to merit publication in Nature Methods?

2. How general is the method? Immune cells are incredibly well defined by their cell surface markers - would you expect this method to be largely domain specific? (ie Rev. 3, point 6). If you wanted to study epithelial cell interactions in brain, or lung, or skin, some tumor, etc, would this method be of any use?

I am not ultimately convinced of either of these points, but it is really a question of degrees rather than absolutes. I will simply defer to the editor to judge how the novelty/generality issues should impact publishing in this case.

Reviewer #3 (Remarks to the Author):

In revision, the authors have undertaken a comprehensive set of experiments and clarifications, which address my comments. In particular, there are now numerous technical controls as well as a limitations section, which advises future users on parameters to optimize. I'm also more convinced the approach is fairly robust to noise. There are also important additions demonstrating that previously acquired data can be analyzed by the approach.

I also believe the approach will have broad utility because of its flexibility and accessibility, which is actually unique among all interaction-based approaches, which tend to be complicated.

One lingering minor point left to the discretion of the editor and authors is the name, as "Interactomics" I think isn't descriptive enough for the method, which doesn't assess all interactions and is neither the first nor only method to study interactions. Could it be called PICtR?

I otherwise have no additional points to raise.

Version 2:

Decision Letter:

6th May 2025

Dear Simon,

I am pleased to inform you that your Article, "Ultra-high scale cytometry-based cellular interaction mapping", has now been accepted for publication in Nature Methods. The received and accepted dates will be 7 Mar, 2024 and 6 May, 2025. This note is intended to let you know what to expect from us over the next month or so, and to let you know where to address any further questions.

Over the next few weeks, your paper will be copyedited to ensure that it conforms to Nature Methods style. Once your paper is typeset, you will receive an email with a link to choose the appropriate publishing options for your paper and our Author Services team will be in touch regarding any additional information that may be required. It is extremely important that you let us know now whether you will be difficult to contact over the next month. If this is the case, we ask that you send us the contact information (email, phone and fax) of someone who will be able to check the proofs and deal with any last-minute problems.

If you are active on Twitter/X or Bluesky, please e-mail me your and your coauthors' handles so that we may tag you when the paper is published.

Best regards,
Madhura

Madhura Mukhopadhyay, PhD
Senior Editor
Nature Methods

** Visit the Springer Nature Editorial and Publishing website at http://editorial-jobs.springernature.com?utm_source=ejP_NMeth_email&utm_medium=ejP_NMeth_email&utm_campaign=ejp_Nmeth for more information about our career opportunities. If you have any questions please click [here](mailto:editorial.publishing.jobs@springernature.com).**

Open Access This Peer Review File is licensed under a Creative Commons Attribution 4.0 International License, which permits use, sharing, adaptation, distribution and reproduction in any medium or format, as long as you give appropriate credit to the original author(s) and the source, provide a link to the Creative Commons license, and indicate if changes were made. In cases where reviewers are anonymous, credit should be given to 'Anonymous Referee' and the source. The images or other third party material in this Peer Review File are included in the article's Creative Commons license, unless indicated otherwise in a credit line to the material. If material is not included in the article's Creative Commons license and your intended use is not permitted by statutory regulation or exceeds the permitted use, you will need to obtain permission directly from the copyright holder.

First point-by-point reply

Reviewer #1:

Remarks to the Author:

In the manuscript, Vonficht et al describe a method to use standard cytometry to map changes in interaction between cell types. They apply the method to different tasks for studying the immune system - an artificially induced interaction using an engineered antibody, a murine CAR-T cell activity model and an ex-vivo experiment using a therapeutic antibody that crosslinks leukemic cells and T-cells, and viral infection in a mouse model. I am not an immunologist, but from my horizon the experiments are impressive and could potentially be developed further into individual projects, if proper validation experiments and further experimentation to understand mechanistic details were included. The biology here is not my area of expertise, and I leave this to the other reviewers.

However, the methodological innovation is somewhat less clear to me. The method consists of 1. Cytometry of cells stained with cell-type specific markers. 2. Dimensionality reduction based on all measured parameters, followed by classification to create clusters of cells (each hopefully representing either a single cell type or a specific combination of cells). 3. Supervised classification of each cluster to establish which cell type or combination of cell types constitutes it. Duplet detection is still done based on FSC area/width, but data points in a cluster “votes” for classifying the whole cluster rather than just affecting their own classification into single or doublet.

Using double/multi-positive cytometry events to identify specific interactions is not new. The novel step of this method is point 2: an initial classification of cells in a pre-processing step, to enable downstream work to be done on the level of clusters instead of single events (singlets/multiplets). Therefore, a lot is riding on the assumption that this step is important for the quality of inferences, and that it is robust enough to be used broadly.

We thank the reviewer for recognizing the utility of our approach and the comprehensive nature of our analysis. We also appreciate the reviewer’s critical evaluation, which has been invaluable in significantly enhancing the quality of our manuscript. Below, we have thoroughly addressed all of the reviewer’s comments, including those mentioned above.

The major issue is that the methodological choices are not well motivated, alternatives not explored and the limits/drawbacks of the method not quantified. This is the major issue precluding publication, and the authors have to significantly strengthen this part of the manuscript. Below are some specific points that have to be addressed. However, it is not an exhaustive list: the manuscript would greatly benefit from a more thorough treatment of the methodology in general.

We thank the reviewer for the critical assessment, and fully agree that this aspect should be strengthened. In order to motivate methodological choices, we present a multi-step quantitative benchmarking procedure:

1. As already described in the original manuscript, a feature importance analysis yielded the FSC ratio to be most discriminative between singlets and doublets.
2. In the framework of the revision, we tested different thresholding and classification methods on the FSC ratio and identified several approaches, including Otsu thresholding, to have similar accuracy and to be comparable to manual gating, with the advantage of being reproducible and data-driven.
3. To further improve classification and identification of interacting cells, we explored clustering-based approaches for simultaneous multiplet discrimination and annotation. Clustering based on cell type markers and scatter properties – including the FSC ratio – but without image-based features yielded results comparable to those achieved when including image-based features.
4. Having identified a set of flow cytometry parameters that is effective at identifying singlet and multiplet events, we tested several community detection algorithms using these features as input. Methods from the single-cell genomics field consistently outperformed methods stemming from the flow cytometry field. In the updated version of our PICtR software package, we implement several of the well performing algorithms, leaving a customizable choice to expert users.

Since the first step was already included in our initial manuscript, the following sections provide a detailed description of the remaining three benchmarking steps:

Benchmarking of thresholding and classification methods based on the FSC ratio. We first tested a range of different classification thresholds for singlet vs. multiplet discrimination based on the FSC ratio only, which was identified as the most important single feature in the feature importance analysis (see Figure 1B of the manuscript). We evaluated several approaches on the distribution of the FSC ratio, as often applied in image thresholding tasks (Otsu, IsoData, Intermodos, RenyiEntropy, Triangle, Li, Shanbhag, Huang, Mean). In addition, we employed kmeans clustering with $k = 2$ and a finite Gaussian mixture model (GMM) with $G = 2$ as classification methods. Finally, we included manual gating of singlet/multiplet events in our comparison, as traditionally used in the flow cytometry field. All approaches (listed in Table R1) were tested across 4 technical replicates for which we obtained the ground truth classification via manual annotation of the respective images ($n = 3865$ in total). Several approaches, including Otsu thresholding, were comparable to manual gating, with the advantage of being reproducible and data-driven (Fig. R1A).

Table R1: Thresholding methods evaluated for the FSC ratio.

Method	Short Description, information on factorization rank wherever appropriate	Ref.
Manual Gating	Manual gating strategy based on scatter parameters from $n = 3$ analysts	-
Gaussian Mixture Model	Finite Gaussian mixture model on the FSC ratio, $G = 2$	Fraley and Raftery, 2011, Journal of the American Statistical Association; Fraley and Raftery, 2007, Journal of Classification
Huang	Fuzzy set theory to partition data into meaningful regions, using Shannon's entropy function	Huang and Wang, 1995, Pattern recognition
Intermodos	Iterative smoothing of a bimodal histogram until only two local maxima remain; the threshold is given as the mean between the two	Prewitt and Mendelsohn, 1966, Annals of the New York Academy of Sciences

IsoData	Iterative thresholding evaluated by the composite average of the data below and above the threshold	Ridler and Calvard, 1978, IEEE Transactions on Systems, Man and Cybernetics
kmeans	Minimization of the distance between data points and their assigned cluster's center, $k = 2$	Hartigan and Wong, 1979, Journal of the Royal Statistical Society
Li	Minimization of the cross-entropy between segmented and unsegmented data	Li and Tam, 1998, Pattern Recognition Letters
Mean	Selecting the mean of the data as the threshold	Glasbey, 1993, CVGIP: Graphical Models and Image Processing
Otsu	Minimization of intra-group variance	Otsu, 1979, IEEE Transactions on Systems, Man and Cybernetics
Renyi Entropy	Maximization of the information between distributions using Renyi's entropy method	Kapur, Sahoo and Wong, 1985, Computer Vision, Graphics, and Image Processing
Shanbhag	Modification of the Renyi/maximum entropy method	Shanbhag, 1994, CVGIP: Graphical Models and Image Processing
Triangle	For a line from the histogram's peak to its farthest extreme; the threshold is the maximum distance of the line to the histogram	Zack, Rogers, and Latt, 1977, Journal of Histochemistry & Cytochemistry

Comparison between scatter-based thresholding and clustering-based approaches for singlet vs multiplet discrimination. To systematically compare the thresholding or gating methods based on the scatter alone to clustering approaches, we performed a series of analyses. First, we employed Louvain clustering on cell type markers only, followed by classifying clusters into singlets and multiplets based on FSC ratio and subsequent annotation based on co-expression of mutually exclusive cell type markers. This cluster-based approach considerably outperformed classifying cells directly based on FSC-thresholding alone (Fig. R1B, comparison first and second bar). Notably, incorporating both cell type markers and scatter properties – including the FSC ratio – into the initial clustering, followed by classifying clusters based on the FSC ratio and subsequent annotation based on co-expression of mutually exclusive cell type markers, further improved the classification (Fig. R1B, third bar). Importantly, this approach yielded results comparable to those achieved when all important features including image-based features were utilized, therefore serving as an optional approach to map physically interacting cells with conventional cytometry with no ground truth data available (Fig. R1B, comparison third and fourth bar). We have now included these analyses in the revised version of our manuscript.

Figure R1. Benchmarking of cytometry-based cellular interaction mapping.

A. Performance of different classification methods on the FSC ratio as measured by the F1 score. Manual image annotation served as the ground truth; see Methods for details. $n = 4$ replicates; bars indicate the mean F1 score. **B.** Performance of different classification methods as measured by the F1 score. Cells were classified by Otsu thresholding of the FSC ratio (dark blue), or by Louvain clustering of the indicated parameters followed by cluster-based classification driven by the proportion of cells exceeding the FSC ratio threshold (light blue). Louvain clustering was performed for $n = 100$ iterations, and for $n = 4$ replicates. Bars indicate the mean F1 score. **C.** Performance of different clustering methods regarding their ability to resolve singlet and interacting populations. All algorithms were used for $n = 100$ iterations on conventional flow parameters including forward scatter parameters, side scatter parameters, cell type markers and the FSC ratio, see Methods for details. $n = 4$ replicates; bars indicate the mean F1 score.

Benchmarking of clustering approaches for singlet vs multiplet discrimination. Having identified a general approach that employs conventional flow cytometry parameters effective at identifying singlet and multiplet events, we tested several community detection algorithms using these features as input. Candidates were selected based on their popularity in the single-cell genomics and flow cytometry fields or based on their performance on high-dimensional flow and mass cytometry data as evaluated by Weber and Robinson (Weber and Robinson, 2016, Cytometry Part A). Each method was run for $n = 100$ iterations and the performance was reported as F1 scores based on the ground-truth classification (Fig. R1C). Methods from the single-cell genomics field consistently outperformed methods from the flow cytometry field, such as FlowSOM or FlowMeans. Depending on the data set, some algorithms might be more appropriate than others. We have thus implemented several alternatives in our package (listed in Table R2, which has also been incorporated in the methods section of our revised manuscript), leaving a possible customization option for expert users.

The remaining parts of our analysis pipeline are based on established single-cell workflows as implemented in the Seurat (Hao et al., 2024, Nature Biotechnology) framework.

Table R2: Methods evaluated for distinguishing singlet and interacting communities.

Method	Short Description	Ref.
FlowMeans	Based on kmeans clustering, finds non-spherical clusters. Max. 10 iterations, standard Mahalanobis distance and no standardization	Aghaeepour et al., 2010, Cytometry Part A
FlowSOM	Two-level clustering using self-organizing maps. meta k = 20	Van Gassen et al., 2015, Cytometry Part A
HDBSCAN	Hierarchical clustering algorithm based on density. Used with a minimum cluster size of 100	Campello et al., 2015, ACM Trans. Knowl. Discov. Data
Immunoclust	Iterative clustering using finite mixture models, expectation maximization and integrated classification likelihood	Sørensen et al., 2015, Cytometry Part A
Leiden	Modification of the Louvain algorithm. Used with resolution = 1	Traag et al., 2019, Scientific Reports
Louvain	Optimizes modularity by iteratively assigning nodes of a network to clusters so that links within communities outweigh links between communities. Used with resolution = 1	Blondel et al., 2008, J. Stat. Mech.
Phenograph	Constructs a graph based on phenotypic similarity before clustering. Used with $k = 10$ and Louvain or Leiden clustering (resolution = 1, 10 iterations)	Levine et al., 2015, Cell
Rclusterpp	Hierarchical clustering using Ward's method and Euclidean distances, $k = 20$ for cutting the tree	Linderman et al., 2022, Rclusterpp: Linkable C++ Clustering. R package version 0.2.6,

The authors start with a simple experiment where they obtain a microscopy-based ground-truth of the number of cells along with the other cytometric measurements. Based on this, they do three analyses: 1. Split the data into singlets/doublets based on “FSC ratio”, where the threshold is determined based on minimizing variance in the two groups (“Otsu’s thresholding”). 2. Classify cells in singlets and doublets using a random forest classifier. 3. Leiden-classification based on all marker data. Per-cell thresholded FSC ratio is compared to per-cluster classification and since the latter has better precision/recall the authors use this approach.

Other options need to be explored. For example, how well does a random forest classifier trained on the FSC/SSC parameters perform? Or some other ML approach trained on the non-marker measurements? Using markers to identify doublet clusters and then use the same markers to determine which combination interacted could theoretically create artifacts where a double-positive cell population is confused with doublets of two single-positives. This cannot happen if only signal shape is used to determine doublets. Due to the large amounts of data involved, I am not convinced that optimizing precision/recall is critical here, and that avoiding potential systematic errors might be more important. This area needs to be fully explored.

We thank the reviewer for the in-depth analysis and are happy to comment. We aimed to develop a framework that is applicable to conventional flow cytometry data. Supervised

approaches such as a random forest classifier can successfully be trained on image-enabled flow cytometry datasets, using the manual annotation of the images as labels for the data. However, these labels are not available in conventional flow cytometry, and therefore we employed unsupervised approaches such as Louvain community detection in our framework and used the information gained from the image-enabled flow cytometry dataset only as ground truth data to evaluate our pipeline.

For the ground truth experiment in Figure 1 for which labeled data is available, we performed a comparison between various supervised machine learning algorithms and the *Interact-omics* approach, as well as a FSC ratio thresholding only (Fig. R2). Supervised models were trained on non-marker measurements with the manual annotation of the images as labels. As seen in Figure R2, the best performing supervised methods are comparable or inferior to FSC ratio thresholding only, and are outperformed by the *Interact-omics* approach, which combines Louvain clustering on all conventional flow parameters followed by the cluster-based classification into singlets and multiplets based on the FSC-ratio.

Figure R2. Comparison of supervised machine learning approaches vs. *Interact-omics*. Performance of different classification methods as measured by the F1 score. Linear Discriminant Analysis (LDA, Ripley, 1996, Pattern Recognition and Neural Networks), a Naive Bayes classifier, a classification tree (Breiman et al., 1984, Classification and Regression Trees) and random forest classifier (Breiman, 2001, Machine Learning) were trained on the scatter parameters (forward and side scatters, FSC ratio) with labels from the manual ground truth annotation of the experiment in Figure 1. To the right, results for FSC ratio thresholding only (Otsu) and the entire *Interact-omics* pipeline (from Fig. 1E) are shown for comparison. Results for each replicate ($n = 4$) are shown in the scatter plot and bars indicate the mean F1 score.

Confusion of double-positive cell populations and doublets of two single-positives can be ruled out based on the following two-step approach in *Interact-omics*: (i) Unsupervised clustering is performed on all conventional flow parameters (forward scatter, side scatter, available cell type markers and the FSC ratio) and each event is preliminarily classified as having a low or high FSC ratio (e.g. via Otsu thresholding). Then, (ii) the number of events with a high FSC ratio within each cluster decides whether the cluster is classified as a singlet or interacting cell cluster. Only if a cluster is classified as an interacting cell cluster through this process *and*

contains more than one mutually-exclusive cell type marker, it is annotated as an interacting population between the given interaction partners. Therefore, a population of singlets that express both markers would not be classified as an interacting population a priori. Of note, careful design of the used flow cytometry panels contributes to discriminatory power. Moreover, we have provided additional experimental benchmarking in response to the reviewer's next question.

How sensitive is the method to systematic technical artifacts? The problem is mentioned, but needs to be quantified. This should be done with a fully analyzed experiment to determine if actual inferences are materially different. Could possibly be done with simulated data, although a real experiment would be much more convincing.

We thank the reviewer for raising this highly relevant question. The proposed method quantifies single-cell landscapes, as well as the frequencies and types of interactions among physically interacting cells in a given sample at the time of analysis. This provides a snapshot of current interactions, which may include both random technical interactions and specific interaction that reflect biological effects. When analyzing interactions derived from an *in vivo* setting, new interactions might be acquired during sample preparation. These are inherent limitations to the growing family of methods that map the current state of physically interacting cells, such as PIC-seq, CIM-seq, SPEAC-seq, Clumb-seq, paired-cell sequencing or imaging cytometry. We fully agree with the reviewer that systematically understanding these limitations is crucial for interpreting the results of the proposed and related methods. Therefore, we have conducted a series of *ex vivo* and *in vivo* benchmarking experiments to thoroughly assess the applicability and limitations of our approach, including analyses of the functionality and specificity of cellular interactions.

1. *Ex vivo* cellular mixing experiment

To assess the extent to which additional cellular interactions are formed during culture and how different experimental settings impact this process, we conducted an extensive *ex vivo* benchmarking experiment involving labeling and mixing of cell populations. We first induced cellular interactions in PBMCs with CytoStim or left cells untreated. The PBMCs were then split and labeled with two different fluorescently conjugated antibodies against CD45 (Fig. R3A). Subsequently, the distinctively labeled PBMCs were reunited and processed at varying cell concentrations, processing times, and fixation methods, followed by mapping cellular interactions using our approach. This allowed us to effectively identify singlets positive for each label (Fig. R3B-C), as well as cellular interactions that were either single- or double-positive for both labels (Fig. R3D-F). Double-positive cellular interactions for the two introduced labels were newly acquired during the second incubation period, while single-positive interactions could have been acquired initially or throughout the culture period. Focusing on B-T cell interactions, as one of the most frequent interactions induced by CytoStim, our experiments revealed that single-positive interactions increased robustly upon CytoStim treatment. In contrast, double-positive (newly acquired) interactions showed only a mild increase, which appeared negligible compared to the induction of single-positive interactions (approximately 5-10-fold higher in single positives compared to double positives) (Fig. R3G). Increasing incubation periods post-CytoStim did not cause a stark increase in newly acquired (double-positive) interactions, while a trend toward a decrease in CytoStim-induced (single-positive) B-T cell interactions was observed (Fig. R3G). Even at 4 hours post-

incubation, CytoStim-induced single-positives were on average 7-fold more frequent than their double-positive counterparts. These data suggest that *ex vivo* induced cellular interactions are relatively stable, and newly acquired interactions occur but have a relatively minor impact in this setting. With increasing cell concentrations, a mild baseline increase across all interactions (both single and double positives, with and without CytoStim) was observed (Fig. R3G). However, the relative effect of CytoStim-induced single-positive interactions compared to newly acquired double-positive interactions remained comparable. These data reinforce the importance of maintaining stable cell concentrations within experimental settings but also demonstrate that *ex vivo* induced cellular interactions can be effectively measured across distinct concentrations, providing stability across conditions.

Finally, we evaluated the impact of chemical fixation on newly formed *ex vivo* interactions. PBMCs were either fixed or left unfixed after +/- CytoStim treatment, labeling, reuniting and staining. Overall, only a minor impact of fixation was observed, with a trend towards higher single and double positive interactions upon CytoStim treatment in fixed samples (Fig. R3G). This experiment suggests that cellular interactions can be retrieved using fixation or leaving samples in their native state, provided the same conditions are applied to all samples. We recommend, however, to fix samples as early as possible during the sample preparation workflow.

Collectively, these data provide quantitative insights into how experimental settings impact background interactions while demonstrating that *ex vivo* modulations of cellular interactions can be effectively quantified in the explored settings. Besides incubation times, cell numbers, and fixation, flow rates significantly impact baseline interactions, as previously demonstrated (Fig. R3H-J). Therefore, maintaining these parameters constant maximizes the recovery of signal to background interactions. We have now included a limitations and guidelines section in the manuscript detailing how to optimize the experimental setting for *ex vivo* cytometry-based cellular interaction mapping.

Figure R3. Effect of sample processing methods on ex vivo cellular interactions.

A. Schematic depiction of the experimental approach. **B.** Overall cellular landscape across all experimental conditions. $n = 3,292,837$. **C.** Feature plots for the UMAP display in B, colored by the two differently labeled antibodies against CD45. **D.** Interacting landscape across all experimental conditions. $n = 23,620$. **E.** Dot plot for the CD45 signals in interacting populations, showcasing single-positive and double-positive interactions. **F.** Feature plots for the UMAP from panel D, colored by the CD45 signal intensities. **G.** Quantification of single-positive and double-positive, newly acquired interactions in CytoStim™ treated and untreated samples, across different experimental conditions. Left: Varying incubation times at 4°C after mixing, mimicking long sample processing times. Middle: Different cellular concentrations ranging from 25,000 to 250,000 cells in 50 μ L during staining and acquisition. Right: Fixation with 2 % paraformaldehyde after staining compared to no fixation. The number of replicates is shown in each plot and ranges from $n = 2$ to 3. **H.** Short-term cultures of PBMCs with or without CytoStim™ measured at low or high flow rate ($n = 4$). Impact of flow rate on cellular interactions is relatively mild. Two-way ANOVA (CytoStim™: $F(1,13) = 189.138$, $P = 4.01e-09$, flow rate: $F(1,13) = 6.598$, $P = 0.023$), followed by Tukey's Honest Significant Differences test for the flowrate. **I.** Impact of flow rate and cellular concentration on cellular interactions in murine

spleens is more pronounced. Left: Baseline interactions in spleens measured with different flow rates ($n = 4$). Right: Baseline interactions in spleens at different cell densities but constant flow rate ($n = 4$). One-way ANOVA (flow rate: $F(2,9) = 115.749$, $P = 3.79e-07$, cell concentration: $F(2,9) = 61.397$, $P = 5.68e-06$), followed by Tukey's Honest Significant Differences tests. J. Boxplots of T*B cell interactions upon Blinatumomab treatment that were either fixed or fixed after a freeze-thaw cycle ($n = 4$). P values were determined with a two-sided Welch's t-test.

2. *In vivo* benchmarking to imaging flow cytometry

To provide additional information on the reliability of interactions assessed in particular *in vivo* settings, we performed ImageStream-based imaging flow cytometry of LCMV-infected spleens at day 7 post-infection, similar to our experimental setup for organism-wide cytometry-based cellular interaction mapping presented in the manuscript (Fig. 5). This experimental approach has some analogy to the introductory experiment described in Figure 1 of manuscript, however, here, we present a dataset which uses a different technology, with more events analyzed and applied to more complex *in vivo* setting (see below). While imaging cytometry requires specialized instruments, has a low cellular throughput than flow cytometry, and is limited in the number of measured fluorochromes and cell types that can be resolved, it provides morphological imaging information that can be used to classify cells into single versus interacting cells. Due to the limited number of available channels, we utilized a 6-plex panel focusing on T and B cell interactions. Upon data generation, we extracted the fluorescence intensity values and applied the PICtR workflow without considering any morphological information (Fig. R4A-C). As the FSC ratio cannot be extracted during the ImageStream workflow, clustering and interaction identification was performed on fluorescence channel values only. Consequently, the results presented here likely underestimate the performance of the *Interact-omics* approach. We then evaluated the performance of the *Interact-omics* approach with information gained from both images and fluorescence values regarding (i) discriminating singlets from interacting cells, (ii) identifying cell type combinations of interactions, and (iii) deriving qualitative changes in LCMV-induced interactions. For this purpose, we first compared the results of the *Interact-omics* workflow to an image segmentation-based classification of singlets and multiplets, demonstrating a high concordance between the approaches regarding the discrimination between single cells and multiplets (Fig. R4D). While neither image-based segmentation nor our approach reflects ground truth data, the high concordance strongly supports the validity of the *Interact-omics* results. Manual inspection of randomly selected images predicted as singlets or interacting cells further verified the accuracy of the *Interact-omics* approach (Fig. R4C).

Next, we immunophenotypically characterized interacting populations using conventional gating on marker expression values extracted from images (Fig. 4RG). As expected, this revealed a high concordance with the cluster-based annotations from the *Interact-omics* workflow. Manual inspection of randomly selected images regarding the localized expression patterns of lineage-specific markers confirmed the expected types of interactions (Fig. R4C). Finally, we compared the LCMV-induced alterations of cellular interactions as detected by *Interact-omics* vs. the imaging flow cytometry approach. Due to the low plexity and low cellular throughput of the imaging flow cytometry approach, only very crude interacting cell populations could be defined, restricting the comparison to the populations depicted in Fig. R4B. Nonetheless, interacting cell populations identified with both approaches showed qualitatively

matching alterations induced by LCMV, including a marked increase in interactions involving antigen-specific CD4 T cells (Fig. R4F).

Collectively, these data confirm the accuracy of the *Interact-omics* approach in discriminating single and interacting cells and in quantifying the types of interactions in case-control settings.

Figure R4. Comparison of the *Interact-omics* approach to imaging flow cytometry.

A. UMAP representation of the overall cellular landscape derived from the fluorescent intensity values, $n = 306538$. Intensity values are based on the sum of the pixel intensities in the mask as selected by ImageStream[®], background subtracted. The experiment corresponds to day 7 in Supplementary Figure 12. **B.** UMAP representation of the interacting landscape, $n = 8683$. The heterogeneous cluster is mostly comprised of likely B*CD4*CD8 multiplets. The unknown cluster expresses CD19 and CD3 but no other T cell markers, hindering confident annotation. **C.** Pseudo-colored example images for cellular interactions in the brightfield and fluorescence channels. **D.** Left: UMAP displays from A and B colored by the number of cells identified through image segmentation. Right: Bar plots comparing the populations identified through *Interact-omics* and the image segmentation. **E.** Left: UMAP displays from A and B colored by

populations as identified through conventional gating. Right: Bar plots comparing the populations identified through *Interact-omics* and conventional gating. NA indicates that the event does not fall into any conventional gate. **F.** Fold changes of the frequencies +/- LCMV infection. Holm-corrected estimated marginal means comparison. Left: Populations identified by *Interact-omics*. Right: Populations identified through conventional gating. **G.** Gating strategy for conventional gating. Abbreviations: Ag = antigen-specific, UMAP = uniform manifold approximation and projection, BF = brightfield.

3. *In vivo* mixing experiments to characterize newly acquired cellular interactions

The presented approach measures cellular interactions following sample preparation *ex vivo*. Consequently, for *in vivo* applications, additional cellular interactions can be acquired or lost during sample preparation. While this limitation applies to all cellular interaction mapping approaches that do not rely on specialized mouse models or measure co-localization *in situ*, it remains poorly characterized to what extent this occurs, whether newly acquired interactions are random or directed, and how representative the identified interactions are of the *in vivo* situation.

To evaluate these questions, we utilized congenic mouse models differing in variants of the pan-hematopoietic cell marker CD45, allowing identification of respective immune cells as CD45.1 or CD45.2 using variant-sensitive antibodies. First, we transferred LCMV-specific CD4 T cells (SMARTA: CD90.1 positive, CD45.2 positive) into CD45.2 mice, followed by LCMV infection (group A) (Fig. R5A). Non-infected control CD45.2 mice were included (group B). In parallel, we infected CD45.1 mice with LCMV (group C) or left them untreated (group D). On day 7 post-infection, spleens from all groups were harvested. Spleens from group A (infected, CD45.2) and group B (non-infected, CD45.2) were either processed individually or mixed in 1:1 ratios with spleens from group C (infected, CD45.1) or group D (non-infected, CD45.1) before tissue homogenization and processing. Applying the *Interact-omics* workflow to these single and mixed samples resulted in single-cell and interacting cell landscapes of populations that were either single positive or double positive for CD45.1 and CD45.2 (Fig. R5B-E), with double-positive populations being newly acquired interactions during processing. Notably, we observed a substantial acquisition of new interactions during sample processing (Fig. R5F). However, these newly acquired interactions did not occur randomly but were highly correlated with interactions induced upon infection (Fig. R5G). In particular, newly acquired interactions in mixed spleens from infected mice compared to non-infected controls were highly correlated with infection-induced single positive interactions in both mixed and non-mixed spleens (Fig. R5G). This suggests that while new interactions can be acquired during sample preparation, they are not random but reflect actual biological effects and likely are a proxy for cellular interactions occurring *in vivo* (see also next section about qualitative comparison to *in situ* methods).

Figure R5. Interactions acquired a posteriori after *in vivo* experiment.

A. Schematic overview of the experimental approach. LCMV-specific CD4⁺ T cells were transferred into CD45.2 host mice 5 days before infection with LCMV (group A) or the respective control (group B). Additionally, CD45.1 host mice were infected with LCMV (group C) or left untreated (group D). $n = 3$ for groups A, C, D and $n = 4$ for group B. **B.** Single-cell landscape of all experimental groups. Out of $n = 23,490,812$ processed cells, $n = 245,316$ are shown in the UMAP display. **C.** Feature plots for panel B, colored by the expression of the congenital markers CD45.1 and CD45.2. **D.** Interacting landscape across all experimental groups. Out of $n = 731621$ identifiable interactions, $n = 93065$ are shown. **E.** Dot plots showcasing the expression of the congenital markers CD45.1 and CD45.2 in interacting populations from the unmixed controls for the untreated and infected conditions, and the mixed spleens from infected mice (infected+infected, group A + group C) or untreated mice (control+control, group B + group D). **F.** Bar plots depicting the log₂ fold changes (FC) between the LCMV infected and untreated conditions for each interacting population. Solid bars indicate the log₂FC between group A (infected) and group B (control). Semi-transparent bars show the log₂FC for single-positive interactions in mixed samples (A+C for the infected condition, and

B+D for the control). Transparent bars depict log₂FC between the respective double-positive interactions, which were definitely acquired *ex vivo*. *n* = 3 mixes. **G.** Linear relationships between the log₂FC between infected and control conditions for unmixed controls, single-positive interactions after mixing and double-positive interactions after mixing. *n* = 3 mixes.

4. Qualitative comparison to *in situ* methods

To further explore whether *in vivo*-derived interactions measured using our approach resemble those occurring *in situ*, we qualitatively compared our LCMV *Interact-omics* results with imaging and *in situ* interaction mapping of approaches investigating LCMV infections:

- Monocyte–B Cell Interactions: Imaging has identified a drastic increase in monocyte–B cell interactions following LCMV infection (Sammicheli et al., 2016, Science Immunology). In line with this, our approach also identified a major increase in monocyte–B cell interactions upon LCMV infection, matching the kinetics described by Sammicheli et al.
- CD8 T Cell Interactions: A recent study mapped the interactions of antigen-specific CD8 T cells following LCMV infection using a newly developed universal version of the labeling immune partnerships by SorTagging intercellular contacts (uLIPSTIC) mouse model (Nakandakari-Higa et al., 2024, Nature). Although this model records past rather than current interactions and investigated slightly different time points, the types of interactions that antigen-specific CD8 T cells undergo can be compared. Notably, there was a high concordance in the types of interactions that antigen-specific CD8 T cells engage in within the uLIPSTIC model and our approach, including interactions with B cells, CD4 T cells, bystander CD8 T cells, NK cells, Neutrophils and other myeloid cells.
- Antigen-Specific T Cell Activation and Proliferation: Activation and proliferation of transferred antigen-specific T cells have been described to start in spleens around 3 days post LCMV infection and increase thereafter (Olson et al., 2012, PLOS Pathogens). A delay in kinetics has been described in non-draining lymph nodes compared to the spleen. Similarly, our approach identified initial interactions of antigen-specific T cells at day 3 post-infection, followed by a more pronounced increase at day 7. Consistent with Olson et al., the spleen showed more pronounced antigen-specific T cell interactions compared to non-draining lymph nodes at matching time points, likely due to more rapid kinetics.

Collectively, these observations suggest that while it cannot be unequivocally determined whether all measured interactions in the *Interact-omics* approach have occurred *in vivo*, the interactions are not random but reflect biological effects. The results align fully with those from *in situ* methods. Still, we have now acknowledged the limitations and outlined resulting guidelines on how to optimize our approach in the revised version of the manuscript.

How well does the proposed FSC thresholding work compared to alternative strategies? For conventional cytometry, one would create a “singlet” and a “doublet” box/ellipse/polygon gate rather than a single threshold, and typically double-gate based on both FSC-H vs FSC-W and FSH-area vs FSC-W. Such an approach removes borderline doublet events from the analysis. How does this conventional approach compare to an unsupervised threshold?

We thank the reviewer for this interesting question. We have addressed this point as part of this reviewer's previous request on a more general benchmarking (Fig. R1A, Table R1). Here, we enquired three expert cytometry users to perform manual gating based on a "singlet" and a "doublet" box/ellipse/polygon gate. Notably, this resulted in a performance comparable to data-driven approaches for scatter-based thresholding/gating, including Otsu thresholding on FSC-ratio (Fig. R1A). However, manual gating, like similar approaches, was considerably inferior to the proposed *Interact-omics* approach, employing an initial clustering on all cell type-specific markers and scatter properties, followed by a cluster-based classification into singlets and multiplets based on FSC-ratio and subsequent annotation based on exclusive markers (Fig. R1B). Notably, in contrast to a conventional gating approach, the proposed framework not only discriminates singlets from multiplets but also quantitatively resolves cellular interaction partners.

How well does the method perform when the markers do not perfectly mark all the cell types in the data? For exploratory research this will often be the case - especially if non-immune cells are analyzed, since specific markers are often not available. Experiments can be done in silico based on existing experiments, by removing the data for progressively more markers and determining if results are corrupted.

We thank the reviewer for highlighting this important aspect. In flow cytometry, the design of the panel largely determines the granularity of resolution for both cell types and cell states, which also extends to cellular interactions. To address the reviewer's point, we conducted an analysis where we progressively identified the least important feature (marker) and reduced the number of features by sequentially removing the least significant one from the input to the workflow (Fig. R6; item A for the data underlying main Figure 2A—cellular interactions in human PBMCs after CytoStim—and item B for the data underlying main Figure 5B—virus-induced cellular interaction networks in mice—as representative examples). We then compared the results obtained from the reduced and full feature sets using the adjusted Rand index. As shown in Figure R6, the adjusted Rand index remains consistent within a range of approximately 15 to 25 features, demonstrating robustness to the inclusion or exclusion of individual features.

Figure R6. Comparison of adjusted Rand index across different numbers of features. **A.** Feature importance analysis for data from Figure 2A. The adjusted Rand index was calculated after iteratively removing the least important feature. **B.** Feature importance

analysis for data from Figure 5B. The adjusted rand index was calculated after iteratively removing the least important feature.

Notably, we provide fully optimized high parametric panels for mouse and human that enable cytometry-based immune interaction mapping at high resolution and can be adapted for new experiments. To practically demonstrate that our approach can also be applied in settings where high parametric panels have not been optimized for our method, we applied the PICtR workflow on a publicly available dataset of juvenile idiopathic arthritis (JIA) (Attrill et al., 2024, *Clinical and Experimental Immunology*) (Fig. R7).

JIA is an autoimmune disease characterized by chronic inflammation of the joints, leading to pain, swelling, and eventual joint damage. While it is hypothesized that an abnormal interaction among immune cells, specifically T cells, B cells, and myeloid cells, contributes to the production of inflammatory cytokines and autoantibodies that drive the disease, the actual interaction processes remain poorly understood.

Within this study, Attrill and colleagues analyzed PBMC samples from healthy donors, JIA patients with active and inactive disease as well as synovial fluid samples from JIA patients with active disease. The data is available on www.flowrepository.org under FLOWRepository ID FR-FCM-Z6VC. FCS files were downloaded and preprocessed as described in our Methods section. Additionally, flowAI (a QC algorithm) was run on all FCS files and FCS files with anomalies in their flow rate were excluded from further analysis. High quality samples were then analyzed with the PICtR workflow. Results described by Attrill et al. concerning the single-cell landscape could be reproduced (Fig. R7A, B). Focusing on the interacting cell landscape, we explored three comparisons: 1. PBMCs from healthy donors vs. PBMCs from patients with JIA (Fig. R7D-F), 2. PBMCs from JIA patients with active disease vs. inactive disease (Fig. R7G-I), 3. PBMCs from JIA patients with active disease vs. synovial fluid of JIA patients with active disease (Fig. R7J-L). Notably, we discovered both quantitative and qualitative changes in cellular interactions in the blood of patients with inactive versus active disease, as well as between the blood and synovial fluid of affected joints (Fig. R7D-L). Among the most intriguing findings is the enrichment of T cells interacting with B cells, which predominantly comprise a FoxP3-expressing regulatory T cell phenotype in patients with inactive disease (Fig. R7I). However, these interactions switch to an inflammatory, non-regulatory phenotype in patients with active disease. Similarly, major qualitative differences of interactions between CD4 T cells and monocytes were observed between blood and synovial fluid of patients with active disease (Fig. R7L). While several of these findings require further validation, they provide a first quantitative framework for understanding changes in immune cell interactions that may contribute to disease progression and help identify targets for therapeutic intervention. Collectively, this analysis demonstrates that our approach can be applied in settings where high parametric panels have not been optimized for our method, given that the sample set has been acquired under constant experimental conditions (see limitations and guidelines section of the manuscript).

Figure R7. Interacting cell landscape in juvenile idiopathic arthritis (JIA).

A. Schematic describing publicly available spectral flow cytometry data of PBMCs and SFMC of JIA patients (Attrill et al., 2024; Clinical and Experimental Immunology). Three comparisons (indicated by the arrows) were made for the interacting cell landscape **B**. UMAP of the overall cellular landscape. Recorded cells were processed with PICtR, out of 7,843,646 cells, 80,000 sketched cells are displayed. **C**. UMAP of interacting cells ($n = 12,908$) **D**. Point density UMAP (left panel) and differential abundance (right panel) of interacting cells comparing PBMCs from healthy donors vs. JIA patients. **E**. Quantitative comparisons of interacting cell frequencies between PBMCs from healthy donors ($n=18$) and JIA patients ($n=36$). Top: Non-adjusted frequencies. Bottom: Interaction frequencies normalized by the harmonic mean of the singlet frequencies of the contributing cells (see Methods). P values were determined with a two-sided t-test and adjusted for multiple testing using Benjamini-Hochberg correction. **F**. Qualitative differences in CD4T*cl.mono interactions. P values were determined with a two-sided Wilcoxon rank sum test and adjusted for multiple testing using Benjamini-Hochberg correction. **G**. Point density UMAP (left panel) and differential abundance (right panel) of interacting cells comparing PBMCs from inactive JIA vs. active JIA patients. **H**. Quantitative comparisons of interacting cell frequencies between PBMCs from inactive JIA ($n=18$) and active JIA patients ($n=36$). Top: Non-adjusted frequencies. Bottom: Interaction frequencies normalized by the harmonic mean of the singlet frequencies of the contributing cells (see Methods). P values were determined with a two-sided t-test and adjusted for multiple testing using Benjamini-Hochberg correction. **I**. Qualitative differences in CD4T*cl.mono interactions. P values were determined with a two-sided Wilcoxon rank sum test and adjusted for multiple testing using Benjamini-Hochberg correction. **J**. Point density UMAP (left panel) and differential abundance (right panel) of interacting cells comparing SFMC from active JIA vs. PBMCs from active JIA patients. **K**. Quantitative comparisons of interacting cell frequencies between SFMC from active JIA ($n=18$) and PBMCs from active JIA patients ($n=36$). Top: Non-adjusted frequencies. Bottom: Interaction frequencies normalized by the harmonic mean of the singlet frequencies of the contributing cells (see Methods). P values were determined with a two-sided t-test and adjusted for multiple testing using Benjamini-Hochberg correction. **L**. Qualitative differences in CD4T*cl.mono interactions. P values were determined with a two-sided Wilcoxon rank sum test and adjusted for multiple testing using Benjamini-Hochberg correction.

correction. G. Point density UMAP (left panel) and differential abundance (right panel) of interacting cells comparing PBMCs of JIA patients with inactive disease (n=11) vs. active (n=25). H. Quantitative comparisons of interacting cell frequencies between PBMCs from JIA with inactive and active disease. Top: Non-adjusted frequencies. Bottom: Interaction frequencies normalized by the harmonic mean of the singlet frequencies of the contributing cells (see Methods). P values were determined with a two-sided t-test and adjusted for multiple testing using Benjamini-Hochberg correction. I. Qualitative differences in T*B interactions. P values were determined with a two-sided Wilcoxon rank sum test and adjusted for multiple testing using Benjamini-Hochberg correction. J. Point density UMAP (left panel) and differential abundance (right panel) of interacting cells comparing PBMCs of JIA patients with active disease vs. SFMC of active disease. K. Quantitative comparisons of interacting cell frequencies between PBMCs of JIA patients with active disease (n=25) vs. SFMC of active disease (n=8). Top: Non-adjusted frequencies. Bottom: Interaction frequencies normalized by the harmonic mean of the singlet frequencies of the contributing cells (see Methods). P values were determined with a two-sided t-test and adjusted for multiple testing using Benjamini-Hochberg correction. L. Qualitative differences in CD4T*mono interactions. P values were determined with a two-sided Wilcoxon rank sum test and adjusted for multiple testing using Benjamini-Hochberg correction. Abbreviations: UMAP = uniform manifold approximation and projection, PBMC = peripheral blood mononuclear cells, SFMC = synovial fluid mononuclear cells. Red asterisks in cell type labels indicate interactions between the respective cell types.

Clustering is such a critical part of this method that better guidance is needed. The potential for issues is well demonstrated when comparing Fig. S1A, which shows the 14 unsupervised clusters to Fig 1D, which shows the expert-curated 5 clusters. The only clusters that remain untouched are the three doublet/triplet clusters. How were the parameters for the initial clustering selected? Did you already know the expected doublet clusters and adjusted based on this?

We thank the reviewer for highlighting this important aspect. We fully agree that clustering is critical. In this analysis, we follow widely accepted practices from the single-cell genomics field, where data is commonly over-clustered relative to the expected number of cell types (Weber et al., 2019, communications biology; Saeys et al., 2018, Nat. Rev. Immunol.). This approach ensures the detection of smaller populations, while broader populations can be merged if they express markers indicative of the same cell type or if finer annotations are unnecessary for the specific question. Another common strategy in single-cell genomics is to sub-cluster certain subsets of cells for a more detailed resolution.

In line with these conventions, the *Interact-omics* workflow iterates through multiple resolutions, annotating based on one that yields more clusters than anticipated based on the marker panel. Merging is performed afterwards if necessary. In the framework of the revision process, we re-analyzed the data underlying Figure 1 using 100 iterations of Louvain clustering. In the example provided, we initially over-clustered the data into 14 clusters (Fig. R8A). These were then merged into singlet T cells, B cells, CD33^{high} myeloid cells, CD33^{low} myeloid cells, and other CD45-positive cells due to their marker expression patterns and low FSC ratio. Two clusters were identified to contain cellular interactions based on high FSC ratio and co-expression of cell type-exclusive markers (Fig. R8B, C). While the majority of clusters showed a high degree of homogeneity, one of the cellular interaction clusters displayed

heterogeneous marker expression and was therefore subjected to subclustering, yielding in total three clusters of interacting cells: B*T, My*T, and My*T*B cell populations (Fig. R8C, D).

It is important to note that in the example shown in Fig. 1, a low-plex panel was used due to the acquisition method involving an imaging flow cytometer, which was simultaneously used to generate ground truth data. As a result, the resolution between distinct interacting cell clusters is lower compared to all other analyses presented in the manuscript.

Figure R8: Clustering and annotation in PICtR.

A. UMAP embedding colored by the 14 Louvain clusters based on conventional flow cytometry parameters. Corresponds to Figure 1F. **B.** Stacked bar plot for each of the 14 clusters colored by whether events fall above or below the FSC ratio threshold (Otsu-based, see Figure 1D). Out of the 14 clusters, those clusters containing the most cells exceeding the FSC ratio threshold are classified as interacting clusters, here clusters 9 and 11. **C.** Feature plots for the UMAP embedding in panel A, colored by the expression of the cell type markers and the FSC ratio. **D.** Annotated UMAP embedding from panel A (corresponds to Figure 1F). Due to the marker expression in C, some clusters are merged and a focused analysis of clusters representing cellular interactions resulted in 3 populations, namely B*T, My*T and My*T*B populations.

“Singlet” clusters seem to often contain around 10% doublets (by FSC), considerably higher than I would expect if they were homotypic interactions (See Fig. S5D). Also in the ground-truth experiment Fig 1E, there seems to be ~5-10% of singlets in the doublet classes. In Fig S1B it does not look like there is a threshold that separates the two populations, but rather a continuum. Are these issues experiment-specific? Is it a side-effect of the sketching procedure (i.e selecting rare events)?

We thank the reviewer for highlighting this aspect. To clarify, the plots showing the frequency of classified cells within a specific cluster or population focus on either classification based solely on the FSC ratio (as in the former Figure S5D, now Figure S6D) or classification based on clustering combined with FSC ratio thresholding, compared to the ground truth annotation (as in the former Figure 1E, now Figure 1H). The stacked bar plots presented for events below or above the FSC ratio threshold (e.g., Figure 1G, Supplementary Figure 2C) may therefore still contain misclassified cells, as classification based solely on the FSC ratio is less accurate than the two-step approach introduced by us.

When classifying cells only based on FSC-ratio thresholding (without prior clustering), we generally observe that ground-truth singlets which are misclassified as multiplets are predominantly myeloid cells, likely due to their distinct scattering parameters (Fig. R9). Depending on the data set and the heterogeneity of included cell types, the FSC ratio might not have a clear bimodal distribution. Of note, this is not a side effect of the sketching procedure, since thresholding of the FSC ratio is performed on all events.

The proposed two-step approach, which involves unsupervised clustering prior to classification into singlet and interacting clusters, mitigates these errors and therefore provides superior accuracy of 97.8 % and a mean F1 score of 0.91 for the ground-truth data set (see Figure 1E of the revised manuscript, Supplementary Figure 1M, Fig. R1B).

Figure R9. Properties of singlets misclassified by thresholding of the FSC ratio. Dot plot displaying the forward scatter area (FSC-A) and forward scatter height (FSC-H) properties of ground truth singlets; the Otsu threshold of the FSC ratio is shown as a diagonal line. The bar plots show all ground truth singlets split into correctly classified and misclassified events according to the FSC ratio threshold and are colored by marker expression. $n = 4$ replicates; error bars indicate the standard deviation.

Minor issues

Line 38: "Saliva", not "Salvia"

We have now adjusted this.

Line 41-43: "These technologies mostly focus on interaction of pre-defined cell types...". It is not relevant what the majority methods do; both Andrews and Boisset papers allow analysis of non-pre-defined cell types and are cited here. The difference in throughput and price is a major advantage, and of course highly relevant. However, these differences are similar to differences in price between any mRNA and cytometry based method, and the drawbacks are also the same: cytometry-based methods operate only on cell types with known markers whereas mRNA-based methods are hypothesis-free. It is not appropriate to compare such vastly different methods mainly on price per cell.

We agree with the statements of the reviewer and have now adjusted the wording in the manuscript.

Fig 1H: How many data points constitute the boxplots? Boxplots show a distribution and are only suitable if the number of observations is high. Please indicate n= and plot individual points instead of boxplots if n is low.

We have now implemented this throughout the manuscript and display individual data points where applicable. Moreover, we now provide a description of the number of data points for each experiment in the figure legend.

Fig 2D: This figure is confusing - what does the color shading mean? What does the thickness mean?

It looks like the left/right side were one plot that was later separated, which makes the plot look a little bit awkward (lines originate at random positions within fields) - maybe better to keep them together and indicate -/+CytoStim with color?

We have now adapted the plot according to the suggestion of the reviewer.

Fig 3E: Here n is clearly too low for drawing boxplots. Please just show the individual observations instead.

This has now been implemented.

Fig 6F-I: Same thing, no boxplots with low n.

This has now been implemented.

Fig S4 J,K: Same issue with boxplots.

This has now been implemented.

Fig S1 F and G. These figures are meant to show that cell concentration does not have a strong effect on doublet formation in PBMCs, but the ranges are too different - the lowest spleen concentration is higher than the highest PBMC concentration.

We thank the reviewer for highlighting this aspect. As part of our newly conducted, extensive *ex vivo* benchmarking, we now provide a more detailed analysis across a range of human PBMC concentrations, including similar ranges as for the spleen samples analyzed (see Supplementary Figure 4G or Fig. R3B). These analyses show that as PBMC concentrations increase (from 25,000 to 250,000 cells per well, corresponding to 500 to 5,000 cells/ μ L during staining and acquisition), there is a slight increase in overall detected interactions. Similarly, though to a more pronounced extent, a significant increase in interactions is observed among splenocytes as their concentration increases (from 2,500 cells/ μ L to 10,000 cells/ μ L; see Supplementary Figure 4I). Importantly, our intention was not primarily to imply that cell concentration is more significant in one tissue type over another, but rather to emphasize that, alongside technical factors, the tissue type being studied can also impact the observed cellular interactions. Therefore, we strongly recommend maintaining consistent experimental conditions and directly comparing matching tissue types for accurate assessment. In our manuscript, we state: "Collectively, these data provide quantitative insights into how experimental settings impact background interactions while demonstrating that *ex vivo* modulations of cellular interactions can be effectively quantified in the explored settings. Besides incubation times, cell numbers and fixation, flow rates and tissue type may significantly impact baseline interactions (Supplementary Figure 4H-I). Therefore, maintaining these parameters constant is critical for maximizing the recovery of signal to background interactions. Limitations and guidelines are further detailed in a separate section (see Limitations and Guidelines)."

Line 710-718: Should be rewritten to precisely describe the algorithm, or if this is not possible removed and simply replaced by the reference (Hao et al?). Currently it just reads as word-salad and does not aid understanding.

We have now adapted this accordingly.

Line 721-722: What is "the maximum number of dimensions in the data"? Do you mean the actual dimensionality of the dataset (eg. the number of measured values per cytometry event)? In that case reducing this to $n-1$ seems strangely redundant - the data is (almost) the same size, so why the dimensionality reduction? How have different n been evaluated to arrive at $n-1$ as the optimum?

We apologize for the misleading wording. Indeed "the maximum number of dimensions in the data" meant the actual dimensionality of the dataset. We also agree that reducing to $n-1$ dimensions has minimal impact; however, it does offer a slight improvement in runtime, as shown in items B and D in Figure R10, with only marginal changes to the adjusted Rand index (items A and C in Figure R10). Based on this benchmark, there is flexibility in selecting the factorization rank. We opted for $n-1$ and, for consistency, kept it constant throughout the manuscript. In the PICtR software package, we have updated this parameter from being hard-coded to a customizable option for users."

Figure R10. Comparison of adjusted Rand index and runtime performance across different numbers of principal components. A. The adjusted Rand index was calculated after iteratively removing the last principal component. Data of Fig. 2A of the manuscript. **B.** Runtime analysis (in sec.) for 1 mio. cells performed when iteratively removing the last principal component. Data of Fig. 2A of the manuscript. **C.** The adjusted Rand index was calculated after iteratively removing the last principal component. Data of Fig. 5B of the manuscript. **D** Runtime analysis (in sec.) for 1 mio. cells performed when iteratively removing the last principal component. Data of Fig. 5B of the manuscript.

Line 726-: Clustering is clearly critical to obtaining accurate results using this method, so some guidelines would be in order here.

We fully agree. Choosing a resolution requires curation and expert knowledge about the expected cell identities in a given tissue/sample/panel and the observation of patterns of marker expression (e.g. feature plots), cf. Luecken and Theis (Luecken and Theis, 2019, mol systems biology). We now provide additional guidelines on this topic in the package’s vignette (section “3.3.1 Guidelines for Clustering”).

Line 735-737: This cannot be universally true. Each cluster has a fraction of cells with FSC ratio > threshold, and the top 15-20% of the clusters are classified as doublets. But in Fig. 1 three out of seven clusters are doublet/triplet clusters, which is >40%. Even if you mean the initial 14 clusters, it amounts to >20% of the clusters. This part of the method has to be better described!

We thank the reviewer for highlighting this important point. The 15-20% estimate for clusters was intended as a general guideline, reflecting our broader experience across various setups. However, we fully agree that this percentage can vary significantly depending on the specific experimental context and biological questions being addressed. In light of this, we have decided to avoid making such general recommendations and instead offer more tailored guidance.

Line 749-751: This section (“Annotation of physically interacting cells”) describes a different method for determining interacting cell clusters “The key criterion for identification of interacting cell clusters was a high FSC-ratio and the presence of more than one cell type-

specific marker.” Which one is correct? Or has different heuristics been applied to different parts of the manuscript?

We thank the reviewer for pointing out this aspect and apologize for the confusion caused by this ambiguity. The process of annotating interacting cells is the same throughout the manuscript. Briefly, we perform clustering, and select interacting populations based on the amount of events with a high FSC ratio that fall into a given cluster and annotate the interaction partners based on mutually exclusive cell type markers. The only exception is the ImageStream data for the *in vivo* benchmarking to imaging flow cytometry that we included in the revision, since ImageStream data does not contain the forward scatter parameter necessary for calculating the FSC ratio. Here, interacting populations were only annotated based on the presence of mutually exclusive cell type markers, probably undermining the performance of our approach. We have now improved the wording in the manuscript to avoid any confusion (see Methods section).

Reviewer #2:

Remarks to the Author:

The paper proposes a new method for detecting and studying physical cell-cell interactions using flow cytometry. The task is very important and there's great interest and progress recently in developing methods for solving it. The proposed method has potential advantages as compared with existing methods, e.g. in cell throughput, potential ease of use, and applicability e.g. to human samples because no genetic engineering is required. The method relies on using parameters measured in flow cytometry that are demonstrated to work well in discriminating single cells from cell doublets and multiplets. The results of application of the method in several contexts are reasonable and actually very interesting, especially in analysis of immunotherapy response. The paper is well written. The software can be run and the results can be reproduced (but see below). However, I have questions about robustness, precision and recall of the approach with respect to true physiological cell interactions *in vivo*, which may affect applicability and interpretability of the method.

We thank this reviewer for the insightful comments and suggestions that have greatly enhanced the quality of our manuscript. In response to the reviewer's questions, we have performed an extensive benchmarking and characterization of our method. Taking the respective insights into consideration, we have added a new section to the manuscript detailing the limitations of our approach and providing guidelines for best practices to optimize performance. Additionally, in response to the reviewer's request, we have demonstrated that our approach can infer true biological cellular interaction networks from existing cytometry data, given specific preconditions, thereby significantly broadening the method's applicability and scope. A detailed response to each of the reviewer's points is provided below.

More specific comments:

While I am not an expert in technical aspects of flow cytometry, it is convincing that the method can detect and report some cell interactions. However, it is unclear how physiologically relevant these detected cell interactions are, and how representative they are of all interactions

in vivo. How many and which of the detected cell interactions form as a result of sample manipulations? In principle some cells (e.g. DCs and T cells) may be "sticky" and may form interactions in a suspension as a result or byproduct of some technical manipulations with samples, but not *in vivo*. But does it really happen, and if yes, does it happen in a biased manner, more so for some cell types and cell states than others? Also, which of the cell interactions happening *in vivo* in physiological contexts are actually preserved in all sample preparation and sample handling and then detected with this method? Is it for the strongest interactions? or only interactions of some specific types? or some random ones?

We thank the reviewer for raising this highly relevant question. The proposed method quantifies single-cell landscapes, as well as the frequencies and types of interactions among physically interacting cells in a given sample at the time of analysis. This provides a snapshot of current interactions, which may include both random technical interactions and specific interactions that reflect biological effects. When analyzing interactions derived from an *in vivo* setting, new interactions might be acquired during sample preparation. These are inherent limitations to the growing family of methods that map the current state of physically interacting cells, such as PIC-seq, CIM-seq, SPEAC-seq, Clumb-seq, paired-cell sequencing or imaging cytometry. We fully agree with the reviewer that systematically understanding these limitations is crucial for interpreting the results of the proposed and related methods. Therefore, we have conducted a series of *ex vivo* and *in vivo* benchmarking experiments to thoroughly assess the applicability and limitations of our approach, including analyses of the functionality and specificity of cellular interactions.

1. *Ex vivo* cellular mixing experiment

To assess the extent to which additional cellular interactions are formed during culture and how different experimental settings impact this process, we conducted an extensive *ex vivo* benchmarking experiment involving labeling and mixing of cell populations. We first induced cellular interactions in PBMCs with CytoStim or left cells untreated. The PBMCs were then split and labeled with two different fluorescently conjugated antibodies against CD45 (Fig. R11A). Subsequently, the distinctively labeled PBMCs were reunited and processed at varying cell concentrations, processing times, and fixation methods, followed by mapping cellular interactions using our approach. This allowed us to effectively identify singlets positive for each label (Fig. R11B, C), as well as cellular interactions that were either single- or double-positive for both labels (Fig. R11D-F). Double-positive cellular interactions for the two introduced labels were newly acquired during the second incubation period, while single-positive interactions could have been acquired initially or throughout the culture period. Focusing on B-T cell interactions, as one of the most frequent interactions induced by CytoStim, our experiments revealed that single-positive interactions increased robustly upon CytoStim treatment. In contrast, double-positive (newly acquired) interactions showed only a mild increase, which appeared negligible compared to the induction of single-positive interactions (approximately 5-10-fold higher in single positives compared to double positives) (Fig. R11G). Increasing incubation periods post-CytoStim did not cause a stark increase in newly acquired (double-positive) interactions, while a trend toward a decrease in CytoStim-induced (single-positive) B-T cell interactions was observed (Fig. R11G). Even at 4 hours post-incubation, CytoStim-induced single positives were on average 7-fold more frequent than their double-positive counterparts. These data suggest that *ex vivo* induced cellular interactions are

relatively stable, and newly acquired interactions occur but have a relatively minor impact in this setting. With increasing cell concentrations, a mild baseline increase across all interactions (both single and double positives, with and without CytoStim) was observed (Fig. R11G). However, the relative effect of CytoStim-induced single-positive interactions compared to newly acquired double-positive interactions remained comparable. These data reinforce the importance of maintaining stable cell concentrations within experimental settings but also demonstrate that *ex vivo* induced cellular interactions can be effectively measured across distinct concentrations, providing stability across conditions. Finally, we evaluated the impact of chemical fixation on newly formed *ex vivo* interactions. PBMCs were either fixed or left unfixed after +/- CytoStim treatment, labeling, reuniting and staining. Overall, only a minor impact of fixation was observed, with a trend towards higher single and double positive interactions upon CytoStim treatment in fixed samples (Fig. R11G). This experiment suggests that cellular interactions can be retrieved using fixation or leaving samples in their native state, provided the same conditions are applied to all samples. We recommend, however, to fix samples as early as possible during the sample preparation workflow. Collectively, these data provide quantitative insights into how experimental settings impact background interactions while demonstrating that *ex vivo* modulations of cellular interactions can be effectively quantified in the explored settings. Besides incubation times, cell numbers, and fixation, flow rates significantly impact baseline interactions, as previously demonstrated (Fig. R11H-J). Therefore, maintaining these parameters constant maximizes the recovery of signal to background interactions. We have now included a limitations and guidelines section in the manuscript detailing how to optimize the experimental setting for *ex vivo* cytometry-based cellular interaction mapping.

Figure R11. Effect of sample processing methods on ex vivo cellular interactions.

A. Schematic depiction of the experimental approach. **B.** Overall cellular landscape across all experimental conditions. $n = 3,292,837$. **C.** Feature plots for the UMAP display in B, colored by the two differently labeled antibodies against CD45. **D.** Interacting landscape across all experimental conditions. $n = 23,620$. **E.** Dot plot for the CD45 signals in interacting populations, showcasing single-positive and double-positive interactions. **F.** Feature plots for the UMAP from panel D, colored by the CD45 signal intensities. **G.** Quantification of single-positive and double-positive, newly acquired interactions in CytoStimTM treated and untreated samples, across different experimental conditions. Left: Varying incubation times at 4°C after mixing, mimicking long sample processing times. Middle: Different cellular concentrations ranging from 25,000 to 250,000 cells in 50 μ L during staining and acquisition. Right: Fixation with 2 % paraformaldehyde after staining compared to no fixation. The number of replicates is shown in each plot and ranges from $n = 2$ to 3. **H.** Short-term cultures of PBMCs with or without CytoStimTM measured at low or high flow rate ($n = 4$). Impact of flow rate on cellular interactions is relatively mild. Two-way ANOVA (CytoStimTM: $F(1,13) = 189.138$, $P = 4.01e-09$, flow rate: $F(1,13) = 6.598$, $P = 0.023$), followed by Tukey's Honest Significant Differences test for the flowrate. **I.** Impact of flow rate and cellular concentration on cellular interactions in

murine spleens is more pronounced. Left: Baseline interactions in spleens measured with different flow rates ($n = 4$). Right: Baseline interactions in spleens at different cell densities but constant flow rate ($n = 4$). One-way ANOVA (flow rate: $F(2,9) = 115.749$, $P = 3.79e-07$, cell concentration: $F(2,9) = 61.397$, $P = 5.68e-06$), followed by Tukey's Honest Significant Differences tests. J. Boxplots of T*B cell interactions upon Blinatumomab treatment that were either fixed or fixed after a freeze-thaw cycle ($n = 4$). P values were determined with a two-sided Welch's t-test.

2. *In vivo* benchmarking to imaging flow cytometry

To provide additional information on the reliability of interactions assessed in particular *in vivo* settings, we performed ImageStream-based imaging flow cytometry of LCMV-infected spleens at day 7 post-infection, similar to our experimental setup for organism-wide cytometry-based cellular interaction mapping presented in the manuscript (Fig. R12). While imaging cytometry requires specialized instruments, has a low cellular throughput, and is limited in the number of measured fluorochromes and cell types that can be resolved, it provides morphological imaging information that can classify cells into single versus interacting cells. Due to the limited number of available channels, we utilized an 6-plex panel focusing on T and B cell interactions. Upon data generation, we extracted the fluorescence intensity values and applied the PICtR workflow without considering any morphological information (Fig. R12A-C). As the FSC ratio cannot be extracted during the ImageStream workflow, clustering and interaction identification was performed on fluorescence channel values only. Consequently, the results presented here likely underestimate the performance of the *Interact-omics* approach. We then evaluated the performance of the *Interact-omics* approach with information gained from both images and fluorescence values regarding (i) discriminating singlets from interacting cells, (ii) identifying cell type combinations of interactions, and (iii) deriving qualitative changes in LCMV-induced interactions. For this purpose, we first compared the results of the *Interact-omics* workflow to an image segmentation-based classification of singlets and multiplets, demonstrating a high concordance between the approaches regarding the discrimination between single cells and multiplets (Fig. R12D). While neither image-based segmentation nor our approach reflects ground truth data, the high concordance strongly supports the validity of the *Interact-omics* results. Manual inspection of randomly selected images predicted as singlets or interacting cells further verified the accuracy of the *Interact-omics* approach (Fig. R12C).

Next, we immunophenotypically characterized interacting populations using conventional gating on marker expression values extracted from images (Fig. R12E). As expected, this revealed a high concordance with the cluster-based annotations from the *Interact-omics* workflow. Manual inspection of randomly selected images regarding the localized expression patterns of lineage-specific markers confirmed the expected types of interactions (Fig. R12C). Finally, we compared the LCMV-induced alterations of cellular interactions as detected by *Interact-omics* vs. the imaging flow cytometry approach. Due to the low plexity and low cellular throughput of the imaging flow cytometry approach, only very crude interacting cell populations could be defined, restricting the comparison to the populations depicted in Fig. R12B. Nonetheless, interacting cell populations identified with both approaches showed qualitatively matching alterations induced by LCMV, including a marked increase in interactions involving antigen-specific CD4 T cells (Fig. R12F). Collectively, these data confirm the accuracy of the

Interact-omics approach in discriminating single and interacting cells and in quantifying the types of interactions in case-control settings.

Figure R12. Comparison of the *Interact-omics* approach to imaging flow cytometry.

A. UMAP representation of the overall cellular landscape derived from the fluorescent intensity values, $n = 306538$. Intensity values are based on the sum of the pixel intensities in the mask as selected by ImageStream@X, background subtracted. The experiment corresponds to day 7 in Supplementary Figure 11. **B.** UMAP representation of the interacting landscape, $n = 8683$. The heterogeneous cluster is mostly comprised of likely B*CD4*CD8 multiplets. The unknown cluster expresses CD19 and CD3 but no other T cell markers, hindering confident annotation. **C.** Pseudo-colored example images for cellular interactions in the brightfield and fluorescence channels. **D.** Left: UMAP displays from A and B colored by the number of cells identified through image segmentation. Right: Bar plots comparing the populations identified through *Interact-omics* and the image segmentation. **E.** Left: UMAP displays from A and B colored by populations as identified through conventional gating. Right: Bar plots comparing the

populations identified through *Interact-omics* and conventional gating. NA indicates that the event does not fall into any conventional gate. **F.** Fold changes of the frequencies +/- LCMV infection. Holm-corrected estimated marginal means comparison. Left: Populations identified by *Interact-omics*. Right: Populations identified through conventional gating. **G.** Gating strategy for conventional gating. Abbreviations: Ag = antigen-specific, UMAP = uniform manifold approximation and projection, BF = brightfield.

3. *In vivo* mixing experiments to characterize newly acquired cellular interactions

The presented approach measures cellular interactions following sample preparation *ex vivo*. Consequently, for *in vivo* applications, additional cellular interactions can be acquired or lost during sample preparation. While this limitation applies to all cellular interaction mapping approaches that do not rely on specialized mouse models or measure co-localization *in situ*, it remains poorly characterized to what extent this occurs, whether newly acquired interactions are random or directed, and how representative the identified interactions are of the *in vivo* situation.

To evaluate these questions, we utilized congenic mouse models differing in variants of the pan-hematopoietic cell marker CD45, allowing identification of respective immune cells as CD45.1 or CD45.2 using variant-sensitive antibodies. First, we transferred LCMV-specific CD4 T cells (SMARTA: CD90.1 positive, CD45.2 positive) into CD45.2 mice, followed by LCMV infection (group A) (Fig. R13A). Non-infected control CD45.2 mice were included (group B). In parallel, we infected CD45.1 mice with LCMV (group C) or left them untreated (group D). On day 7 post-infection, spleens from all groups were harvested. Spleens from group A (infected, CD45.2) and group B (non-infected, CD45.2) were either processed individually or mixed in 1:1 ratios with spleens from group C (infected, CD45.1) or group D (non-infected, CD45.1) before tissue homogenization and processing. Applying the *Interact-omics* workflow to these single and mixed samples resulted in single-cell and interacting cell landscapes of populations that were either single positive or double positive for CD45.1 and CD45.2 (Fig. R13B-E), with double-positive populations being newly acquired interactions during processing. Notably, we observed a substantial acquisition of new interactions during sample processing (Fig. R13F). However, these newly acquired interactions did not occur randomly but were highly correlated with interactions induced upon infection (Fig. R13G). In particular, newly acquired interactions in mixed spleens from infected mice compared to non-infected controls were highly correlated with infection-induced single positive interactions in both mixed and non-mixed spleens (Fig. R13G). This suggests that while new interactions can be acquired during sample preparation, they are not random but reflect actual biological effects and likely are a proxy for cellular interactions occurring *in vivo* (see also next section about qualitative comparison to *in situ* methods).

Figure R13. Interactions acquired a posteriori after *in vivo* experiment.

A. Schematic overview of the experimental approach. LCMV-specific CD4⁺ T cells were transferred into CD45.2 host mice 5 days before infection with LCMV (group A) or the respective control (group B). Additionally, CD45.1 host mice were infected with LCMV (group C) or left untreated (group D). *n* = 3 for groups A, C, D and *n* = 4 for group B. **B.** Single-cell landscape of all experimental groups. Out of *n* = 23,490,812 processed cells, *n* = 245,316 are shown in the UMAP display. **C.** Feature plots for panel B, colored by the expression of the congenital markers CD45.1 and CD45.2. **D.** Interacting landscape across all experimental groups. Out of *n* = 731621 identifiable interactions, *n* = 93065 are shown. **E.** Dot plots showcasing the expression of the congenital markers CD45.1 and CD45.2 in interacting populations from the unmixed controls for the untreated and infected conditions, and the mixed spleens from infected mice (infected+infected, group A + group C) or untreated mice (control+control, group B + group D). **F.** Bar plots depicting the log₂ fold changes (FC) between the LCMV infected and untreated conditions for each interacting population. Solid bars indicate the log₂FC between group A (infected) and group B (control). Semi-transparent bars show the log₂FC for single-positive interactions in mixed samples (A+C for the infected condition, and

B+D for the control). Transparent bars depict log₂FC between the respective double-positive interactions, which were definitely acquired *ex vivo*. n = 3 mixes. **G.** Linear relationships between the log₂FC between infected and control conditions for unmixed controls, single-positive interactions after mixing and double-positive interactions after mixing. n = 3 mixes.

4. Qualitative comparison to *in situ* methods

To further explore whether *in vivo*-derived interactions measured using our approach resemble those occurring *in situ*, we qualitatively compared our LCMV *Interact-omics* results with imaging and *in situ* interaction mapping of approaches investigating LCMV infections:

- Monocyte–B Cell Interactions: Imaging has identified a drastic increase in monocyte–B cell interactions following LCMV infection (Sammicheli et al., 2016, Science Immunology). In line with this, our approach also identified a major increase in monocyte–B cell interactions upon LCMV infection, matching the kinetics described by Sammicheli et al.
- CD8 T Cell Interactions: A recent study mapped the interactions of antigen-specific CD8 T cells following LCMV infection using a newly developed universal version of the labeling immune partnerships by SorTagging intercellular contacts (uLIPSTIC) mouse model (Nakandakari-Higa et al., 2024, Nature). Although this model records past rather than current interactions and investigated slightly different time points, the types of interactions that antigen-specific CD8 T cells undergo can be compared. Notably, there was a high concordance in the types of interactions that antigen-specific CD8 T cells engage in within the uLIPSTIC model and our approach, including interactions with B cells, CD4 T cells, bystander CD8 T cells, NK cells, Neutrophils and other myeloid cells.
- Antigen-Specific T Cell Activation and Proliferation: Activation and proliferation of transferred antigen-specific T cells have been described to start in spleens around 3 days post LCMV infection and increase thereafter (Olson et al., 2012, PLOS Pathogens). A delay in kinetics has been described in non-draining lymph nodes compared to the spleen. Similarly, our approach identified initial interactions of antigen-specific T cells at day 3 post-infection, followed by a more pronounced increase at day 7. Consistent with Olson et al., the spleen showed more pronounced antigen-specific T cell interactions compared to non-draining lymph nodes at matching time points, likely due to more rapid kinetics.

Collectively, these observations suggest that while it cannot be unequivocally determined whether all measured interactions in the *Interact-omics* approach have occurred *in vivo*, the interactions are not random but reflect biological effects. The results align fully with those from *in situ* methods. Still, we have now acknowledged the limitations and outlined resulting guidelines on how to optimize our approach in the revised version of the manuscript.

The method is interesting, new and worthy, and some limitations are OK and expected but then should be stated and acknowledged clearly. The above questions may be hard to address fully, but should probably be addressed at least somewhat. E.g. how the results of the method will change after varying some parameters of sample preparation or tissue dissociation? Or some additional experimental manipulations with the sample, to varying degree, in the attempt

to break more of the cell-cell interactions, or to create more of artificial ones? The CytoStim or the antigen experiments address these questions but I think only very partially.

We thank this reviewer for this comment and fully agree. As described above, we have now performed a comprehensive benchmarking on experimental settings. These results indicate that varying parameters both of sample preparation, tissue dissociation and cytometric settings can have strong impact on the measured cell interactions. However, if these parameters are maintained constant, biological cellular interactions can be effectively quantified in case-control settings. As we fully agree with this reviewer on the significance of this topic, we have included the extensive *ex vivo* and *in vivo* benchmarking studies into the manuscript. Additionally, we included a section on limitations and guidelines that describes limitations but also provides insights into best practices on minimizing potential technical artifacts.

A related but different question is which of the detected cell-cell interactions are functional and which are just by random chance because the cells were next to each other even if *in vivo* (e.g. in a tissue)?

We agree with the reviewer that this is an important point. We have addressed this by conducting extensive *ex vivo* and *in vivo* mixing experiments with differentially labeled populations, as described above (see sections 1-3 and Figures R11, R12, R13). These experiments demonstrate that while new interactions can be acquired during sample preparation, these interactions are not random but reflect biological effects. Moreover, we have performed additional experiments and analyses that unequivocally show that the measured interactions are highly enriched for functional interactions rather than random ones. These include demonstrating that: i) Virus-specific T cells display significantly higher interactions with their predicted target cells in infected animals compared to on one hand bystander T cells and on the other hand what would be expected by chance (Fig. 6 of the manuscript). ii) Arthritis patients - which we ventured into in the framework of exploring cellular interacting mapping in publicly available datasets - exhibit an enrichment of cellular interactions associated with disease activity and display a unique phenotype, associated with reduced immune regulatory control (see response to the next reviewer's question; Fig. S13 of the manuscript).

To further demonstrate that cellular interactions acquired by our approach are not random but instead have a biological correlate, we evaluated whether the identified interacting cells exhibit intracellular signaling as a consequence of their interaction. For this purpose, we established a high-plex cytometry panel that includes an antibody detecting phosphorylation (pY142) of intracellular CD3 zeta (CD247), a transmembrane signaling adaptor protein phosphorylated upon T cell receptor signaling and T cell activation. Using this panel, we investigated intracellular TCR signaling in cellular interactions induced in human PBMCs upon CytoStim (which crosslinks antigen-presenting cells with T cells) and Blinatumomab (which crosslinks B cells with T cells) treatment (Fig. R14A). Consistent with our previous results and the molecular mechanisms of the inducers used, we observed few background interactions at homeostasis but noted specific induction of B - T cell interactions upon Blinatumomab treatment and broader myeloid and B cell interactions with T cells upon CytoStim treatment (Fig. R14B-D). As expected, CytoStim-induced interactions caused strong phosphorylation of the intracellular CD3 zeta domain in both T*B and T*Myeloid interactions, as well as in

T*B*Myeloid triplets, demonstrating functional T cell receptor engagement in the interacting T cells (Fig. R14 E, F). In line with its more specific cross-linking activities, Blinatumomab caused a specific increase in phosphorylation of the intracellular CD3 zeta domain in T cells involved in interactions with B cells, but to a much lower degree in interactions not involving B cells (Fig. R14 E, F).

Collectively, these findings demonstrate that our approach is capable of identifying interactions that reflect actual biological effects.

Figure R14. Characterization of downstream effects of functional interactions by assessment of phosphorylated CD247. **A.** Schematic overview of the experimental approach. PBMCs from 4 healthy donors were incubated in the presence or absence of Blinatumomab (Blina) or Cytostim (CS). **B.** UMAP of the overall cellular landscape. Recorded cells were processed with PICtR, out of 1,204,382 cells, 70,954 sketched cells are displayed. **C.** UMAP of interacting cells ($n = 52,239$) **D.** Point density UMAP of interacting cells split into the conditions control, Blinatumomab and Cytostim. **E.** Histograms of scaled fluorescence intensity of pCD247 for each interacting cell population from C. Left panel: T*B interactions. Middle panel: T*My interactions. Right panel: T*B*My. **F.** Mean fluorescence intensity of pCD247 per donor and condition. Left panel: T*B interactions. Middle panel: T*My interactions. Right panel: T*B*My. *P* values were determined with a paired t-test. Abbreviations: UMAP = uniform manifold approximation and projection, Ctrl. = control, Blina = Blinatumomab, CS = Cytostim. Red asterisks in cell type labels indicate interactions between the respective cell types.

The core of the method is application of their software PICtR to compensated and transformed FACS data. It is unclear if the software can detect physical cell interactions in any such previously published data from other experiments and other labs (if yes, it should be demonstrated), or some specific experimental details are important to make it work successfully (then these details should be clearly stated).

We thank the reviewer for raising this highly important point. Indeed, the PICtR workflow can be applied to existing datasets, provided that the data have been generated following the guidelines outlined in the manuscript. These guidelines include the inclusion of adequate control samples for relative case-control comparison, the use of a multi-color panel with cell type-discriminating markers, and the avoidance of experimental batch effects during sample preparation and acquisition (e.g., consistent flow rates, cell concentrations, voltages, etc.) (see also Limitations and Guidelines in the manuscript). To demonstrate the applicability of our approach for analyzing cellular interactions in previously generated datasets, we applied the PICtR workflow on a publicly available dataset of juvenile idiopathic arthritis (JIA) (Attrill et al., 2024, *Clinical and Experimental Immunology*) (Fig. R15).

JIA is an autoimmune disease characterized by chronic inflammation of the joints, leading to pain, swelling, and eventual joint damage. While it is hypothesized that an abnormal interaction among immune cells, specifically T cells, B cells, and myeloid cells, contributes to the production of inflammatory cytokines and autoantibodies that drive the disease, the actual interaction processes remain poorly understood.

Within this study, Attrill and colleagues analyzed PBMC samples from healthy donors, JIA patients with active and inactive disease as well as synovial fluid samples from JIA patients with active disease. The data is available on www.flowrepository.org under FLOWRepository ID FR-FCM-Z6VC. FCS files were downloaded and preprocessed as described in our Methods section. Additionally, flowAI (a QC algorithm) was run on all FCS files and FCS files with anomalies in their flow rate were excluded from further analysis. High quality samples were then analyzed with the PICtR workflow. Results described by Attrill et al. concerning the single-cell landscape could be reproduced (Fig. R15B). Focusing on the interacting cell landscape, we explored three comparisons: 1. PBMCs from healthy donors vs. PBMCs from patients with JIA (Fig. R15 D-F), 2. PBMCs from JIA patients with active disease vs. inactive disease (Fig. R15 G-I), 3. PBMCs from JIA patients with active disease vs. synovial fluid of JIA patients with active disease (Fig. R15 J-L). Notably, we discovered both quantitative and qualitative changes in cellular interactions in the blood of patients with inactive versus active disease, as well as between the blood and synovial fluid of affected joints (Fig. R15 D-L). Among the most intriguing findings is the enrichment of T cells interacting with B cells, which predominantly comprise a FoxP3-expressing regulatory T cell phenotype in patients with inactive disease (Fig. R15I). However, these interactions switch to an inflammatory, non-regulatory phenotype in patients with active disease. Similarly, major qualitative differences of interactions between CD4 T cells and monocytes were observed between blood and synovial fluid of patients with active disease (Fig. R15L). While several of these findings require further validation, they provide a first quantitative framework for understanding changes in immune cell interactions that may contribute to disease progression and help identify targets for therapeutic intervention.

Figure R15. Interacting cell landscape in juvenile idiopathic arthritis (JIA).

A. Schematic describing publicly available spectral flow cytometry data of PBMCs and SFMC of JIA patients (Attrill et al., 2024, Clinical and Experimental Immunology). Three comparisons (indicated by the arrows) were made for the interacting cell landscape **B**. UMAP of the overall cellular landscape. Recorded cells were processed with PICtR, out of 7,843,646 cells, 80,000 sketched cells are displayed. **C**. UMAP of interacting cells ($n = 12,908$) **D**. Point density UMAP (left panel) and differential abundance (right panel) of interacting cells comparing PBMCs from healthy donors vs. JIA patients. **E**. Quantitative comparisons of interacting cell frequencies between PBMCs from healthy donors ($n=18$) and JIA patients ($n=36$). Top: Non-adjusted frequencies. Bottom: Interaction frequencies normalized by the harmonic mean of the singlet frequencies of the contributing cells (see Methods). P values were determined with a two-sided t-test and adjusted for multiple testing using Benjamini-Hochberg correction. **F**. Qualitative differences in CD4T*cl.mono interactions. P values were determined with a two-sided Wilcoxon rank sum test and adjusted for multiple testing using Benjamini-Hochberg

correction. **G.** Point density UMAP (left panel) and differential abundance (right panel) of interacting cells comparing PBMCs of JIA patients with inactive disease (n=11) vs. active (n=25). **H.** Quantitative comparisons of interacting cell frequencies between PBMCs from JIA with inactive and active disease. Top: Non-adjusted frequencies. Bottom: Interaction frequencies normalized by the harmonic mean of the singlet frequencies of the contributing cells (see Methods). P values were determined with a two-sided t-test and adjusted for multiple testing using Benjamini-Hochberg correction. **I.** Qualitative differences in T*B interactions. P values were determined with a two-sided Wilcoxon rank sum test and adjusted for multiple testing using Benjamini-Hochberg correction. **J.** Point density UMAP (left panel) and differential abundance (right panel) of interacting cells comparing PBMCs of JIA patients with active disease vs. SFMC of active disease. **K.** Quantitative comparisons of interacting cell frequencies between PBMCs of JIA patients with active disease (n=25) vs. SFMC of active disease (n=8). Top: Non-adjusted frequencies. Bottom: Interaction frequencies normalized by the harmonic mean of the singlet frequencies of the contributing cells (see Methods). P values were determined with a two-sided t-test and adjusted for multiple testing using Benjamini-Hochberg correction. **L.** Qualitative differences in CD4T*mono interactions. P values were determined with a two-sided Wilcoxon rank sum test and adjusted for multiple testing using Benjamini-Hochberg correction. Abbreviations: UMAP = uniform manifold approximation and projection, PBMC = peripheral blood mononuclear cells, SFMC = synovial fluid mononuclear cells. Red asterisks in cell type labels indicate interactions between the respective cell types.

In the viral infection experiment, why aren't T cell-DC interactions found? Is it possible that the flow panel doesn't allow to distinguish between DCs and monocytes and macrophages? Also, I can see cDC2 annotation in Fig 5B, but not cDC1, though cDC1 are mentioned in Fig 5C.

We thank the reviewer for the question and would like to elaborate on this point. Our panel can differentiate between monocytes, macrophages, pDCs, cDC1s, and cDC2 singlet populations. However, annotating cellular interactions can be more challenging due to the mixed signals from interacting cells within a cluster. Therefore, depending on the cluster's representation of distinct types of interaction partners and the number of identified interactions, we made more fine-grained distinctions in some cases, while applying a more conservative annotation in others.

Regarding T cell-DC interactions, as expected, we identified interactions between antigen-specific CD4 T cells and DCs, specifically in the spleens of infected mice at day 7 post-infection (Fig. R16, Fig. S9). Please note that in the LCMV *in vivo* figures (Fig. 5 of the manuscript), only the most abundant populations are highlighted due to limited space available.

Figure R16. Frequency of antigen-specific CD4T*DC interactions across time and organs.

The frequency of antigen-specific CD4T*DC interactions per mouse were quantified across live cells and compared within the respective organ at three timepoints (naive, day3 post infection, day7 post infection). Abbreviations: BM = bone marrow, LN = lymph node.

Not sure if reporting relative frequencies of cell interactions (e.g. in Fig 5C) is relevant. Shouldn't it be normalized by frequencies of individual cell types which are probably also changing between time points?

We thank the reviewer for this thoughtful comment. We fully agree that there are several approaches to visualizing changes in cellular interactions, each with its own advantages and disadvantages. Overall, we provide three types of measures on cellular interactions:

1. **Relative frequencies of interactions among all interactions (e.g., Figure 5C):**
This provides a global understanding of how the composition of cellular interactions change over time.
2. **Relative frequencies of cellular interactions among all measured events (e.g., most of Figure 6):** This shows how prevalent certain interactions are in relation to all cells and other interactions.
3. **Normalized frequencies using the harmonic mean (e.g., Figure 6I):** This indicates the relative enrichment of frequencies as compared to expected random interactions based on the respective frequencies of the single interaction partners.

We have deliberately used all three measures to characterize the observed changes in cellular interactions. To provide a global overview of all observed changes, the display in Figure 5C, complemented by a heatmap on cluster frequencies in Figure S10 and Figure 6, appeared most fitting. In response to this reviewer's comment, we have now more explicitly explained the meaning and rationale of these different methods of analyzing changes in cellular interactions in the manuscript.

The exact Zenodo link to the software provided in the submission actually leads to an older version of the software, v2 dated Feb 2014. We tried to run this but it is not well documented and returns an error. However, there's v5 from Feb 27, 2024, which we tried to use to

reproduce some analysis in the manuscript, specifically the results for the LCMV experiment. It worked quite well. However, it was unclear how exactly subsampling (or sketching?) works, because following the instructions to the best of our ability resulted in 5000 cell analysis, but Fig 5 reports much larger numbers of cells. This Zenodo link to v5 includes the file PICtR-manual.pdf, which is indeed a manual of the R package. But it will be helpful to include some examples of running the analysis to reproduce at least some figures in the paper exactly, and walk the users through all steps (a notebook and step-by-step instructions, or what sometimes is called "vignette" for R packages).

We apologize for the confusion and have now updated the link. The demo data set included in the package is only a subset of the data in Supplementary Figure 4 since full data sets are too large and too computationally demanding for an example workflow. It contains a subset of the cells from group A and group B. (see ?demo_lcmv for details). Therefore, the demo only samples 5,000 cells (customizable through the `n_sketch_cells` parameter, defaults to 50,000). We now include step-by-step instructions in the form of a complete but preliminary vignette to walk users through the PICtR pipeline using this demo data set. A finalized version of the vignette is currently under development. Full data sets ensuring reproducibility of the figures can be found in the Zenodo repository.

Reviewer #3:

Remarks to the Author:

In this study, the authors developed Interactomics, which is a cytometry-based workflow that can detect multiplet interactions. In principle, it does fill an open niche in the investigation of immune cell interactions in high-dimensional space with high throughput. However, it is not deployed to its full potential in this manuscript at present. One could imagine performing a lower throughput or more time consuming approach such as PIC-seq or LIPSTIC that yields more information than using Interactomics. I find several key weaknesses with the manuscript at present: 1) A lack of mechanistic depth in any conclusions that are generated; 2) Concerns about the ab initio classification of the interactions detectable by the method; and 3) The long-term utility and flexibility of the method to detect more specific (and probably more valuable) interactions among more narrowly defined cell subsets. The strengths of the study are the authors' development of a creative platform that could be widely adopted, demonstration of proof-of-concept data across multiple mouse and human systems, and well-designed studies across time and stimulation. In general, if the authors can address the above concerns, I believe it could be a valuable tool. Additional points are listed below:

We thank the reviewer for his/her constructive feedback on our manuscript. We appreciate the recognition of the strengths in our approach and the valuable suggestions for improvement. In response to the reviewers' feedback, we have: 1) demonstrated how our approach can be expanded to provide mechanistic or biological insights, 2) clarified concerns regarding the ab initio classification of cellular interactions and performed comprehensive technical benchmarking of our method (see the final section of reviewer response), and 3) demonstrated its broad applicability across various model systems, tissues, and existing datasets, including the evaluation of more narrowly defined cell subset interactions. Additionally, we have addressed the specific points raised by the reviewer as outlined below.

1. The advantages over pre-existing approaches to study immune cell interactions are not clear. For example, "Interactomics" is entirely descriptive, while LIPSTIC and PIC-seq gain mechanistic information. Investigation and validation of the findings by Interactomics is needed in mouse and human systems.

We appreciate the reviewer's comment and would like to provide further clarification. We do not view the *Interact-omics* workflow as a "competitor" to methods such as LIPSTIC or PIC-seq, which offer high-resolution insights into past (LIPSTIC) and current (PIC-seq) cellular interactions. While these methods are powerful tools for mechanistically exploring cellular interactions, they are limited in throughput, by cost, and by the need for extensive preparations (e.g., mouse models for LIPSTIC and cell sorting of predefined interaction partners followed by sequencing for PIC-seq).

In contrast, the *Interact-omics* approach is a versatile, high-throughput, cost-effective, and straightforward workflow that can be readily adopted by any laboratory with cytometry experience using standard equipment and no additional prerequisites. The *Interact-omics* workflow excels at quantifying cellular interactions; it can generate quantitative datasets on potential changes in cellular interaction networks across all immune cells as a "byproduct" of a cytometry experiment. Additionally, it can be used to mine potential alterations in cellular interactions in existing datasets (see below). Specific interactions identified through this workflow can then be further investigated using more mechanistic, low-throughput approaches like LIPSTIC or PIC-seq. Furthermore, as outlined in our responses to the reviewer's questions below, the *Interact-omics* workflow panel can be adapted to investigate the functional associations of cellular interactions, such as activation, immune-regulation or intracellular phosphorylation states.

2. Additional validation of the functional implications of the interactions detected is needed on the level of intracellular signaling. Even if cell types are interacting, no mechanistic data are included in the manuscript to define that the interactions are controlling cellular physiology using this method. Many findings reported in the manuscript could be discerned using other approaches.

We agree with the reviewer that exploring the functionality of cellular interactions at the level of intracellular signaling is of high interest. To address this point, we established a high-plex cytometry panel that includes an antibody detecting phosphorylation (pY142) of intracellular CD3 zeta (CD247), a transmembrane signaling adaptor protein phosphorylated upon T cell receptor signaling and T cell activation. Using this panel, we investigated intracellular TCR signaling in cellular interactions induced in human PBMCs upon CytoStim (which crosslinks antigen-presenting cells with T cells) and Blinatumomab (which crosslinks B cells with T cells) treatment (Fig. R17A). Consistent with our previous results and the molecular mechanisms of the inducers used, we observed few background interactions at homeostasis but noted specific induction of B - T cell interactions upon Blinatumomab treatment and broader myeloid and B cell interactions with T cells upon CytoStim treatment (Fig. R17B-D). As expected, CytoStim-induced interactions caused strong phosphorylation of the intracellular CD3 zeta

domain in both T*B and T*Myeloid interactions, as well as in T*B*Myeloid triplets, demonstrating functional T cell receptor engagement in the interacting T cells (Fig. R17 E, F). In line with its more specific cross-linking activities, Blinatumomab caused a specific increase in phosphorylation of the intracellular CD3 zeta domain in T cells involved in interactions with B cells, but to a much lower degree in interactions not involving B cells (Fig. R17 E, F).

Collectively, these findings demonstrate the capability of our approach to study intracellular signaling in interacting cells. Additionally, in response to another enquiry from this reviewer, we show that interacting cells of patients with active autoimmune diseases acquire distinct cellular states, further evidencing the capacity of our approach to identify interactions that control cellular physiology (see below).

Figure R17. Characterization of functional interactions by assessment of phosphorylated CD247. **A.** Schematic overview of the experimental approach. PBMCs from 4 healthy donors were incubated in the presence or absence of Blinatumomab (Blina) or Cytostim (CS). **B.** UMAP of the overall cellular landscape. Recorded cells were processed with PICtR, out of 1,204,382 cells, 70,954 sketched cells are displayed. **C.** UMAP of interacting cells ($n = 52,239$). **D.** Point density UMAP of interacting cells split into the conditions. **E.** Histograms of scaled fluorescence intensity of pCD247 for each interacting cell population from C. Left panel: T*B interactions. Middle panel: T*My interactions. Right panel: T*B*My. **F.** Mean fluorescence intensity of pCD247 per donor and condition. Left panel: T*B interactions. Middle panel: T*My interactions. Right panel: T*B*My. P values were determined with a paired t-test. Abbreviations: UMAP = uniform manifold approximation and projection, Ctrl. = control,

Blina = Blinatumomab, CS = Cytostim. Red asterisks in cell type labels indicate interactions between the respective cell types.

3. One main limitation of the study is the focus on CytoStim as an inducer of "ground truth" cellular interactions. This approach would primarily label memory lymphocytes and APCs. The argument for developing this method is that it is more high throughput and generalizable than current approaches, but the training dataset undermines this. Additional "ground truth" interaction-driving approaches should be incorporated into the study and tested. Ideally, multiple antibody panels/stimuli that would detect distinct classes of interactions, compiled into a PICtR database, would make the method more robust.

We apologize for any confusion caused by the potentially misleading depiction of our approach. To clarify, Figure 1 illustrates how we employed imaging cytometry to obtain both ground truth data on cellular interactions (from imaging) and cytometric readouts from the same cells. This setup allowed us to derive general features, principles and rules to identify, discriminate, and quantify physically interacting cells from cytometry data without prior ground truth knowledge. Importantly, CytoStim was employed solely as a tool to induce a specific set of interactions, serving as a means to evaluate the specificity of our approach. The features identified and the methodology established are therefore independent of the interaction inducer and model system, and do not require prior training. With each experiment, interacting populations are identified based on clustering of multi-dimensional space, followed by employing the FSC ratio and presence of two or more mutually exclusive cell type markers for singlet to multiplet discrimination and cell type annotation.

Throughout the manuscript, we demonstrate the versatility of our approach across various model systems (e.g., mouse and human) and tissues (e.g., blood, bone marrow, spleen, lymph nodes, synovial fluid). During the revision process, we further benchmarked our approach against imaging flow cytometry in the context of the LCMV infection model presented in Figure 5, demonstrating high specificity and accuracy (cf. additional benchmarking at the end of the reviewer comments). To prevent future confusion, we have rephrased and restructured Figure 1 and the accompanying text to more clearly convey the methodology and its applications.

4. The CytoStim experiments would also induce artificial interactions between cell types, which may not reflect true physiology. A trained dataset focused on physiologic interactions during immunization, or equivalent, might make a better training dataset and be more comprehensive.

As depicted above, the *Interact-omics* approach operates in an agnostic manner and does not require prior training.

5. The algorithm was trained on human data, which was then applied to the mouse. How generalizable are interactions in the two systems to one another?

As depicted above, the *Interact-omics* approach operates in an agnostic manner and does not require prior training. We have now validated our approach both in humans (Fig. 1) and mouse using imaging cytometry (cf. additional benchmarking at the end of the reviewer comments).

6. Almost all experiments are performed *in vitro* or from PBMCs, so while the throughput is increased, the relevance is somewhat diminished. To showcase the power of this tool, the authors could investigate additional fluids not tractable with other tools, and glean new insights not previously known, as they mentioned in the introduction.

We thank the reviewer for raising this point and fully agree with the importance of demonstrating a broad applicability across different tissues and systems. In the revised version of the manuscript (including both original and new analyses), we demonstrate the successful applicability of our approach for human primary blood (Figs. 2A-E, 3F-K, S4, S8), bone marrow (Fig. 4) and synovial fluid (Fig. S13), as well as for murine spleens (Figs. 2F-J, 3A-E, 5, 6, S12), lymph nodes (Fig. 5, 6) and bone marrow (Fig. 5, 6), and in diverse *ex vivo* and *in vivo* settings. As part of the response to this reviewer's 9th comment, we elaborate in more detail the novel findings and applications that can be derived from the revised version of the manuscript.

7. An important feature of this approach is to delineate specific molecular insights that are not possible with conventional flow cytometry. However, the authors only capitalized on broad cell lineage markers. How well are interactions actually detected and validated when a panel of markers is used that is focused on subsets of intra-lineage cells? For example, can Interactomics be applied to CD4 Th cell subset differentiation?

We thank the reviewer for raising this important question. Our approach is intentionally designed with a broad backbone panel of primarily cell type-specific markers, enabling the quantitative measurement of cellular interactions across all major immune cell types. However, additional markers for intra-lineage differentiation can be easily incorporated to provide more detailed annotation of immune cell subsets. To practically demonstrate this, we applied the PICtR workflow to a published ultra-high plex Arthritis cytometry dataset (Fig. R18A-C, see response to reviewer comment 10) and an unpublished inflammatory bowel disease dataset from our laboratory (Fig. R18D-F), both of which include a sufficient number of cell type-specific and subset-specific markers. These analyses reveal that interactions can be resolved at the intra-lineage level if the respective markers are included. Examples include interactions of CD16 vs. CD56^{bright} NK cell subsets (Fig. R18B), CD14 classical vs. CD16 non-classical monocytes, intracellular measurements of FoxP3 to identify interactions involving regulatory T cells (Fig. R18C), B cell subsets characterized by the presence or absence of IgD (Fig. R18E, F), and activation and regulatory markers to identify functional immune cell subsets. The approach is flexible and can be fine-tuned for specific research questions.

Figure R18. Intra-lineage characterization of interacting cells in two different medical conditions. **A.** UMAP of interacting cells ($n = 12,908$) from juvenile idiopathic arthritis (JIA) patients. **B.** Mean scaled fluorescence intensity of CD16 (left boxplot) and CD56 (right boxplot) in B*NK interactions. P values were determined with a two-sided Wilcoxon rank sum test. **C.** Mean scaled fluorescence intensity of FoxP3 in T*B interactions compared between JIA patients with active (act.) vs. inactive (inact.) disease. P value was determined with a two-sided Wilcoxon rank sum test. **D.** UMAP of interacting cells ($n = 40,938$ out of 422,073 cells) from juvenile inflammatory bowel disease (IBD) patients. **D.** Point density UMAP of interacting cells split into the conditions. **E.** Mean scaled fluorescence intensity of IgD compared in IgD+ vs. IgD- B*CD4T cell interactions. P value was determined with a two-sided Wilcoxon rank sum test. **F.** Mean scaled fluorescence intensity of IgD compared in IgD+ vs. IgD- B*classical monocyte interactions. P value was determined with a two-sided Wilcoxon rank sum test. Abbreviations: UMAP = uniform manifold approximation and projection, cl.mono = classical monocytes, noncl.mono = non-classical monocytes, neutro = neutrophils, NK = natural killer cells.

8. The inability to detect homotypic interactions is a weakness. If different starting panels of antibodies were used to train the parameters of the system, would different interactions be prioritized as detected in test datasets? For example, if the 24-plex panels contained much more specific markers for B cell subsets, could these interactions be detected?

As detailed above, while we developed the concept and features for distinguishing singlets from multiplets through a systematic comparison with ground truth imaging cytometry data, any subsequent analyses are not based on the dataset from Figure 1. Instead, analyses can be carried out on any pre-existing or novel dataset, assuming some guidelines are met (see Limitations and Guidelines in the revised manuscript). The antibody panels can be adapted according to the user's specific research question.

To address the reviewer's question about how modifying the antibody panel affects the resolution of cellular interactions, we systematically investigated how reducing or extending the antibody panel impacts performance in dissecting interactions. For this purpose, we conducted an analysis where we progressively identified the least important feature (marker) and reduced the number of features by sequentially removing the least significant one from the input to the workflow (Fig. R19; item A for the data underlying main Figure 2A—cellular interactions in human PBMCs after CytoStim—and item B for the data underlying main Figure 5B—virus-induced cellular interaction networks in mice—as representative examples). We then compared the results obtained from the reduced and full feature sets using the adjusted Rand index. As shown in Figure R19, the adjusted Rand index remains consistent within a range of approximately 15 to 25 features, demonstrating robustness to the inclusion or exclusion of individual features for identifying more broadly defined cell type interactions.

Nonetheless, as this reviewer rightly noted, the presence of specific markers is crucial for identifying interactions among distinct immune cell subsets, as addressed in the response to the previous comment. Similarly, adding more markers specific for subsets of a single lineage could potentially resolve homotypic interactions among sub-lineage cell types within the same broader lineage. However, since homotypic interactions lack the co-expression of mutually exclusive main lineage markers (e.g. double positive of CD3 and CD19), the resulting annotations of homotypic interactions may remain less reliable. As a result, we have opted not to annotate homotypic interactions in the examples presented in the manuscript. That said, if of particular interest, custom panel adaptations or future extensions to the computational framework could facilitate the study of these interactions.

Figure R19. Comparison of adjusted Rand index across different numbers of features. **A.** Feature importance analysis for data from Figure 2A. The adjusted Rand index was calculated after iteratively removing the least important feature. **B.** Feature importance analysis for data from Figure 5B. The adjusted Rand index was calculated after iteratively removing the least important feature.

9. Almost all findings are descriptive. If only a fraction of interactions are detectable by Interactomics, and there is no *in vivo* validation of the method, it is hard to take much new information from the study. At least one substantial finding detected by Interactomics should be validated *in vivo* (or in a related human system).

As this manuscript is a methods paper, the primary focus has been on showcasing the broad applicability of the technique rather than presenting a specific new finding. Nonetheless, our study introduces several novel aspects that are of high relevance to both science and medicine, as outlined in the following:

1. **Characterization and therapy response prediction of Blinatumomab.**

Immunotherapies, such as Blinatumomab, have revolutionized the treatment of certain cancers. However, therapy resistance remains a major hurdle. Deciphering the causes underlying therapy resistance and developing predictive assays has the potential to 1) redirect patients that would benefit from alternative therapies, to 2) save substantial financial resources to public health care systems and to 3) expand Blinatumomab treatment to other B cell cancers beyond ALL. Our study provides an important step towards these goals by confirming previous features associated with therapy resistance, and, for the first time, demonstrating that frequencies of cellular interactions prior to treatment or upon induction with Blinatumomab can serve as predictive biomarkers for therapy response and failure. Specifically, our findings suggest that Blinatumomab fails to induce effective B - T cell interactions in patient samples with unbalanced T cell to B cell ratios and provides quantitative limits for the definition of “unbalanced”, and that T - myeloid interactions at baseline are associated with therapy failure. Most notably, our data suggest that the efficiency of Blinatumomab induced B - T interactions *ex vivo* in patient samples prior treatment may be used to predict actual response. While the presented results still require additional validation, they have been performed on a substantial number of patients (n = 42). Based on these results, we are currently engaged in a larger effort with clinical centers and industry for designing personalized response prediction tools.

2. **Organism-wide cellular interaction mapping of virus infection.** Infectious agents and pathogens induce complex cascades of organ-specific immune reactions *in vivo*, comprising cell-cell interactions, cell expansion and cellular trafficking, jointly establishing first line defense as well as long-lasting adaptive immunity. However, the comprehension in the scientific community of such pathogen-induced cellular immune dynamics remains poorly characterized. In particular, there is a lack of quantitative insights into organotypic differences in the composition, order and kinetics of cellular interactions induced upon pathogen exposure *in vivo*. Complementary to a paper published during the revision process at Nature that investigated past interaction of CD8 T cells (Nakandakari-Higa et al., 2024) using an elegant mouse model-based approach (uLIPSTIC), we here have unveiled a quantitative cellular interactions network among immune cells across various time points and organs. This approach validates previously reported interactions, quantifies organ-specific differences in kinetics and interaction patterns, and provides a framework to study cellular interaction affinities organism-wide.

3. **Quantitative and qualitative alterations of cellular interactions in arthritis.** Juvenile idiopathic arthritis (JIA) is an autoimmune disease characterized by chronic

inflammation of the joints, leading to pain, swelling, and eventually joint damage. While it is hypothesized that abnormal interactions among immune cells, specifically T cells, B cells, and myeloid cells, contribute to the production of inflammatory cytokines and autoantibodies that drive the disease, the actual interaction processes remain poorly understood. As part of the response to this reviewer's 10th comment (see below), we have applied the PICtR workflow on a large published JIA dataset and discovered both quantitative and qualitative changes in cellular interactions in the blood of patients with inactive versus active disease and between blood and synovial fluid of affected joints. Among the interesting findings is that T cells interacting with B cells are highly enriched in FoxP3-expressing regulatory T cells in patients with inactive disease, but display an inflammatory, non-regulatory phenotype in patients with active disease. While several of those findings require additional validations, these data provide a first quantitative framework for understanding changes in cellular interactions of immune cells that might contribute to disease and may help to identify targets for therapeutic intervention.

In summary, we believe that our revised manuscript provides numerous new biological and clinically relevant insights and approaches. In general, we acknowledge that some points require further exploration beyond this study, as the primary focus of this method paper was to demonstrate broad applicability across various model systems, diseases, and clinical applications.

10. Can PICtR reevaluate cellular interactions in previously reported datasets? It would be valuable to mine new insights from previous datasets in an agnostic way with orthogonal antibody panels.

We thank the reviewer for raising this highly important point. Indeed, the PICtR workflow can be applied to existing datasets, provided that generation of the data is compliant with minimum requirements as described in the guidelines outlined in the manuscript. These guidelines include the inclusion of adequate control samples for relative case-control comparison, the use of a multi-color panel with cell type-discriminating markers, and the avoidance of experimental batch effects during sample preparation and acquisition (e.g., consistent flow rates, cell concentrations, voltages, etc.) (see "Limitations and Guidelines" of the manuscript). To demonstrate the applicability of our approach for analyzing cellular interactions in previously generated datasets, we applied the PICtR workflow to a publicly available dataset of juvenile idiopathic arthritis (JIA) (Attrill et al., 2024, *Clinical and Experimental Immunology*) (Fig. R20). JIA is an autoimmune disease characterized by chronic inflammation of the joints, leading to pain, swelling, and eventually joint damage. While it is hypothesized that an abnormal interaction among immune cells, specifically T cells, B cells, and myeloid cells, contributes to the production of inflammatory cytokines and autoantibodies that drive the disease, the actual interaction processes remain poorly understood.

Within this study, Attrill and colleagues analyzed PBMC samples from healthy donors, JIA patients with active and inactive disease as well as synovial fluid samples from JIA patients with active disease. The data is available on www.flowrepository.org under FLOWRepository ID FR-FCM-Z6VC. FCS files were downloaded and preprocessed as described in our Methods section. Additionally, flowAI (a QC algorithm) was run on all FCS files and FCS files with anomalies in their flow rate were excluded from further analysis. High quality samples were then analyzed with the PICtR workflow. Results described by Attrill et al. concerning the single-cell landscape could be reproduced (Fig. R20B). Focusing on the interacting cell landscape,

we explored three comparisons: 1. PBMCs from healthy donors vs. PBMCs from patients with JIA (Fig. R20D-F), 2. PBMCs from JIA patients with active disease vs. inactive disease (Fig. R20G-I), 3. PBMCs from JIA patients with active disease vs. synovial fluid of JIA patients with active disease (Fig. R20J-L). Notably, we discovered both quantitative and qualitative changes in cellular interactions in the blood of patients with inactive versus active disease, as well as between the blood and synovial fluid of affected joints (Fig. R20 D-L). Among the most intriguing findings is the enrichment of T cells interacting with B cells, which predominantly comprise a FoxP3-expressing regulatory T cell phenotype in patients with inactive disease (Fig. R20I). However, these interactions switch to an inflammatory, non-regulatory phenotype in patients with active disease. Similarly, major qualitative differences of interactions between CD4 T cells and monocytes were observed between blood and synovial fluid of patients with active disease (Fig. R20L). While several of these findings require further validation, they provide a first quantitative framework for understanding changes in immune cell interactions that may contribute to disease progression and help identify targets for therapeutic intervention.

Figure R20. Interacting cell landscape in juvenile idiopathic arthritis (JIA).

A. Publicly available spectral flow cytometry data of PBMCs and SFMC of JIA patients (Attrill et al., 2024; Clinical and Experimental Immunology). Three comparisons (indicated by the arrows) were made for the interacting cell landscape **B.** UMAP of the overall cellular landscape. Recorded cells were processed with PICtR, out of 7,843,646 cells, 80,000 sketched cells are displayed. **C.** UMAP of interacting cells ($n = 12,908$) **D.** Point density UMAP (left panel) and differential abundance (right panel) of interacting cells comparing PBMCs from healthy donors vs. JIA patients. **E.** Quantitative comparisons of interacting cell frequencies between PBMCs from healthy donors and JIA patients. Non-adjusted frequencies (top panel) and adjusted for singlet frequencies (bottom panel) are displayed. P values were determined with a two-sided t-test and adjusted for multiple testing. **F.** Qualitative differences in CD4T*cl.mono interactions. P values were determined with a two-sided Wilcoxon rank sum test and adjusted for multiple testing. **G.** Point density UMAP (left panel) and differential abundance (right panel) of interacting cells comparing PBMCs of JIA patients with inactive

disease vs. active. **H.** Quantitative comparisons of interacting cell frequencies between PBMCs from JIA with inactive and active disease. Non-adjusted frequencies (top panel) and adjusted for singlet frequencies (bottom panel) are displayed. *P* values were determined with a two-sided t-test and adjusted for multiple testing. **F.** Qualitative differences in T*B interactions. *P* values were determined with a two-sided Wilcoxon rank sum test and adjusted for multiple testing. **J.** Point density UMAP (left panel) and differential abundance (right panel) of interacting cells comparing PBMCs of JIA patients with active disease vs. SFMC of active disease. **K.** Quantitative comparisons of interacting cell frequencies between PBMCs of JIA patients with active disease vs. SFMC of active disease. Non-adjusted frequencies (top panel) and adjusted for singlet frequencies (bottom panel) are displayed. *P* values were determined with a two-sided t-test and adjusted for multiple testing. **L.** Qualitative differences in CD4T*mono interactions. *P* values were determined with a two-sided Wilcoxon rank sum test and adjusted for multiple testing. Abbreviations: UMAP = uniform manifold approximation and projection, PBMC = peripheral blood mononuclear cells, SFMC = synovial fluid mononuclear cells. Red asterisks in cell type labels indicate interactions between the respective cell types.

11. Even if multiplets are detected in flow cytometry data, it is not clear how they are differentiated as "real" interactions and not simply noise, for example, cells clumping within a tube for technical reasons. Controls guarding against the possibilities of technical contamination would help enrich for meaningful signals.

We fully agree with this comment and recognize the importance of adequate controls. In our proposed approach, interactions that reflect actual biological effects are always derived as statistical enrichments compared to control settings using the same experimental and technical framework (case-control setting). Throughout the revised manuscript, we have demonstrated, across multiple settings, diseases, and model systems, that this approach can derive plausible results that reflect actual biological effects.

Nonetheless, we agree with this reviewer that systematically evaluating technical limitations is crucial for interpreting the results of the proposed methods. Therefore, we have conducted a series of *ex vivo* and *in vivo* benchmarking experiments to thoroughly assess the applicability and limitations of our approach, including analyses of the functionality and specificity of cellular interactions.

1. *Ex vivo* cellular mixing experiment

To assess the extent to which additional cellular interactions are formed during culture and how different experimental settings impact this process, we conducted an extensive *ex vivo* benchmarking experiment involving labeling and mixing of cell populations. We first induced cellular interactions in PBMCs with CytoStim or left cells untreated. The PBMCs were then split and labeled with two different fluorescently conjugated antibodies against CD45 (Fig. R21A). Subsequently, the distinctively labeled PBMCs were reunited and processed at varying cell concentrations, processing times, and fixation methods, followed by mapping cellular interactions using our approach. This allowed us to effectively identify singlets positive for each label (Fig. R21B, C), as well as cellular interactions that were either single- or double-positive for both labels (Fig. R21D-F). Double-positive cellular interactions for the two introduced labels were newly acquired during the second incubation period, while single-

positive interactions could have been acquired initially or throughout the culture period. Focusing on B-T cell interactions, as one of the most frequent interactions induced by CytoStim, our experiments revealed that single-positive interactions increased robustly upon CytoStim treatment. In contrast, double-positive (newly acquired) interactions showed only a mild increase, which appeared negligible compared to the induction of single-positive interactions (approximately 5-10-fold higher in single positives compared to double positives) (Fig. R21G). Increasing incubation periods post-CytoStim did not cause a stark increase in newly acquired (double-positive) interactions, while a trend toward a decrease in CytoStim-induced (single-positive) B-T cell interactions was observed (Fig. R21G). Even at 4 hours post-incubation, CytoStim-induced single positives were on average 7-fold more frequent than their double-positive counterparts. These data suggest that *ex vivo* induced cellular interactions are relatively stable, and newly acquired interactions occur but have a relatively minor impact in this setting. With increasing cell concentrations, a mild baseline increase across all interactions (both single and double positives, with and without CytoStim) was observed (Fig. R21G). However, the relative effect of CytoStim-induced single-positive interactions compared to newly acquired double-positive interactions remained comparable. These data reinforce the importance of maintaining stable cell concentrations within experimental settings but also demonstrate that *ex vivo* induced cellular interactions can be effectively measured across distinct concentrations, providing stability across conditions. Finally, we evaluated the impact of chemical fixation on newly formed *ex vivo* interactions. PBMCs were either fixed or left unfixed after +/- CytoStim treatment, labeling, reuniting and staining. Overall, only a minor impact of fixation was observed, with a trend towards higher single and double positive interactions upon CytoStim treatment in fixed samples (Fig. R21G). This experiment suggests that cellular interactions can be retrieved using fixation or leaving samples in their native state, provided the same conditions are applied to all samples. We recommend, however, to fix samples as early as possible during the sample preparation workflow. Collectively, these data provide quantitative insights into how experimental settings impact background interactions while demonstrating that *ex vivo* modulations of cellular interactions can be effectively quantified in the explored settings. Besides incubation times, cell numbers, and fixation, flow rates significantly impact baseline interactions, as previously demonstrated (Fig. R21H-J). Therefore, maintaining these parameters constant maximizes the recovery of signal to background interactions. We have now included a limitations and guidelines section in the manuscript detailing how to optimize the experimental setting for *ex vivo* cytometry-based cellular interaction mapping.

Figure R21. Effect of sample processing methods on ex vivo cellular interactions.

A. Schematic depiction of the experimental approach. **B.** Overall cellular landscape across all experimental conditions. $n = 3,292,837$. **C.** Feature plots for the UMAP display in B, colored by the two differently labeled antibodies against CD45. **D.** Interacting landscape across all experimental conditions. $n = 23,620$. **E.** Dot plot for the CD45 signals in interacting populations, showcasing single-positive and double-positive interactions. **F.** Feature plots for the UMAP from panel D, colored by the CD45 signal intensities. **G.** Quantification of single-positive and double-positive, newly acquired interactions in CytoStim™ treated and untreated samples, across different experimental conditions. Left: Varying incubation times at 4°C after mixing, mimicking long sample processing times. Middle: Different cellular concentrations ranging from 25,000 to 250,000 cells in 50 μ L during staining and acquisition. Right: Fixation with 2 % paraformaldehyde after staining compared to no fixation. The number of replicates is shown in each plot and ranges from $n = 2$ to 3. **H.** Short-term cultures of PBMCs with or without CytoStim™ measured at low or high flow rate ($n = 4$). Impact of flow rate on cellular interactions is relatively mild. Two-way ANOVA (CytoStim™: $F(1,13) = 189.138$, $P = 4.01e-09$, flow rate: $F(1,13) = 6.598$, $P = 0.023$), followed by Tukey's Honest Significant Differences test for the flowrate. **I.** Impact of flow rate and cellular concentration on cellular interactions in murine

spleens is more pronounced. Left: Baseline interactions in spleens measured with different flow rates ($n = 4$). Right: Baseline interactions in spleens at different cell densities but constant flow rate ($n = 4$). One-way ANOVA (flow rate: $F(2,9) = 115.749$, $P = 3.79e-07$, cell concentration: $F(2,9) = 61.397$, $P = 5.68e-06$), followed by Tukey's Honest Significant Differences tests. J. Boxplots of T*B cell interactions upon Blinatumomab treatment that were either fixed or fixed after a freeze-thaw cycle ($n = 4$). P values were determined with a two-sided Welch's t-test.

2. *In vivo* benchmarking to imaging flow cytometry

To provide additional information on the reliability of interactions assessed in particular *in vivo* settings, we performed ImageStream-based imaging flow cytometry of LCMV-infected spleens at day 7 post-infection, similar to our experimental setup for organism-wide cytometry-based cellular interaction mapping presented in the manuscript (Fig. R22). While imaging cytometry requires specialized instruments, has a low cellular throughput, and is limited in the number of measured fluorochromes and cell types that can be resolved, it provides morphological imaging information that can classify cells into single versus interacting cells. Due to the limited number of available channels, we utilized a 6-plex panel focusing on T and B cell interactions. Upon data generation, we extracted the fluorescence intensity values and applied the PICtR workflow without considering any morphological information (Fig. R22A-C). As the FSC ratio cannot be extracted during the ImageStream workflow, clustering and interaction identification was performed on fluorescence channel values only. Consequently, the results presented here likely underestimate the performance of the *Interact-omics* approach. We then evaluated the performance of the *Interact-omics* approach with information gained from both images and fluorescence values regarding (i) discriminating singlets from interacting cells, (ii) identifying cell type combinations of interactions, and (iii) deriving qualitative changes in LCMV-induced interactions. For this purpose, we first compared the results of the *Interact-omics* workflow to an image segmentation-based classification of singlets and multiplets, demonstrating a high concordance between the approaches regarding the discrimination between single cells and multiplets (Fig. R22D). While neither image-based segmentation nor our approach reflects ground truth data, the high concordance strongly supports the validity of the *Interact-omics* results. Manual inspection of randomly selected images predicted as singlets or interacting cells further verified the accuracy of the *Interact-omics* approach (Fig. R22C).

Next, we immunophenotypically characterized interacting populations using conventional gating on marker expression values extracted from images (Fig. R22E). As expected, this revealed a high concordance with the cluster-based annotations from the *Interact-omics* workflow. Manual inspection of randomly selected images regarding the localized expression patterns of lineage-specific markers confirmed the expected types of interactions (Fig. R22C). Finally, we compared the LCMV-induced alterations of cellular interactions as detected by *Interact-omics* vs. the imaging flow cytometry approach. Due to the low plexity and low cellular throughput of the imaging flow cytometry approach, only very crude interacting cell populations could be defined, restricting the comparison to the populations depicted in Fig. R22B. Nonetheless, interacting cell populations identified with both approaches showed qualitatively matching alterations induced by LCMV, including a marked increase in interactions involving antigen-specific CD4 T cells (Fig. R22F).

Collectively, these data confirm the accuracy of the *Interact-omics* approach in discriminating single and interacting cells and in quantifying the types of interactions in case-control settings.

Figure R22. Comparison of the *Interact-omics* approach to imaging flow cytometry.

A. UMAP representation of the overall cellular landscape derived from the fluorescent intensity values, $n = 306538$. Intensity values are based on the sum of the pixel intensities in the mask as selected by ImageStream[®], background subtracted. The experiment corresponds to day 7 in Supplementary Figure 11. **B.** UMAP representation of the interacting landscape, $n = 8683$. The heterogeneous cluster is mostly comprised of likely B*CD4*CD8 multiplets. The unknown cluster expresses CD19 and CD3 but no other T cell markers, hindering confident annotation. **C.** Pseudo-colored example images for cellular interactions in the brightfield and fluorescence channels. **D.** Left: UMAP displays from A and B colored by the number of cells identified through image segmentation. Right: Bar plots comparing the populations identified through *Interact-omics* and the image segmentation. **E.** Left: UMAP displays from A and B colored by populations as identified through conventional gating. Right: Bar plots comparing the

populations identified through *Interact-omics* and conventional gating. NA indicates that the event does not fall into any conventional gate. **F.** Fold changes of the frequencies +/- LCMV infection. Holm-corrected estimated marginal means comparison. Left: Populations identified by *Interact-omics*. Right: Populations identified through conventional gating. **G.** Gating strategy for conventional gating. Abbreviations: Ag = antigen-specific, UMAP = uniform manifold approximation and projection, BF = brightfield.

3. *In vivo* mixing experiments to characterize newly acquired cellular interactions

The presented approach measures cellular interactions following sample preparation *ex vivo*. Consequently, for *in vivo* applications, additional cellular interactions can be acquired or lost during sample preparation. While this limitation applies to all cellular interaction mapping approaches that do not rely on specialized mouse models or measure co-localization *in situ*, it remains poorly characterized to what extent this occurs, whether newly acquired interactions are random or directed, and how representative the identified interactions are of the *in vivo* situation.

To evaluate these questions, we utilized congenic mouse models differing in variants of the pan-hematopoietic cell marker CD45, allowing identification of respective immune cells as CD45.1 or CD45.2 using variant-sensitive antibodies. First, we transferred LCMV-specific CD4 T cells (SMARTA: CD90.1 positive, CD45.2 positive) into CD45.2 mice, followed by LCMV infection (group A) (Fig. R23A). Non-infected control CD45.2 mice were included (group B). In parallel, we infected CD45.1 mice with LCMV (group C) or left them untreated (group D). On day 7 post-infection, spleens from all groups were harvested. Spleens from group A (infected, CD45.2) and group B (non-infected, CD45.2) were either processed individually or mixed in 1:1 ratios with spleens from group C (infected, CD45.1) or group D (non-infected, CD45.1) before tissue homogenization and processing. Applying the *Interact-omics* workflow to these single and mixed samples resulted in single-cell and interacting cell landscapes of populations that were either single positive or double positive for CD45.1 and CD45.2 (Fig. R23B-E), with double-positive populations being newly acquired interactions during processing. Notably, we observed a substantial acquisition of new interactions during sample processing (Fig. R23F). However, these newly acquired interactions did not occur randomly but were highly correlated with interactions induced upon infection (Fig. R23G). In particular, newly acquired interactions in mixed spleens from infected mice compared to non-infected controls were highly correlated with infection-induced single positive interactions in both mixed and non-mixed spleens (Fig. R23G). This suggests that while new interactions can be acquired during sample preparation, they are not random but reflect actual biological effects and likely are a proxy for cellular interactions occurring *in vivo* (see also next section about qualitative comparison to *in situ* methods).

Figure R23. Interactions acquired a posteriori after *in vivo* experiment.

A. Schematic overview of the experimental approach. LCMV-specific CD4+ T cells were transferred into CD45.2 host mice 5 days before infection with LCMV (group A) or the respective control (group B). Additionally, CD45.1 host mice were infected with LCMV (group C) or left untreated (group D). $n = 3$ for groups A, C, D and $n = 4$ for group B. **B.** Single-cell landscape of all experimental groups. Out of $n = 23,490,812$ processed cells, $n = 245,316$ are shown in the UMAP display. **C.** Feature plots for panel B, colored by the expression of the congenital markers CD45.1 and CD45.2. **D.** Interacting landscape across all experimental groups. Out of $n = 731621$ identifiable interactions, $n = 93065$ are shown. **E.** Dot plots showcasing the expression of the congenital markers CD45.1 and CD45.2 in interacting populations from the unmixed controls for the untreated and infected conditions, and the mixed spleens from infected mice (infected+infected, group A + group C) or untreated mice (control+control, group B + group D). **F.** Bar plots depicting the log₂ fold changes (FC) between the LCMV infected and untreated conditions for each interacting population. Solid bars indicate the log₂FC between group A (infected) and group B (control). Semi-transparent bars show the log₂FC for single-positive interactions in mixed samples (A+C for the infected condition, and

B+D for the control). Transparent bars depict log₂FC between the respective double-positive interactions, which were definitely acquired *ex vivo*. $n = 3$ mixes. **G.** Linear relationships between the log₂FC between infected and control conditions for unmixed controls, single-positive interactions after mixing and double-positive interactions after mixing. $n = 3$ mixes.

4. Qualitative comparison to *in situ* methods

To further explore whether *in vivo*-derived interactions measured using our approach resemble those occurring *in situ*, we qualitatively compared our LCMV *Interact-omics* results with imaging and *in situ* interaction mapping of approaches investigating LCMV infections:

- Monocyte–B Cell Interactions: Imaging has identified a drastic increase in monocyte–B cell interactions following LCMV infection (Sammicheli et al., 2016, Science Immunology). In line with this, our approach also identified a major increase in monocyte–B cell interactions upon LCMV infection, matching the kinetics described by Sammicheli et al.
- CD8 T Cell Interactions: A recent study mapped the interactions of antigen-specific CD8 T cells following LCMV infection using a newly developed universal version of the labeling immune partnerships by SorTagging intercellular contacts (uLIPSTIC) mouse model (Nakandakari-Higa et al., 2024, Nature). Although this model records past rather than current interactions and investigated slightly different time points, the types of interactions that antigen-specific CD8 T cells undergo can be compared. Notably, there was a high concordance in the types of interactions that antigen-specific CD8 T cells engage in within the uLIPSTIC model and our approach, including interactions with B cells, CD4 T cells, bystander CD8 T cells, NK cells, Neutrophils and other myeloid cells.
- Antigen-Specific T Cell Activation and Proliferation: Activation and proliferation of transferred antigen-specific T cells have been described to start in spleens around 3 days post LCMV infection and increase thereafter (Olson et al., 2012, PLOS Pathogens). A delay in kinetics has been described in non-draining lymph nodes compared to the spleen. Similarly, our approach identified initial interactions of antigen-specific T cells at day 3 post-infection, followed by a more pronounced increase at day 7. Consistent with Olson et al., the spleen showed more pronounced antigen-specific T cell interactions compared to non-draining lymph nodes at matching time points, likely due to more rapid kinetics.

Collectively, these observations suggest that while it cannot be unequivocally determined whether all measured interactions in the *Interact-omics* approach have occurred *in vivo*, the interactions are not random but reflect biological effects. The results align fully with those from *in situ* methods. Still, we have now acknowledged the limitations and outlined resulting guidelines on how to optimize our approach in the revised version of the manuscript.

Minor points:

1. Biological replicates are not mentioned in at least some figure legends.

We have now depicted the number of used replicates in each figure legend.

2. Fig. 1E it is not clear how a My*⁺T*B interaction (3 cells) could be represented by a doublet or singlet. Same with other interactions in that panel. It should be shown more clearly.

We apologize for the confusion. Our approach quantifies all selected live events acquired by flow cytometry. The vast majority of these events are typically single cells, with fewer interacting cells comprising two partner cells (doublets), and, with decreasing frequency, triplets and higher-order multiplets, depending on the sample source and processing. Using distinct types of imaging cytometry, we have demonstrated that our approach can correctly identify multiplets, though the confidence of the order of multiplicity decreases with higher-order interactions. We have now more explicitly clarified this in the manuscript.

3. In Fig. 3K there is not information learned, since the bifunctional antibody simply induces interactions mostly non-specifically.

Blinatumomab crosslinks CD19, a pan B cell marker, and CD3, which is expressed across T cell subsets. Figure 3C demonstrates our approach's ability to accurately depict the specificity and kinetics of Blinatumomab-induced cellular interactions across various T cell subsets with B cells. Consistent with this, we observe a rapid induction of interactions between B cells and CD4 T cells, CD8 T cells, gamma-delta T cells, and NK T cells, while other interactions remain unchanged, demonstrating the technical soundness of the approach. The additional information introduced in Figure 3K is in particular the time course and precise quantification of the time intervals for increase in and half-life of induced cellular interactions. Conceptually new points are presented in Figure 4.

4. In Fig. 5E, the distance in PCA space analysis is unusual. A different metric should be used.

We appreciate the reviewer's comment and would like to explain the rationale behind our approach. The goal of Figure 5E is to quantitatively summarize similarities and differences in abundances of single cells and cellular interactions across various organs and time points. Capturing such high-dimensional relationships concisely is challenging. PCA leaves Euclidean distances invariant and is thus well-suited for assessment of global similarities or differences in single-cell and interaction landscapes. We have now more explicitly stated the rationale for this approach in the figure legends.

5. The text formatting reads like a single long paragraph and would be more readable if broken into demarcated sections.

We have now adapted the manuscript employing shorter paragraphs to make it more readable.

6. The name of the method "Interactomics" should be changed as many methods at this point perform interactomics. A more specific name should be used.

The name was originally chosen due to its ultra-high throughput and capacity to map interactions across a wide range of cell types simultaneously. Alternatively, we could refer to the method simply as cytometry-based cellular interaction mapping. However, we are hesitant to change the name without prior consultation with the editor and will defer to their discretion for the final decision.

Second point-by-point reply

Reviewer #1:

Remarks to the Author:

The revised manuscript is much improved, and I can see that the authors have made an honest effort to address my specific concerns. They have added several new experiments addressing quality control issues, and improved figures and text throughout.

My only remaining concerns are about novelty and generality

1. As I wrote in my original comments, counting co-stained doublets is not novel. Previous mRNA-based doublet methods derived their claim to originality on using the highly sensitive transcriptional profile to find new states and boost sensitivity. This method introduces an automated clustering step to the co-staining protocol - is this novel enough to merit publication in Nature Methods?

We thank the reviewer for their comment and would like to take this opportunity to outline the novelty and broader impact of our approach, which extends well beyond the points mentioned, comprising a set of technical, computational and conceptual novelties.

Technical and Computational Novelty

Our approach encompasses a combination of experimental and computational steps that, for the first time, enable efficient and unbiased cytometry-based cellular interaction mapping across entire cellular interaction networks. This includes:

1. Data-derived ultra-high plex cytometry panels capable of mapping cellular interactions across a variety of cell types, organs and model systems.
2. Rational extraction of cytometry-based parameters associated with cellular interactions.
3. Implementation of a ready-to-use, highly sensitive, performant and benchmarked computational framework for the analysis of large-scale cytometry datasets spanning multiple tens of millions of events, while retaining rare interactions.
4. Statistical framework for interpreting cellular interactions.

Conceptual Novelty

- Therapy response prediction: For the first time, we demonstrate that cellular interaction networks serve as powerful predictors of clinical responses in immunotherapies — a topic of critical medical relevance and broad impact.
- Insights into autoimmune diseases: For the first time, we uncover significant differences in cellular interactions within the context of autoimmune diseases, providing critical insights that could explain disease pathology.
- Organism-wide insights: Our work maps organism-wide cellular interaction networks in response to viral infections, offering unprecedented insights into systemic cellular immune cell dynamics.

Broad Impact

- **Wide applicability:** Our approach has broad utility across multiple fields and is immediately applicable to various research contexts.
- **Cost-effective and high-throughput:** The methodology is designed to reduce costs while enabling ultra high-throughput, making it accessible to a wider research community, and a prime candidate for implementation into clinical routines.
- **Enhanced insights from existing data:** By unlocking additional layers of information from existing datasets, our approach amplifies the value of previously collected data, enabling deeper and more comprehensive analyses.

This combination of technical, computational, and conceptual advances underscores the significant novelty and impact of our work.

2. How general is the method? Immune cells are incredibly well defined by their cell surface markers - would you expect this method to be largely domain specific? (ie Rev. 3, point 6). If you wanted to study epithelial cell interactions in brain, or lung, or skin, some tumor, etc, would this method be of any use?

Our method is broadly applicable across a wide range of applications and research fields. In the revised manuscript, we showcase compelling use cases spanning multiple domains, including:

1. Infectious diseases
2. Autoimmune diseases
3. Hematology
4. Immunotherapies
5. Personalized medicine

The revised manuscript provides examples of cellular interaction mapping of more than 50 interaction types among 18 cell types in mice and of more than 30 interaction types among 14 cell types in humans, encompassing a variety of organs such as lymph nodes, spleen, blood, synovial fluid, and bone marrow.

Additionally, in unpublished work from our laboratory, we have demonstrated the versatility of this approach in fields such as tumor immunology, bone regeneration, and inflammation.

While new setups may require adaptations, our method is fundamentally applicable to all cells, tissues, and organs that can be analyzed using flow cytometry. In the revised version of the manuscript, we have now included an analysis of a published mouse intestinal epithelium dataset (Funk et al., 2023). Here, we demonstrate that our approach can not only be used to map cellular interactions across immune cells but to also measure interactions with epithelial cells (see Supplementary Figure 5).

Our method is specifically tailored for mapping dynamic interaction networks among cells using flow cytometry, making compatibility of tissues and cells with cytometry an obvious requirement. For cellular structures in static tissues, such as neurons in the brain, spatial

omics approaches are more appropriate. However, this limitation is not unique to our method, but applies universally to all techniques in this category.

Importantly, our approach is specifically designed to address a critical gap by enabling the mapping of dynamically changing cellular interaction networks at unprecedented scale, particularly in non-static tissues which are inaccessible to spatial techniques. The ability of our approach for ultra-high throughput mapping of dynamic cellular interactions represents a key strength of our method.

I am not ultimately convinced of either of these points, but it is really a question of degrees rather than absolutes. I will simply defer to the editor to judge how the novelty/generality issues should impact publishing in this case.

Reviewer #2:

Remarks to the Author:

The authors did a major revision, adding substantial and interesting new experiments and analysis and very thorough responses to my and other reviewer's questions. I think the paper can be published. At the discretion of the editor and the authors, they may decide to take into account my remaining concerns described below.

My main remaining concern, expressed also in my previous review, is the relevance of the cell-cell interactions detected by this method for the physiological *in vivo* contexts. New experiments and analysis presented by the authors in Fig S4 and S12 confirm that this is an important concern and limitation of the method. Via a clever labeling with different labels for otherwise equivalent cells, in several experiments they determine a substantial portion of interactions that are a result of the *ex vivo* sample manipulation, including cell culture, stimulation etc.

(By the way, it is inconvenient and confusing that seemingly the same exact figures are included several times with different identifiers, e.g. S4, R3, R11 etc.)

For example, in Figure S4, the authors show the results where doublet cells labeled with two different CD45 labels indicate cell interactions formed after cell isolation during cell stimulation treatment. They focus mostly on T-B cell interactions, and indeed the number of such newly acquired interactions is relatively small, though clearly noticeable. However, upon inspection of their results, for example it seems that for CD4 T - CD8 T cell interactions (see UMAP in Fig S4F), the number of newly acquired interactions is actually very large, comparable with other single-CD45-label interactions. This also suggests that the number (or fraction) of such artificial non-physiological interactions is variable depending on the interacting cell types, making it even more difficult methodologically.

We thank the reviewer for their comment and agree with their observations. We focused on B*T interactions as these are the most prevalent interactions upon CytoStim™ treatment. The observation that other interactions show little to no difference between single positive and double-positive interaction frequencies mainly applies to background interactions such as interactions between CD4+ and CD8+ T cells, which do not increase specifically upon

CytoStim™ treatment. In general, we recommend focusing on case-control settings, and to only regard interactions that show significant changes relative to the control.

It is even more problematic in the results in Fig S12. This is an experiment for LCMV infection, with a similar idea of using two CD45 labels for identifying double-labeled CD45.1-CD45.2 interactions that are definitely a result of *ex vivo* manipulation, i.e. not obtained *in vivo*. In this experiment, the log₂FC of interaction intensity in infection vs. control is a measure of biological enrichment of cell-cell interactions of different types. It is striking that this measure for double-labeled interactions is very highly correlated with such measure for single-labeled interactions, see Fig S12F. This suggests that single-labeled interactions cannot be truly distinguished from those obtained as technical artifacts, for any pair of the detected interacting cell types. This makes the physiological relevance of the entire experiment questionable.

I agree with the authors that this problem is common for a family of related methods, including their method, PIC-seq and others. Their new data helps understand this problem very well.

We sincerely thank the reviewer for their thoughtful feedback and concerns. The proposed approach captures the current state of cellular interactions, encompassing both newly acquired and *in vivo*-derived interactions. Notably, our data demonstrate that *in vivo*-derived interactions are non-random and serve as reliable proxies for underlying biological processes, since significant changes are observed relative to the control. To address potential concerns, we have included a detailed section outlining limitations and providing recommendations for minimizing technical artifacts while maximizing biological signals. Additionally, we refer to Supplementary Note 1 and 2 on *ex vivo* and *in vivo* benchmarking, offering detailed insights into the experimental conditions that influence technical interactions.

Some specific suggestions:

- In guidelines and limitations for applying the method, that are now included in the paper, they may specifically include a strong recommendation to always include the types of controls used in Fig S3 and S12. All the cell-cell interactions detected with the proposed new method should be compared against a control set of interactions that are clearly obtained as a result only of *ex vivo* manipulations. The biological relevance of interactions obtained in experiments without such controls is questionable, in my opinion.

We agree with the reviewer that the co-labeling strategy is a valuable tool to assess which interactions have formed during the sample preparation process. Whenever feasible, users of our framework are encouraged to include a similar setup during their adaptation of the method *in vivo*. We have now included such a statement into the manuscript. That said, we believe that in many scenarios, meaningful conclusions about changes in interactions can still be drawn without the more advanced co-labeling setup, provided a robust case-control experimental design is established, as detailed in our manuscript.

- Experiments in Fig S3 and S12 are very interesting, but data is underanalyzed and presented not fully. I encourage the authors to include all the raw and processed data from these experiments, in figure and table formats, so that readers and future users of the method can decide themselves about potential use cases and limitations. For example, see my comment

about CD4 T - CD8 T cell interactions - it would be nice to quantify this and all similar values, in all kinds of reasonable ways, to help understand the method and the data.

We agree with the reviewer that data can be analyzed further, exceeding the scope of our manuscript. We provide raw data in form of FCS files, processed csv files, and the respective code for the benchmarking experiments to enable further analysis of our data by readers and future users.

- In fact, I encourage the authors to include raw and processed data, at different levels of preprocessing, from all of their very interesting experiments presented in the paper (not only Fig S3 and S12) to increase the impact of their study, both the method and the many results they already obtained.

We have uploaded the key results, tables and accompanying analysis code.

Reviewer #3:

Remarks to the Author:

In revision, the authors have undertaken a comprehensive set of experiments and clarifications, which address my comments. In particular, there are now numerous technical controls as well as a limitations section, which advises future users on parameters to optimize. I'm also more convinced the approach is fairly robust to noise. There are also important additions demonstrating that previously acquired data can be analyzed by the approach.

I also believe the approach will have broad utility because of its flexibility and accessibility, which is actually unique among all interaction-based approaches, which tend to be complicated.

One lingering minor point left to the discretion of the editor and authors is the name, as "Interactomics" I think isn't descriptive enough for the method, which doesn't assess all interactions and is neither the first nor only method to study interactions. Could it be called PICtR?

We thank the reviewer for recognizing the broad utility of our method. The name "Interactomics" was originally chosen to emphasize its ultra-high throughput and ability to map interactions across a wide range of cell types simultaneously. Alternatively, the method could be referred to more descriptively as cytometry-based cellular interaction mapping. We are happy to defer to the editor's discretion regarding the preferred terminology.

I otherwise have no additional points to raise.